# Sample Complexity of Forecast Aggregation

**Tao Lin**
Harvard University
Cambridge, MA 02138
tlin@g.harvard.edu

**Yiling Chen**
Harvard University
Cambridge, MA 02138
yiling@seas.harvard.edu

## Abstract

We consider a Bayesian forecast aggregation model where $n$ experts, after observing private signals about an unknown binary event, report their posterior beliefs about the event to a principal, who then aggregates the reports into a single prediction for the event. The signals of the experts and the outcome of the event follow a joint distribution that is unknown to the principal, but the principal has access to i.i.d. "samples" from the distribution, where each sample is a tuple of the experts' reports (not signals) and the realization of the event. Using these samples, the principal aims to find an $\varepsilon$-approximately optimal aggregator, where optimality is measured in terms of the expected squared distance between the aggregated prediction and the realization of the event. We show that the sample complexity of this problem is at least $\tilde{\Omega}(m^{n-2}/\varepsilon)$ for arbitrary discrete distributions, where $m$ is the size of each expert's signal space. This sample complexity grows exponentially in the number of experts $n$. But, if the experts' signals are independent conditioned on the realization of the event, then the sample complexity is significantly reduced, to $\tilde{O}(1/\varepsilon^2)$, which does not depend on $n$. Our results can be generalized to non-binary events. The proof of our results uses a reduction from the distribution learning problem and reveals the fact that forecast aggregation is almost as difficult as distribution learning.

## 1 Introduction

Suppose you want to know whether it will rain tomorrow. A Google search on "weather" returns 40% probability of raining. The weather forecasting app on your phone shows 85%. And one of your friends, who is an expert in meteorology, predicts 65%. How do you aggregate these different predictions into a single, accurate prediction? This problem is called *forecast aggregation* or *forecast combination* [7, 16, 45]. It has innumerable applications in fields ranging from economics, statistics, operations research, to machine learning, decision theory, and of course, climate science.

A straightforward solution to forecast aggregation is to take the (unweighted) average of all experts' forecasts. Simple as it is, unweighted average performs surprisingly well in practice, as observed by many forecast aggregation works from 1960s to 2020s (e.g., [7, 38, 35, 30, 16, 45, 34]). Naturally, when past data of expert forecasts and event outcomes (e.g., historical weather forecasts and outcomes) are available, one may hope to leverage on such data to learn more accurate aggregators. While adjusting the weights of experts in averaging according to their individual historical accuracy has led to improved accuracy for the aggregated prediction [49], interestingly, more sophisticated data-driven methods [23, 35, 49] were often outperformed by the unweighted average. The dominance of simple aggregators over more sophisticated data-driven methods is observed so often in empirical applications that it is termed "the forecast combination puzzle" [44, p.428].

There are many potential explanations for the forecast combination puzzle [43, 23, 15]. In some scenarios, the past events are different from the future events in fundamental ways (e.g., geopolitical

forecasting) and hence past data may not be informative in predicting the future. Another widely accepted conjecture is that the amount of past data is not large enough for a data-intensive method to perform well. Indeed, the sample sizes in many empirical forecast aggregation works are small under today's "big-data" standard (24 in [27], 69 in [42], 87 in [6], 100 in [30]). However, there are aggregation settings where future events are arguably similar to past events and we do have abundant data — for instance, the forecasting of weather, stock prices [21], and some forecasting competitions on Kaggle[1]. Such settings are well-suited for data-driven methods. It is hence tempting to ask *how many* data are needed for data-driven aggregators to achieve high accuracy in these settings.

In this paper, we initiate the study of *sample complexity of forecast aggregation*, building upon a standard Bayesian forecasting model [37, 48, 24]. In this model, there are $n$ experts who observe private signals about an unknown binary event and make posterior predictions about the event. The experts' signals and the event are jointly distributed according to some underlying unknown distribution (information structure) $P$, which determines the optimal way to aggregate the experts' predictions. With sample access to the unknown distribution $P$, we ask the following question:

> *How many samples do we need to approximately learn the optimal aggregator?*

Our model favors the use of data-driven aggregation methods because historical tasks are i.i.d. as future tasks. We show that however, even in this benign model, optimal aggregation in general needs *exponentially many* samples. One may thus expect that data-driven methods can hardly perform well in more realistic and non-i.i.d. scenarios. Nevertheless, for some special, yet interesting, families of information structures, the sample complexity of forecast aggregation is significantly lower.

**Main results**    (1) If $P$ can be an arbitrary discrete distribution, then at least $\tilde{\Omega}(m^{n-2}/\varepsilon)$ samples are needed, and $\tilde{O}(m^n/\varepsilon^2)$ samples are sufficient, to learn an $\varepsilon$-optimal aggregator with high probability, where $m$ is the number of signals an expert can possibly observe.[2] (2) If the experts' signals are conditionally independent, then the sample complexity is exponentially reduced and surprisingly does not depend on the number of experts or the number of signals: $\tilde{O}(1/\varepsilon^2)$ samples are sufficient, and $\tilde{\Omega}(1/\varepsilon)$ samples are necessary, to learn an $\varepsilon$-optimal aggregator with high probability.

**Main techniques**    The main technical part of our paper is to prove the $\tilde{\Omega}(m^{n-2}/\varepsilon)$ lower bound on the sample complexity of forecast aggregation for the general case, via a reduction from the *distribution learning* problem. It is known that learning a discrete distribution with support size $|S|$ in total variation distance $\varepsilon$ requires $\tilde{\Omega}(|S|/\varepsilon^2)$ samples. By reducing distribution learning to forecast aggregation, we obtain the lower bound on the sample complexity of the latter problem. This reduction is highly nontrivial. To do this reduction we define a new distribution learning problem that is different from the one in the literature, which is of independent interest. This reduction also reveals an interesting fact: learning to aggregate optimally on some distribution is almost as difficult as learning the distribution itself. This is a little surprising because one might initially think that aggregation should be easier than distribution learning – we show that this is not the case.

**Structure of the paper**    We discuss related works in Section 1.1. Section 2 introduces our model. Section 3 includes some preliminaries, including the distribution learning problem, which we will use in the proof. Section 4 studies the sample complexity for general distributions. Section 5 focuses on the conditional independence case. Section 6 summarizes how our results can be generalized to non-binary (multi-outcome) events. Section 7 concludes and discusses future directions.

## 1.1 Related Works

**Data-driven aggregation**    Data-driven approaches to forecast aggregation date back to perhaps the first paper on forecast aggregation [7] and have been a standard practice in the literature (see, e.g., surveys [16, 45] and more recent works [51, 42, 4, 47, 22, 40]). Many of these works focus on specific weighted average aggregators like *linear pooling* (weighted arithmetic mean) [51] and *logarithmic pooling* (normalized weighted geometric mean) [42, 40], and the goal is to estimate the optimal weights from data. However, those weighted average aggregators are not necessarily the

---

[1] https://www.kaggle.com/c/m5-forecasting-accuracy

[2] The $\tilde{O}(\cdot)$ and $\tilde{\Omega}(\cdot)$ notations omit logarithmic factors.

optimal (*Bayesian*) aggregator unless the underlying information structure satisfies some strict conditions (e.g., experts' forecasts are equal to the true probability of the event plus normally distributed noises [48]). Our work aims to understand the sample complexity of Bayesian aggregation, namely how many samples are needed to approximate the Bayesian aggregator.

Our work is closely related to a recent work [4] on Bayesian aggregation via *online learning*. For the case of conditionally independent experts, [4] shows that the Bayesian aggregator can be approximated with $\varepsilon = \tilde{O}(\frac{n}{\sqrt{T}})$ regret. By an online-to-batch conversion, this implies a $T = \tilde{O}(\frac{n^2}{\varepsilon^2})$ sample complexity upper bound for the batch learning setting. Our paper studies the batch learning setting. For the conditional independence case, we obtain an improved bound of $\tilde{O}(\frac{1}{\varepsilon^2})$.

**Robust forecast aggregation**   Recent works on "robust forecast aggregation" [2, 39, 18, 3] also study information aggregation problems where the principal does not know the underlying information structure. They take a worst-case approach, assuming that the information structure is chosen adversarially. This often leads to negative results: e.g., a bad approximation ratio [2, 39] or a degenerate maximin aggregator that solely relies on a random expert's opinion [18, 3]. In contrast, we assume sample access to the unknown information structure. Our sample complexity approach is orthogonal and complementary to the robust forecast aggregation approach.

**Sample complexity of mechanism design**   Our work may remind the reader of a long line of research on the sample complexity of revenue maximization in mechanism design (e.g., [17, 20, 36, 19, 5, 11, 29, 10, 25, 50, 28]). In particular, [28] gives a general framework to bound the sample complexity for mechanism design problems that satisfy a "strong monotonicity" property, but this property is not satisfied by our forecast aggregation problem. A key observation from this line of works is that the number of samples needed to learn an $\varepsilon$-optimal auction for $n \geq 1$ independent bidders is increasing in $n$, because when there are more bidders, although we can obtain higher revenue, the optimal auction benchmark is also improved. We see a similar phenomenon that the sample complexity of forecast aggregation increases with the number of experts in the general case, but not in the case where experts are conditionally independent.

## 2   Model

### 2.1   Forecast Aggregation

There are $n \geq 2$ experts and one principal. The principal wants to predict the probability that a binary event $\omega \in \Omega = \{0, 1\}$ happens ($\omega = 1$), based on information provided by the experts. For example, $\omega$ may represent whether it will rain tomorrow. We present binary events to simplify notations. All our results can be generalized to multi-outcome events with $|\Omega| > 2$ (see Section 6). We also refer to $\omega$ as "the state of the world". Each expert $i = 1, \ldots, n$ observes some private signal $s_i \in \mathcal{S}_i$ that is correlated with $\omega$, where $\mathcal{S}_i$ denotes the space of all possible signals of expert $i$. We assume for now that $\mathcal{S}_i$ is finite, with size $|\mathcal{S}_i| = m$. We relax this assumption in Section 5 where we consider conditionally independent signals. Let $\mathcal{S} = \mathcal{S}_1 \times \cdots \times \mathcal{S}_n$ be the joint signal space of all experts; $|\mathcal{S}| = m^n$. Let $P$ be a distribution over $\mathcal{S} \times \Omega$, namely, a joint distribution of signals $\boldsymbol{s} = (s_1, \ldots, s_n)$ and event $\omega$. Since the space $\mathcal{S} \times \Omega$ is discrete, we can use $P(\cdot)$ to denote the probability: $P(\boldsymbol{s}, \omega) = \Pr_P[\boldsymbol{s}, \omega]$. Signals of different experts can be correlated conditioned on $\omega$. We assume that each expert $i$ knows the marginal joint distribution of their own signal $s_i$ and $\omega$, $P(s_i, \omega)$. Neither any expert nor the principal knows the entire distribution $P$. Each expert $i$ reports to the principal a forecast (or prediction) $r_i$ for the event $\omega$, which is equal to the conditional probability of $\omega = 1$ given their signal $s_i$:[3]

$$r_i = P(\omega = 1 \mid s_i) = \tfrac{P(\omega=1)P(s_i|\omega=1)}{P(\omega=1)P(s_i|\omega=1)+P(\omega=0)P(s_i|\omega=0)}. \tag{1}$$

We note that $r_i$ depends on $s_i$ and $P$, but not on $\omega$ or other experts' signals $\boldsymbol{s}_{-i}$. Let $\boldsymbol{r} = (r_1, \ldots, r_n) \in [0, 1]^n$ denote the reports (joint report) of all experts. We sometimes use $\boldsymbol{r}_{-i}$ to

---

[3]One may wonder whether the experts are willing to report $r_i = P(\omega = 1 \mid s_i)$ *truthfully*. This can be guaranteed by a *proper scoring rule*. For example, we can reward each expert the Brier score $C - |r_i - \omega|^2$ after seeing the realization of $\omega$ [9]. Each expert maximizes its expected reward by reporting its belief truthfully.

denote the reports of all experts except $i$. The principal aggregates the experts' reports $r$ into a single forecast $f(r)$ using some *aggregation function*, or *aggregator*, $f : [0,1]^n \to [0,1]$. We measure the performance of an aggregator by the mean squared loss:

$$L_P(f) = \mathbb{E}_P\big[|f(r) - \omega|^2\big]. \tag{2}$$

The notation $\mathbb{E}_P[\cdot]$ makes it explicit that the expectation is over the random draw of $(s, \omega) \sim P$ followed by letting $r_i = P(\omega = 1 \mid s_i)$. We omit $P$ and write $\mathbb{E}[\cdot]$ when it is clear from the context.

Let $f^*$ be the optimal aggregator with respect to $P$, which minimizes $L_P(f)$:

$$f^* = \operatorname*{argmin}_{f:[0,1]^n \to [0,1]} L_P(f) = \operatorname*{argmin}_{f:[0,1]^n \to [0,1]} \mathbb{E}_P\big[|f(r) - \omega|^2\big]. \tag{3}$$

We have the following characterization of $f^*$ and $L_P(f)$: $f^*$ is equal to the Bayesian aggregator, which computes the posterior probability of $\omega = 1$ given all the reports $r = (r_1, \ldots, r_n)$. And the difference between the loss of $f$ and the loss of $f^*$ is equal to their expected squared difference.

**Lemma 2.1.** *The optimal aggregator $f^*$ and any aggregator $f$ satisfy:*

- $f^*(r) = P(\omega = 1 \mid r)$, *for almost every $r$.*
- $L_P(f) - L_P(f^*) = \mathbb{E}_P\big[|f(r) - f^*(r)|^2\big].$

An aggregator $f$ is $\varepsilon$-*optimal* (with respect to $P$) if $L_P(f) \leq L_P(f^*) + \varepsilon$. By Lemma 2.1, this is equivalent to $\mathbb{E}_P\big[|f(r) - f^*(r)|^2\big] \leq \varepsilon$. For an $\varepsilon$-optimal $f$, we also say it $\varepsilon$-*approximates* $f^*$.

**Definition 2.2.** *An aggregator $f$ is $\varepsilon$-*optimal *(with respect to $P$) if* $\mathbb{E}_P\big[|f(r) - f^*(r)|^2\big] \leq \varepsilon$.

**Discussion of the benchmark $f^*$** Our benchmark, the Bayesian aggregator $f^*$, is common in the forecast aggregation literature (e.g., [27, 26, 45]). It is stronger than the typical "best expert" benchmark in no-regret learning (e.g., [13, 22]), but weaker than the "omniscient" aggregator that has access to the experts' *signals*: $f_{\text{omni}}(s) = P(\omega = 1 \mid s)$. If there is a one-to-one mapping between signals $s$ and reports $r$, then $f_{\text{omni}}$ and $f^*$ are the same. Otherwise, $f_{\text{omni}}$ could be much stronger than $f^*$ and an $\varepsilon$-approximation to $f_{\text{omni}}$ using experts' reports only is not always possible.[4] In contrast, an $\varepsilon$-approximation to $f^*$ is always achievable (in fact, achieved by $f^*$ itself). The difference between $f^*$ and $f_{\text{omni}}$ is known as the difference between "aggregating forecasts" and "aggregating information sets" in the literature [27, p.198-199], [26, p.168-169], [45, p.143].

## 2.2 Sample Complexity of Forecast Aggregation

The principal has access to $T$ i.i.d. samples of forecasts and event realizations drawn from the underlying unknown distribution $P$:

$$S_T = \big\{(r^{(1)}, \omega^{(1)}), \ldots, (r^{(T)}, \omega^{(T)})\big\}, \quad (s^{(t)}, \omega^{(t)}) \sim P, \quad r_i^{(t)} = P(\omega = 1 \mid s_i^{(t)}). \tag{4}$$

Here, we implicitly regard $P$ as a distribution over $(r, \omega)$ instead of $(s, \omega)$. The principal uses samples $S_T$ to learn an aggregator $\hat{f} = \hat{f}_{S_T}$, in order to approximate $f^*$. Our main question is:

> *How many samples are necessary and sufficient for finding an $\varepsilon$-optimal aggregator $\hat{f}$*
> *(with probability at least $1 - \delta$ over the random draw of samples)?*

The answer to the above question depends on the family of distributions we are interested in. Let $\mathcal{P}$ denote a family of distributions over $\boldsymbol{S} \times \Omega$. It could be the set of all distributions over $\boldsymbol{S} \times \Omega$, or in Section 5 we will only consider distributions where the signals are independent conditioned on $\omega$. We define the sample complexity of forecast aggregation, with respect to $\mathcal{P}$, formally:

**Definition 2.3.** *The* sample complexity of forecast aggregation *(with respect to $\mathcal{P}$) is the minimum function $T_{\mathcal{P}}(\cdot, \cdot)$ of $\varepsilon, \delta \in (0, 1)$, such that: if $T \geq T_{\mathcal{P}}(\varepsilon, \delta)$, then for any distribution $P \in \mathcal{P}$, with probability at least $1 - \delta$ over the random draw of $T$ samples $S_T$ from $P$ (and over the randomness of the learning procedure if it is randomized), we can obtain an aggregator $\hat{f} = \hat{f}_{S_T}$ satisfying $\mathbb{E}_P[|\hat{f}(r) - f^*(r)|^2] \leq \varepsilon$.*

---

[4]This has been noted by [2, 4, 39]. They give an XOR example where $\omega = s_1 \oplus s_n$, $s_1$ and $s_2$ are i.i.d. Uniform$\{0, 1\}$ distributed. The experts always report $r_i = 0.5$, $f_{\text{omni}}(s_1, s_2) = \omega$, $f^*(r_1, r_2) = 0.5$, so $L_P(f_{\text{omni}}) = 0$ but $L_P(f^*) = 0.25 > 0$. No aggregator that uses experts' reports only can do better than $f^*$.

The principal is assumed to know the family of distributions $\mathcal{P}$ but not the specific distribution $P \in \mathcal{P}$. There should be at least two different distributions in $\mathcal{P}$. Otherwise, the principal knows what the distribution is and there is no need for learning.

# 3 Preliminaries

In this section, we briefly introduce some notions that will be used in our analysis of the sample complexity of forecast aggregation, including some definitions of distances between distributions, the distribution learning problem, and the distinguishing distributions problem.

**Distances between distributions** We recall two distance metrics for discrete distributions: the *total variation distance* and the *(squared) Hellinger distance*.

**Definition 3.1.** *Let $D_1, D_2$ be two distributions on a discrete space $\mathcal{X}$.*

- *The* total variation distance between $D_1$ and $D_2$ is $d_{\mathrm{TV}}(D_1, D_2) = \frac{1}{2} \sum_{x \in \mathcal{X}} |D_1(x) - D_2(x)|$.
- *The* squared Hellinger distance *between $D_1$ and $D_2$ is* $d_{\mathrm{H}}^2(D_1, D_2) = \frac{1}{2} \sum_{x \in \mathcal{X}} \left(\sqrt{D_1(x)} - \sqrt{D_2(x)}\right)^2 = 1 - \sum_{x \in \mathcal{X}} \sqrt{D_1(x)D_2(x)}$.

The total variation distance has the following well-known property that upper bounds the difference between the expected values of a function on two distributions:

**Fact 3.2.** *For any function $h : \mathcal{X} \to [0,1]$, $|\mathbb{E}_{x \sim D_1} h(x) - \mathbb{E}_{x \sim D_2} h(x)| \leq d_{\mathrm{TV}}(D_1, D_2)$.*

In Appendix A we give some properties of the Hellinger distance that will be used in the proofs.

**Distribution learning in total variation distance** Our analysis of the sample complexity of the forecast aggregation problem will leverage on the sample complexity of another learning problem: *learning discrete distributions in total variation distance*. We review this problem below.

Let $\mathcal{D}$ be a family of distributions over $\mathcal{X}$. The *sample complexity of learning distributions in $\mathcal{D}$ within total variation distance $\varepsilon$* is the minimum function $T_{\mathcal{D}}^{\mathrm{TV}}(\varepsilon, \delta)$, such that: if $T \geq T_{\mathcal{D}}^{\mathrm{TV}}(\varepsilon, \delta)$, then for any distribution $D \in \mathcal{D}$, with probability at least $1 - \delta$ over the random draw of $T$ samples from $D$, we can obtain (from the $T$ samples) a distribution $\hat{D}$ such that $d_{\mathrm{TV}}(\hat{D}, D) \leq \varepsilon$.

**Proposition 3.3** (e.g., [12, 33]). *Let $\mathcal{D}_{\mathrm{all}}$ be the set of all distributions over $\mathcal{X}$. Then, $T_{\mathcal{D}_{\mathrm{all}}}^{\mathrm{TV}}(\varepsilon, \delta) = \Theta\left(\frac{|\mathcal{X}| + \log(1/\delta)}{\varepsilon^2}\right)$. In particular, the upper bound can be achieved by using the empirical estimate $\hat{D}_{\mathrm{emp}}$ (which is the uniform distribution over the $T$ samples). The lower bound holds regardless of what learning algorithm is used.*

**Distinguishing distributions** Another learning problem that we will leverage on is the problem of *distinguishing (two) distributions*: given samples from a distribution randomly chosen from $\{D_1, D_2\}$, we are to guess whether the samples are from $D_1$ or $D_2$. The sample complexity of distinguishing distributions is characterized by the squared Hellinger distance. It is known that at least $T = \Omega\left(\frac{1}{d_{\mathrm{H}}^2(D_1, D_2)} \log \frac{1}{\delta}\right)$ samples are needed to distinguish two distributions with probability at least $1 - \delta$. See Appendix A for a formal statement of this result.

# 4 Sample Complexity for General Distributions

In this section we characterize the sample complexity of forecast aggregation for general distributions $P$. We give an exponential (in the number of experts, $n$) upper bound and an exponential lower bound on the sample complexity, as follows:

**Theorem 4.1.** *Let $\mathcal{P}_{\mathrm{all}}$ be the set of all distributions over $\mathcal{S} \times \Omega$, with $|\mathcal{S}| = m^n$. Suppose $n \geq 2$. The sample complexity of forecast aggregation with respect to $\mathcal{P}_{\mathrm{all}}$ is*

$$O\left(\frac{m^n + \log(1/\delta)}{\varepsilon^2}\right) \geq T_{\mathcal{P}_{\mathrm{all}}}(\varepsilon, \delta) \geq \Omega\left(\frac{m^{n-2} + \log(1/\delta)}{\varepsilon}\right). \tag{5}$$

This theorem is for $n \geq 2$. When there is only one expert ($n = 1$), there is no need to learn to aggregate because the optimal "aggregator" $f^*$ simply outputs the forecast given by the only expert: $f^*(r_1) = P(\omega = 1 \mid r_1) = P(\omega = 1 \mid s_1) = r_1$. The sample complexity is 0 in this case.

There is a gap in the dependency on $\varepsilon$ in the upper bound and the lower bound in Theorem 4.1. We conjecture that the tight dependency on $\varepsilon$ should be $\frac{1}{\varepsilon}$ (so the lower bound is tight). See Section 7 for a detailed discussion of this conjecture, where we show that the dependency on $\varepsilon$ in the upper bound can be improved to $\frac{1}{\varepsilon}$ for a large family of distributions.

## 4.1 Proof of the Upper Bound

In this subsection we prove the $O\left(\frac{m^n + \log(1/\delta)}{\varepsilon^2}\right)$ upper bound in Theorem 4.1. This is a direct corollary of the distribution learning result introduced in Section 3.

We regard $P$ as a distribution over $\boldsymbol{r}$ and $\omega$ instead of over $\boldsymbol{s}$ and $\omega$. Then $P$ is a discrete distribution with support size at most $2m^n$ because each possible report $r_i \in [0,1]$ corresponds to some discrete signal $s_i$ in $\mathcal{S}_i$. Let $\hat{P}_{\mathrm{emp}}$ be the empirical distribution of reports and event realizations: $\hat{P}_{\mathrm{emp}} = \mathrm{Uniform}\{(\boldsymbol{r}^{(1)}, \omega^{(1)}), \dots, (\boldsymbol{r}^{(T)}, \omega^{(T)})\}$. By Proposition 3.3, with probability at least $1 - \delta$ over the random draw of $T = O\left(\frac{2m^n + \log(1/\delta)}{\varepsilon^2}\right)$ samples, we have $d_{\mathrm{TV}}(\hat{P}_{\mathrm{emp}}, P) \leq \varepsilon$. According to Fact 3.2, and by the definition of $L_P(f)$, we have: for any aggregator $f : [0,1]^n \to [0,1]$,

$$\left| L_{\hat{P}_{\mathrm{emp}}}(f) - L_P(f) \right| = \left| \mathbb{E}_{\hat{P}_{\mathrm{emp}}}\left[ |f(\boldsymbol{r}) - \omega|^2 \right] - \mathbb{E}_P\left[ |f(\boldsymbol{r}) - \omega|^2 \right] \right| \leq d_{\mathrm{TV}}(\hat{P}_{\mathrm{emp}}, P) \leq \varepsilon.$$

Therefore, if we pick the empirically optimal aggregator $\hat{f}_{\mathrm{emp}} = \mathrm{argmin}_f\, L_{\hat{P}_{\mathrm{emp}}}(f)$, we get

$$L_P(\hat{f}_{\mathrm{emp}}) \leq L_{\hat{P}_{\mathrm{emp}}}(\hat{f}_{\mathrm{emp}}) + \varepsilon \leq L_{\hat{P}_{\mathrm{emp}}}(f^*) + \varepsilon \leq L_P(f^*) + 2\varepsilon,$$

which means that $\hat{f}_{\mathrm{emp}}$ is a $2\varepsilon$-optimal aggregator for $P$.

## 4.2 Proof of the Lower Bound

In this subsection we prove the $\Omega\left(\frac{m^{n-2} + \log(1/\delta)}{\varepsilon}\right)$ lower bound in Theorem 4.1. The main idea is a reduction from the distribution learning problem (defined in Section 3) for a specific family $\mathcal{D}$ of distributions over the joint signal space $\boldsymbol{\mathcal{S}} = \mathcal{S}_1 \times \cdots \times \mathcal{S}_n$. We construct a corresponding family of distributions $\mathcal{P} = \{P_D : D \in \mathcal{D}\}$ for the forecast aggregation problem, such that, if we can obtain an $\varepsilon$-optimal aggregator $\hat{f}$ for $P_D$, then we can convert $\hat{f}$ into a distribution $\hat{D}$ such that $d_{\mathrm{TV}}(\hat{D}, D) \leq O(\sqrt{\varepsilon})$. We then prove that learning $\mathcal{D}$ within total variation distance $\varepsilon_{\mathrm{TV}} = O(\sqrt{\varepsilon})$ requires $\Omega\left(\frac{m^{n-2} + \log(1/\delta)}{\varepsilon_{\mathrm{TV}}^2}\right) = \Omega\left(\frac{m^{n-2} + \log(1/\delta)}{\varepsilon}\right)$ samples. This gives the sample complexity lower bound for the forecast aggregation problem for $\mathcal{P}$ (and hence $\mathcal{P}_{\mathrm{all}}$).

We will need a family of distributions $\mathcal{D}$ that satisfies the following three properties:

**Definition 4.2.** *We say a family of distributions $\mathcal{D}$ satisfies*

1. *$B$-uniformly bounded, if: $D(\boldsymbol{s}) \leq \frac{B}{|\boldsymbol{\mathcal{S}}|} = \frac{B}{m^n}, \forall \boldsymbol{s} \in \boldsymbol{\mathcal{S}}, \forall D \in \mathcal{D}$, where $B \geq 1$ is a constant.*

2. *same marginal across distributions, if: for any $D, D' \in \mathcal{D}$, any $i$, any $s_i \in \mathcal{S}_i$, $D(s_i) = D'(s_i)$.*

3. *distinct marginals across signals, if: for any $D \in \mathcal{D}$, any $i$, any $s_i \neq s_i' \in \mathcal{S}_i$, $D(s_i) \neq D(s_i')$.*

How do we construct the family $\mathcal{P}$? For each distribution $D \in \mathcal{D}$, we construct distribution $P_D$ as follows: the marginal distribution of $\omega$ is $\mathrm{Uniform}\{0,1\}$, i.e., $P_D(\omega = 0) = P_D(\omega = 1) = \frac{1}{2}$; conditioning on $\omega = 0$, the joint signal $\boldsymbol{s}$ is uniformly distributed: $P_D(\boldsymbol{s} \mid \omega = 0) = \frac{1}{|\boldsymbol{\mathcal{S}}|} = \frac{1}{m^n}$, $\forall \boldsymbol{s} \in \boldsymbol{\mathcal{S}}$; conditioning on $\omega = 1$, the joint signal is distributed according to $D$: $P_D(\boldsymbol{s} \mid \omega = 1) = D(\boldsymbol{s}), \forall \boldsymbol{s} \in \boldsymbol{\mathcal{S}}$. The family $\mathcal{P}$ is $\{P_D : D \in \mathcal{D}\}$.

We show that if we can obtain $\varepsilon$-optimal aggregators for distributions in $\mathcal{P}$, then we can learn the distributions in $\mathcal{D}$ within total variation distance $(1 + B)^2 \sqrt{\varepsilon}$, and thus the sample complexity of the former is lower bounded by the sample complexity of the latter:

**Lemma 4.3.** *Let $\mathcal{D}$ be a family of distributions that satisfies the three properties in Definition 4.2. Let $\mathcal{P} = \{P_D : D \in \mathcal{D}\}$ be defined above. Then, $T_{\mathcal{P}}(\varepsilon, \delta) \geq T_{\mathcal{D}}^{\mathrm{TV}}\left((1 + B)^2 \sqrt{\varepsilon}, \delta\right)$.*

*Proof sketch of Lemma 4.3.* The full proof is in Appendix E.1. We give a sketch here. According to the definition of $P_D$, by Bayes' rule, we have

$$P_D(\omega = 1 \mid \boldsymbol{s}) = \frac{\frac{1}{2}P_D(\boldsymbol{s}|\omega=1)}{\frac{1}{2}P_D(\boldsymbol{s}|\omega=0)+\frac{1}{2}P_D(\boldsymbol{s}|\omega=1)} = \frac{D(\boldsymbol{s})}{\frac{1}{m^n}+D(\boldsymbol{s})}. \tag{6}$$

The "distinct marginals across signals" property in Definition 4.2 ensures that there is a one-to-one mapping between signal $s_i$ and report $r_i = P_D(\omega = 1 \mid s_i)$, and hence a one-to-one mapping between joint signal $\boldsymbol{s}$ and joint report $\boldsymbol{r}$. So, the Bayesian aggregator $f^*$ satisfies $f^*(\boldsymbol{r}) = P_D(\omega = 1 \mid \boldsymbol{r}) = P_D(\omega = 1 \mid \boldsymbol{s}) = \frac{D(\boldsymbol{s})}{1/m^n+D(\boldsymbol{s})}$. This gives

$$D(\boldsymbol{s}) = \frac{1}{m^n}\frac{f^*(\boldsymbol{r})}{1-f^*(\boldsymbol{r})}. \tag{7}$$

Suppose we have obtained an $\varepsilon$-optimal aggregator $\hat{f}$ for $P_D$, $\mathbb{E}_{P_D}\big[|\hat{f}(\boldsymbol{r}) - f^*(\boldsymbol{r})|^2\big] \leq \varepsilon$, then we convert $\hat{f}$ into $\hat{D}$ by letting $\hat{D}(\boldsymbol{s}) = \frac{1}{m^n}\frac{\hat{f}(\boldsymbol{r})}{1-\hat{f}(\boldsymbol{r})}$. The total variation distance between $\hat{D}$ and $D$ is:

$$d_{\mathrm{TV}}(\hat{D}, D) = \frac{1}{2}\sum_{\boldsymbol{s}\in\boldsymbol{\mathcal{S}}}|\hat{D}(\boldsymbol{s}) - D(\boldsymbol{s})| \overset{(7)}{=} \frac{1}{2}\sum_{\boldsymbol{s}\in\boldsymbol{\mathcal{S}}}\frac{1}{m^n}\left|\frac{\hat{f}(\boldsymbol{r})}{1-\hat{f}(\boldsymbol{r})} - \frac{f^*(\boldsymbol{r})}{1-f^*(\boldsymbol{r})}\right|. \tag{8}$$

The "$B$-uniformly bounded" property in Definition 4.2 ensures $D(\boldsymbol{s}) = O(\frac{1}{m^n})$, which has two consequences: (1) $P_D(\boldsymbol{s}) = O(\frac{1}{m^n})$; (2) $f^*(\boldsymbol{r}) = O(1)$, which implies $\left|\frac{\hat{f}(\boldsymbol{r})}{1-\hat{f}(\boldsymbol{r})} - \frac{f^*(\boldsymbol{r})}{1-f^*(\boldsymbol{r})}\right| = O\big(|\hat{f}(\boldsymbol{r}) - f^*(\boldsymbol{r})|\big)$ due to Lipschitz property of the function $\frac{x}{1-x}$ for $x = O(1)$. These two consequences imply

$$d_{\mathrm{TV}}(\hat{D}, D) = O\Big(\sum_{\boldsymbol{s}\in\boldsymbol{\mathcal{S}}}\frac{1}{m^n}|\hat{f}(\boldsymbol{r}) - f^*(\boldsymbol{r})|\Big) = O\Big(\sum_{\boldsymbol{s}\in\boldsymbol{\mathcal{S}}}P_D(\boldsymbol{s})|\hat{f}(\boldsymbol{r}) - f^*(\boldsymbol{r})|\Big)$$

$$= O\Big(\mathbb{E}\big[|\hat{f}(\boldsymbol{r}) - f^*(\boldsymbol{r})|\big]\Big) \overset{\text{Jensen's inequality}}{\leq} O\Big(\sqrt{\mathbb{E}\big[|\hat{f}(\boldsymbol{r}) - f^*(\boldsymbol{r})|^2\big]}\Big) = O(\sqrt{\varepsilon}).$$

So, we obtain $d_{\mathrm{TV}}(\hat{D}, D) \leq O(\sqrt{\varepsilon}) = (1 + B)^2\sqrt{\varepsilon}$.

There is a subtlety, however: In the distribution learning problem we are given samples from $D$, which are *signals* $\{\boldsymbol{s}^{(t)}\}_{t=1}^T$. We need to convert them into the corresponding *reports* $\{\boldsymbol{r}^{(t)}\}_{t=1}^T$ as the samples for the forecast aggregation problem. To do this we need to know $P_D$ or $D$, *which we do not know.* So, we make use of the "same marginal across distributions" property in Definition 4.2 here: because all the distributions $D \in \mathcal{D}$ have the same marginal probability $D(s_i) = D'(s_i)$ (but possibly different joint probabilities $D(\boldsymbol{s}) \neq D'(\boldsymbol{s})$), we are able to compute the report

$$r_i^{(t)} = P_D(\omega = 1 \mid s_i^{(t)}) = \frac{\frac{1}{2}P_D(s_i^{(t)}|\omega=1)}{\frac{1}{2}P_D(s_i^{(t)}|\omega=0)+\frac{1}{2}P_D(s_i^{(t)}|\omega=1)} = \frac{D(s_i^{(t)})}{\frac{1}{m}+D(s_i^{(t)})}$$

separately for each expert $i$ without knowing $D$, since we know what the family $\mathcal{D}$ is. This allows us to reduce the distribution learning problem for $\mathcal{D}$ to the forecast aggregation problem for $\mathcal{P}$. $\square$

Then, we find a family of distributions $\mathcal{D}$ that satisfies the three properties in Definition 4.2 and requires many samples to learn.

**Proposition 4.4.** *There exists a family of distributions $\mathcal{D}$ that satisfies the three properties in Definition 4.2 (with $B = e + \frac{1}{2}$) and requires $T_{\mathcal{D}}^{\mathrm{TV}}(\varepsilon_{\mathrm{TV}}, \delta) = \Omega\big(\frac{m^{n-2}+\log(1/\delta)}{\varepsilon_{\mathrm{TV}}^2}\big)$ samples to learn.*

The above sample complexity is smaller than the $\Omega\big(\frac{|\boldsymbol{\mathcal{S}}|+\log(1/\delta)}{\varepsilon_{\mathrm{TV}}^2}\big)$ lower bound in Proposition 3.3 because we are restricting to a smaller set of distributions than the set of all distributions over $\boldsymbol{\mathcal{S}}$. The proof of Proposition 4.4 is analogous to a textbook proof of Proposition 3.3, which uses reductions from the distinguishing distributions problem. See details in Appendix E.2.

**Finishing the proof of Theorem 4.1:** By Lemma 4.3 and Proposition 4.4, plugging in $\varepsilon_{\mathrm{TV}} = (1 + B)^2\sqrt{\varepsilon}$ with $B = e + \frac{1}{2}$, we obtain the lower bound on the sample complexity of forecast aggregation for $\mathcal{P}$ (and hence for $\mathcal{P}_{\mathrm{all}}$):

$$T_{\mathcal{P}}(\varepsilon, \delta) \geq T_{\mathcal{D}}^{\mathrm{TV}}\big((1+B)^2\sqrt{\varepsilon}, \delta\big) = \Omega\Big(\frac{m^{n-2}+\log(1/\delta)}{((1+B)^2\sqrt{\varepsilon})^2}\Big) = \Omega\big(\frac{m^{n-2}+\log(1/\delta)}{\varepsilon}\big).$$

# 5 Sample Complexity for Conditionally Independent Distributions

Section 4 proved that learning $\varepsilon$-optimal aggregators for all discrete distributions needs exponentially many samples. As shown in the proof, this large sample complexity is because the experts' signals can be arbitrarily correlated conditioned on the event $\omega$. Accurate estimation of such correlation requires many samples. So, in this section we restrict attentions to the case where the experts' signals are conditionally independent. It turns out that an $\varepsilon$-optimal aggregator can be learned using only $O(\frac{1}{\varepsilon^2} \log \frac{1}{\varepsilon \delta})$ samples in this case, which does not depend on $n$. The assumption of discrete signal space can be relaxed here. We also investigate two special and interesting families of conditionally independent distributions that admit an even smaller sample complexity of $O(\frac{1}{\varepsilon} \log \frac{1}{\delta})$.

## 5.1 General Conditionally Independent Distributions

Let $P$ be a conditionally independent distributions over $\boldsymbol{S} \times \Omega$, namely, $P(\boldsymbol{s} \,|\, \omega) = \prod_{i=1}^{n} P(s_i \,|\, \omega)$ for all $\boldsymbol{s} \in \boldsymbol{S}$, for $\omega \in \{0, 1\}$. Here, $\mathcal{S}_i$ can be a continuous space, in which case, $P(\cdot \,|\, \omega)$ represents a density function. We introduce some additional notations. Let $p = P(\omega = 1)$ be the prior probability of $\omega = 1$. For technical convenience we assume $p \in (0, 1)$. Define

$$\rho = \tfrac{p}{1-p} = \tfrac{P(\omega=1)}{P(\omega=0)} \in (0, +\infty). \tag{9}$$

We will be working with ratios like "$\frac{r_i}{1-r_i}$" and "$\frac{p}{1-p}$" a lot in this section. We will use the following characterization of the optimal aggregator $f^*$ for conditionally independent distributions:

**Lemma 5.1** ([8]). *For conditionally independent distribution $P$, given signals $\boldsymbol{s} = (s_i)_{i=1}^n$, with corresponding reports $\boldsymbol{r} = (r_i)_{i=1}^n$ where $r_i = P(\omega = 1|s_i)$, the posterior probability of $\omega = 1$ is:*

$$f^*(\boldsymbol{r}) = P(\omega = 1 \mid \boldsymbol{r}) = P(\omega = 1 \mid \boldsymbol{s}) = \frac{1}{1+\rho^{n-1} \prod_{i=1}^n \frac{1-r_i}{r_i}}. \tag{10}$$

*(Define $f^*(\boldsymbol{r}) = 0$ if $\rho^{n-1} \prod_{i=1}^n \frac{1-r_i}{r_i} = +\infty$.)*

Lemma 5.1 implies that one way to learn $f^*$ is to simply learn the value of $\rho$. If we can learn $\rho$ with accuracy $\frac{\sqrt{\varepsilon}}{n}$, then we can learn $\rho^{n-1}$ with accuracy $\sqrt{\varepsilon}$ and obtain an $\hat{f}$ that is $\varepsilon$-close to $f^*$ for every possible input $\boldsymbol{r} \in [0, 1]^n$. However, by standard concentration inequalities, learning $\rho$ with accuracy $\frac{\sqrt{\varepsilon}}{n}$ requires $\tilde{O}(\frac{n^2}{\rho \varepsilon})$ samples, which is larger than the $\tilde{O}(\frac{1}{\varepsilon^2})$ bound we will prove. The key is that we do not actually need $\hat{f}(\boldsymbol{r})$ to be close to $f^*(\boldsymbol{r})$ for *every* $\boldsymbol{r} \in [0, 1]_+^n$; we only need the *expectation* $\mathbb{E}\big[|\hat{f}(\boldsymbol{r}) - f^*(\boldsymbol{r})|^2\big] \le \varepsilon$. This allows us to prove a smaller sample complexity bound, using a pseudo-dimension argument.

The main result of this section is that the sample complexity of forecast aggregation with respect to all conditionally independent distributions is between $\Omega(\frac{1}{\varepsilon} \log \frac{1}{\delta})$ and $O(\frac{1}{\varepsilon^2} \log \frac{1}{\varepsilon \delta})$:

**Theorem 5.2.** *Let $\mathcal{P}_{\mathrm{ind}}$ be the set of all conditionally independent distributions over $\boldsymbol{S} \times \Omega$. Suppose $n \ge 2$. The sample complexity of forecast aggregation with respect to $\mathcal{P}_{\mathrm{ind}}$ is*

$$O\big(\tfrac{1}{\varepsilon^2} \log \tfrac{1}{\varepsilon \delta}\big) \ \ge \ T_{\mathcal{P}_{\mathrm{ind}}}(\varepsilon, \delta) \ \ge \ \Omega\big(\tfrac{1}{\varepsilon} \log \tfrac{1}{\delta}\big). \tag{11}$$

We provide the main ideas of the proof of Theorem 5.2 here. The upper bound $O(\frac{1}{\varepsilon^2} \log \frac{1}{\varepsilon \delta})$ is a corollary of our theorem for multi-outcome events (Theorem C.1), so we only give a sketch here. We note that, according to Lemma 5.1, the optimal aggregator has the form $f^*(\boldsymbol{r}) = \frac{1}{1+\rho^{n-1} \prod_{i=1}^n \frac{1-r_i}{r_i}}$.
We consider the class of aggregators $\mathcal{F} = \big\{ f^\theta : f^\theta(\boldsymbol{r}) = \frac{1}{1+\theta^{n-1} \prod_{i=1}^n \frac{1-r_i}{r_i}} \big\}$ parameterized by $\theta \in (0, +\infty)$. The class of loss functions $\mathcal{G} = \big\{ g^\theta : g^\theta(\boldsymbol{r}, \omega) = |f^\theta(\boldsymbol{r}) - \omega|^2 \big\}$ associated with $\mathcal{F}$ has *pseudo-dimension* $\mathrm{Pdim}(\mathcal{G}) = O(1)$. By the known result (e.g., [1]) that the pseudo-dimension gives a sample complexity upper bound on the uniform convergence of a class of functions, we conclude that the empirically optimal aggregator in $\mathcal{F}$ must be $O(\varepsilon)$-optimal on the true distribution (with probability at least $1 - \delta$), given $O\big(\frac{1}{\varepsilon^2}(\mathrm{Pdim}(\mathcal{G}) \log \frac{1}{\varepsilon} + \log \frac{1}{\delta})\big) = O(\frac{1}{\varepsilon^2} \log \frac{1}{\varepsilon \delta})$ samples.

We prove the $\Omega(\frac{1}{\varepsilon} \log \frac{1}{\delta})$ lower bound by a reduction from the distinguishing distributions problem (introduced in Section 3). We construct two conditionally independent distributions $P^1, P^2$ over

the space $\Omega \times \boldsymbol{S}$ that differ by $d_{\mathrm{H}}^2(P^1, P^2) = O(\varepsilon)$ in squared Hellinger distance. Specifically, $P^1$ has prior $P^1(\omega = 1) = 0.5 - O(\frac{1}{n}) + O(\frac{\sqrt{\varepsilon}}{n})$ and $P^2$ has prior $P^2(\omega = 1) = 0.5 - O(\frac{1}{n}) - O(\frac{\sqrt{\varepsilon}}{n})$; the conditional distributions of each signal, $P^1(s_i \,|\, \omega)$ and $P^2(s_i \,|\, \omega)$, differ by $O(\frac{\varepsilon}{n})$ in squared Hellinger distance; taking the product of $n$ signals, $P^1(\boldsymbol{s} \,|\, \omega)$ and $P^2(\boldsymbol{s} \,|\, \omega)$ differ by $O(\varepsilon)$. The distinguishing distributions problem asks: given $T$ samples from either $P^1$ or $P^2$, tell which distribution the samples come from. We show that, if we can solve the forecast aggregation problem, namely, $\varepsilon$-approximate $f^*(\boldsymbol{r}) = \frac{1}{1 + \rho^{n-1} \prod_{i=1}^n \frac{1-r_i}{r_i}}$, then we can estimate $\rho$ with accuracy $O(\frac{\sqrt{\varepsilon}}{n})$, and hence distinguish $P^1$ and $P^2$. But distinguishing $P^1$ and $P^2$ requires $\Omega(\frac{1}{d_{\mathrm{H}}^2(P^1, P^2)} \log \frac{1}{\delta}) = \Omega(\frac{1}{\varepsilon} \log \frac{1}{\delta})$ samples. This gives the lower bound we claimed. See details in Appendix F.1.

## 5.2 Strongly and Weakly Informative Experts

While the sample complexity of forecast aggregation for general conditionally independent distributions is $O(\frac{1}{\varepsilon^2} \log \frac{1}{\varepsilon\delta})$, under further assumptions this bound can be improved. In particular, we find two special yet natural families of conditionally independent distributions that admit $O(\frac{1}{\varepsilon} \log \frac{1}{\delta})$ sample complexity. In these two cases, the experts are either "very informative" or "very non-informative". Roughly speaking, an expert is "very informative" if the conditional distributions of the expert's signal under event $\omega = 0$ and $\omega = 1$ are significantly different, so the expert's prediction $r_i$ is away from the prior $p$. An expert is "very non-informative" if the opposite is true. Intuitively, an expert being informative should help aggregation and hence reduce the sample complexity. Interestingly though, we show that even if the experts are non-informative the sample complexity of forecast aggregation can also be reduced. See details in Appendix B.

# 6 Extension: Multi-Outcome Events

Our main results regarding the sample complexity of forecast aggregation for binary events (Theorems 4.1 and 5.2) can be generalized to multi-outcome events with $|\Omega| > 2$. We prove that: for general distributions, the sample complexity is $\tilde{\Omega}\big(\frac{m^{n-2}}{\varepsilon}\big) = \Omega\big(\frac{m^{n-2} + \log(1/\delta)}{\varepsilon}\big) \leq T_{\mathcal{P}}(\varepsilon, \delta) \leq O\big(\frac{|\Omega| m^n + \log(1/\delta)}{\varepsilon^2}\big) = \tilde{O}\big(\frac{|\Omega| m^n}{\varepsilon^2}\big)$; for conditionally independent distributions, the sample complexity is $\tilde{\Omega}\big(\frac{1}{\varepsilon}\big) = \Omega\big(\frac{1}{\varepsilon} \log \frac{1}{\delta}\big) \leq T_{\mathcal{P}_{\mathrm{ind}}}(\varepsilon, \delta) \leq O\big(\frac{|\Omega| \log |\Omega|}{\varepsilon^2} \log \frac{1}{\varepsilon} + \frac{1}{\varepsilon^2} \log \frac{1}{\delta}\big) = \tilde{O}\big(\frac{|\Omega|}{\varepsilon^2}\big)$. See Appendix C for details.

# 7 Conclusion and Discussion

In this work, we showed an exponential gap between the sample complexity of forecast aggregation for general distributions, $\tilde{\Omega}(\frac{m^{n-2}}{\varepsilon})$, and conditionally independent distributions, $\tilde{O}(\frac{1}{\varepsilon^2})$. This gap is due to the need of estimating the conditional correlation between experts in the general case, which is not needed in the conditional independence case. Notably, the bound $\tilde{O}(\frac{1}{\varepsilon^2})$ for conditionally independent distributions does not depend on the number of experts.

We discuss the dependency of the sample complexity on $\varepsilon$ and other directions for future works:

**The dependency on $\varepsilon$**  An open question left by our work is the dependency of the sample complexity on the parameter $\varepsilon$. We conjecture that the tight dependency should be $\frac{1}{\varepsilon}$ (so our lower bounds are tight). This is supported by the following evidence:

**Theorem 7.1.** *For the case of $|\Omega| = 2$ and for general distributions, if the distribution $P$ has a minimum joint probability $\min_{(\boldsymbol{s}, \omega) \in \boldsymbol{S} \times \Omega} P(\boldsymbol{s}, \omega) > \frac{c}{m^n}$ for some $c > 0$, then the sample complexity of forecast aggregation is at most $O\big(\frac{m^n}{c\varepsilon}(n \log m + \log \frac{1}{\delta})\big) = \tilde{O}\big(\frac{nm^n}{c\varepsilon}\big)$.*[5]

In particular, this theorem can be applied to distributions that are close to uniform, where $P(\boldsymbol{s}, \omega) \approx \frac{1}{|\boldsymbol{S} \times \Omega|} = \frac{1}{2m^n}$, giving a bound of $\tilde{O}(\frac{nm^n}{\varepsilon})$. Notably, the set of distributions we constructed in the proof of the $\tilde{\Omega}(\frac{m^{n-2}}{\varepsilon})$ lower bound in Theorem 4.1 are also close to uniform. This means that

---

[5]This bound has a better dependency on $\varepsilon$ but worse on $n$ than the $O\big(\frac{m^n}{\varepsilon^2}\big)$ bound in Theorem 4.1.

close-to-uniform distributions have a tight sample complexity bound of the form $\Theta(\frac{f(n,m)}{\varepsilon})$, not $\Theta(\frac{f(n,m)}{\varepsilon^2})$. Moreover, since close-to-uniform distributions are the "most difficult" distributions to learn in the distribution learning problem, it is likely that they are also the most difficult distributions for the forecast aggregation problem, and therefore the tight sample complexity of forecast aggregation should be determined by the sample complexity for those distributions, which is $\Theta(\frac{f(n,m)}{\varepsilon})$.

**Other future directions**

- *The middle ground between fully correlated experts and conditionally independent experts:* An example is the *partial evidence model* in [4]. Applying [4]'s results, one can show that the sample complexity of forecast aggregation in the partial evidence model is at most $\tilde{O}(\frac{n^2}{\varepsilon^2})$.[6] Giving a lower bound for the partial evidence model and exploring other intermediate models is open.

- *Weaker benchmark:* Since the Bayesian aggregator needs exponentially many samples to approximate, can we find a weaker yet meaningful benchmark with a small sample complexity?

- *Samples vs experts:* In reality, obtaining samples of experts' historical forecasts can be difficult, while recruiting experts is easy. Can we achieve better aggregation by recruiting more experts instead of collecting more samples? How many experts do we need?

- *Eliciting more information:* Previous works on information elicitation and aggregation have noticed that better aggregation can be achieved by eliciting more information than agents' own predictions, for example, also eliciting each agent's prediction about other agents' predictions (e.g., [41, 47, 31, 14]). One can ask whether and how eliciting more information can help to reduce the sample complexity of information aggregation.

- *Continuous distributions:* In our model the random variable $\omega$ to be predicted is discrete. One can study a setting where $\omega$ is a continuous random variable and the experts report, e.g., the means of their posterior beliefs about $\omega$. The results for continuous random variables might be very different from the results in this work.

- *Other loss functions:* We focused on the squared loss $\mathbb{E}\big[|f(\boldsymbol{r}) - \omega|^2\big]$ due to its popularity in machine learning problems and its useful property that the difference between the squared losses of any aggregator and the optimal aggregator is equal to their expected squared difference (Lemma 2.1). Alternatively, one can consider other loss functions like the logarithmic loss $\mathbb{E}[\omega \log(f(\boldsymbol{r})) + (1-\omega) \log(1 - f(\boldsymbol{r}))]$ and the absolute loss $\mathbb{E}[|f(\boldsymbol{r}) - \omega|]$. There might be some technical challenges in the analysis of sample complexity for those loss functions, though: e.g., the logarithmic loss can be unbounded [4, 40] and the absolute loss does not enjoy a property like Lemma 2.1.

## Acknowledgments and Disclosure of Funding

We would like to thank Yannai Gonczarowski, Ariel Procaccia, Milind Tambe, David Parkes, Bo Waggoner, Rafael Frongillo, Grant Schoenebeck, Yuqing Kong, and anonymous reviewers for their helpful comments. This research is based upon work supported in part by the National Science Foundation under grants no. IIS-2007887 and no. IIS-2147187.

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

# A    Additional Preliminaries

## A.1    Properties of Hellinger Distance

We review some useful properties of the Hellinger distance. First, the Hellinger distance gives upper bounds on the total variation distance:

**Fact A.1** (e.g., Lemma 2.3 in [46])**.**

- $d_{\mathrm{TV}}(D_1, D_2) \leq \sqrt{2} d_{\mathrm{H}}(D_1, D_2)$.

- $1 - d_{\mathrm{TV}}^2(D_1, D_2) \geq \left(1 - d_{\mathrm{H}}^2(D_1, D_2)\right)^2$. *(This inequality implies the first one.)*

Second, we will use the following lemma to upper bound the squared Hellinger distance between two distributions that are close to each other:

**Lemma A.2.** *Let $D_1$ and $D_2$ be two distributions on $\mathcal{X}$ satisfying $1 - \varepsilon \leq \frac{D_2(x)}{D_1(x)} \leq 1 + \varepsilon$, $\forall x \in \mathcal{X}$. Then, $d_{\mathrm{H}}^2(D_1, D_2) \leq \frac{1}{2}\varepsilon^2$.*

*Proof.* By definition,

$$d_{\mathrm{H}}^2(D_1, D_2) = \frac{1}{2} \sum_{x \in \mathcal{X}} \left(\sqrt{D_1(x)} - \sqrt{D_2(x)}\right)^2 = \frac{1}{2} \sum_{x \in \mathcal{X}} D_1(x) \left(1 - \sqrt{\tfrac{D_2(x)}{D_1(x)}}\right)^2.$$

If $1 - \varepsilon \leq \frac{D_2(x)}{D_1(x)} < 1$, then we have $\left(1 - \sqrt{\tfrac{D_2(x)}{D_1(x)}}\right)^2 \leq \left(1 - \sqrt{1 - \varepsilon}\right)^2 \leq \left(1 - (1 - \varepsilon)\right)^2 = \varepsilon^2$. If $1 + \varepsilon \geq \frac{D_2(x)}{D_1(x)} > 1$, then we have $\left(1 - \sqrt{\tfrac{D_2(x)}{D_1(x)}}\right)^2 \leq \left(\sqrt{1 + \varepsilon} - 1\right)^2 \leq \left((1 + \varepsilon) - 1\right)^2 = \varepsilon^2$. These two cases together imply

$$d_{\mathrm{H}}^2(D_1, D_2) \leq \frac{1}{2} \sum_{x \in \mathcal{X}} D_1(x) \cdot \varepsilon^2 = \frac{1}{2}\varepsilon^2. \qquad \square$$

Third, we will use a property of the Hellinger distance between distributions defined on a product space. Suppose $D_1$ and $D_2$ are two distributions over a product space $\mathcal{X} \times \mathcal{Y}$. They can be decomposed into the marginal distribution of $x \in \mathcal{X}$ and the conditional distribution of $y \in \mathcal{Y}$ given $x$, namely, $D_1(x, y) = D_{1,x}(x) \cdot D_{1,y|x}(y|x)$ and $D_2(x, y) = D_{2,x}(x) \cdot D_{2,y|x}(y|x)$. Then, the squared Hellinger distance between $D_1$ and $D_2$ satisfies the following:

**Lemma A.3.** $d_{\mathrm{H}}^2(D_1, D_2) \leq d_{\mathrm{H}}^2(D_{1,x}, D_{2,x}) + \max_{x \in \mathcal{X}} d_{\mathrm{H}}^2(D_{1,y|x}, D_{2,y|x})$. *In particular, if $x$ and $y$ are independent, then $d_{\mathrm{H}}^2(D_1, D_2) \leq d_{\mathrm{H}}^2(D_{1,x}, D_{2,x}) + d_{\mathrm{H}}^2(D_{1,y}, D_{2,y})$.*

*Proof.* By definition,

$$
\begin{aligned}
d_{\mathrm{H}}^2(D_1, D_2) &= 1 - \sum_{x \in \mathcal{X}} \sum_{y \in \mathcal{Y}} \sqrt{D_1(x, y) D_2(x, y)} \\
&= 1 - \sum_{x \in \mathcal{X}} \sqrt{D_{1,x}(x) D_{2,x}(x)} \sum_{y \in \mathcal{Y}} \sqrt{D_{1,y|x}(y|x) D_{2,y|x}(y|x)} \\
&= 1 - \sum_{x \in \mathcal{X}} \sqrt{D_{1,x}(x) D_{2,x}(x)} + \sum_{x \in \mathcal{X}} \sqrt{D_{1,x}(x) D_{2,x}(x)} \left(1 - \sum_{y \in \mathcal{Y}} \sqrt{D_{1,y|x}(y|x) D_{2,y|x}(y|x)}\right) \\
&= d_{\mathrm{H}}^2(D_{1,x}, D_{2,x}) + \sum_{x \in \mathcal{X}} \sqrt{D_{1,x}(x) D_{2,x}(x)} \cdot d_{\mathrm{H}}^2(D_{1,y|x}, D_{2,y|x}) \\
&\leq d_{\mathrm{H}}^2(D_{1,x}, D_{2,x}) + \sum_{x \in \mathcal{X}} \sqrt{D_{1,x}(x) D_{2,x}(x)} \cdot \max_{x \in \mathcal{X}} d_{\mathrm{H}}^2(D_{1,y|x}, D_{2,y|x}) \\
&\leq d_{\mathrm{H}}^2(D_{1,x}, D_{2,x}) + 1 \cdot \max_{x \in \mathcal{X}} d_{\mathrm{H}}^2(D_{1,y|x}, D_{2,y|x}),
\end{aligned}
$$

where the last inequality is because $\sum_{x \in \mathcal{X}} \sqrt{D_{1,x}(x) D_{2,x}(x)} = 1 - d_{\mathrm{H}}^2(D_{1,x}, D_{2,x}) \leq 1$. $\qquad \square$

Finally, let $D^{\otimes T}$ denote the distribution of $T$ i.i.d. samples from $D$, namely, the product of $T$ independent $D$'s. We have the following lemma that relates $d_{\mathrm{H}}^2(D_1^{\otimes T}, D_2^{\otimes T})$ with $d_{\mathrm{H}}^2(D_1, D_2)$:

**Lemma A.4** (e.g., [32]). $d_{\mathrm{H}}^2(D_1^{\otimes T}, D_2^{\otimes T}) = 1 - \left(1 - d_{\mathrm{H}}^2(D_1, D_2)\right)^T \leq T \cdot d_{\mathrm{H}}^2(D_1, D_2)$.

## A.2 Distinguishing Distributions

Let $D_1, D_2$ be two distributions over a discrete space $\mathcal{X}$. A distribution $D_i$ is chosen uniformly at random from $\{D_1, D_2\}$. Then, we are given $T$ samples from $D_i$ and want to guess whether the distribution is $D_1$ or $D_2$. It is known that at least $T = \Omega\left(\frac{1}{d_{\mathrm{H}}^2(D_1, D_2)} \log \frac{1}{\delta}\right)$ samples are needed to guess correctly with probability at least $1 - \delta$, no matter how we guess. Formally:

**Lemma A.5** (e.g., [32]). *Let $j \in \{1, 2\}$ be the index of the distribution we guess based on the samples. The probability of making a mistake when distinguishing $D_1$ and $D_2$ using $T$ samples, namely $\Pr[j \neq i] = \frac{1}{2}\Pr[j \neq i \,|\, i = 1] + \frac{1}{2}\Pr[j \neq i \,|\, i = 2]$, is at least:*

- $\Pr[j \neq i] \geq \frac{1}{2} - \sqrt{\frac{T}{2}} d_{\mathrm{H}}(D_1, D_2)$.

- $\Pr[j \neq i] \geq \frac{1}{4}\left(1 - d_{\mathrm{H}}^2(D_1, D_2)\right)^{2T} \geq \frac{1}{4}e^{-4T d_{\mathrm{H}}^2(D_1, D_2)}$, *assuming $d_{\mathrm{H}}^2(D_1, D_2) \leq \frac{1}{2}$.*

*The second item implies that, in order to achieve $\Pr[j \neq i] \leq \delta$, we must have $T \geq \frac{1}{4 d_{\mathrm{H}}^2(D_1, D_2)} \log \frac{1}{4\delta}$.*

We provide a proof of this lemma for completeness:

*Proof.* Let $D_1^{\otimes T}$ and $D_2^{\otimes T}$ denote the distributions of $T$ i.i.d. samples from $D_1$ and $D_2$, respectively. The draw of $T$ samples from $D_1$ or $D_2$ is equivalent to the draw of one sample from $D_1^{\otimes T}$ or $D_2^{\otimes T}$. Given one sample from $D_1^{\otimes T}$ or $D_2^{\otimes T}$, the probability of making a mistake when guessing the distribution is at least:

$$
\begin{aligned}
\Pr[j \neq i] &= \frac{1}{2}\Big( \Pr[j = 2 \mid i = 1] + \Pr[j = 1 \mid i = 2] \Big) \\
&= \frac{1}{2}\Big( 1 - \Pr[j = 1 \mid i = 1] + \Pr[j = 1 \mid i = 2] \Big) \\
&= \frac{1}{2} - \frac{1}{2}\Big( \Pr[j = 1 \mid i = 1] - \Pr[j = 1 \mid i = 2] \Big) \\
&= \frac{1}{2} - \frac{1}{2}\Big( \mathbb{E}_{D_1^{\otimes T}}[\mathbb{1}\{j = 1\}] - \mathbb{E}_{D_2^{\otimes T}}[\mathbb{1}\{j = 1\}] \Big) \\
\text{by Fact 3.2} \;\geq\; & \frac{1}{2} - \frac{1}{2} d_{\mathrm{TV}}(D_1^{\otimes T}, D_2^{\otimes T}).
\end{aligned}
\tag{12}
$$

We then upper bound $d_{\mathrm{TV}}(D_1^{\otimes T}, D_2^{\otimes T})$ in two ways, which will prove the two items of the lemma, respectively. First, according to first item of Fact A.1, we have

$$
d_{\mathrm{TV}}(D_1^{\otimes T}, D_2^{\otimes T}) \leq \sqrt{2} d_{\mathrm{H}}(D_1^{\otimes T}, D_2^{\otimes T}).
$$

By Lemma A.4,

$$
d_{\mathrm{H}}(D_1^{\otimes T}, D_2^{\otimes T}) \leq \sqrt{T} d_{\mathrm{H}}(D_1, D_2).
$$

The above two inequalities give $d_{\mathrm{TV}}(D_1^{\otimes T}, D_2^{\otimes T}) \leq \sqrt{2T} d_{\mathrm{H}}(D_1, D_2)$. This proves the first item of the lemma.

Second, according to the second item of Fact A.1 and Lemma A.4, we have

$$
1 - d_{\mathrm{TV}}^2(D_1^{\otimes T}, D_2^{\otimes T}) \geq \left(1 - d_{\mathrm{H}}^2(D_1^{\otimes T}, D_2^{\otimes T})\right)^2 = \left(1 - d_{\mathrm{H}}^2(D_1, D_2)\right)^{2T}
$$

Since $1 - d_{\mathrm{TV}}^2(D_1^{\otimes T}, D_2^{\otimes T}) = \left(1 + d_{\mathrm{TV}}(D_1^{\otimes T}, D_2^{\otimes T})\right)\left(1 - d_{\mathrm{TV}}(D_1^{\otimes T}, D_2^{\otimes T})\right) \leq 2\left(1 - d_{\mathrm{TV}}(D_1^{\otimes T}, D_2^{\otimes T})\right)$, we have

$$
1 - d_{\mathrm{TV}}(D_1^{\otimes T}, D_2^{\otimes T}) \geq \frac{1}{2}\left(1 - d_{\mathrm{H}}^2(D_1, D_2)\right)^{2T}.
$$

Plugging into (12), we get

$$\Pr[j \neq i] \geq \frac{1}{4}\left(1 - d_{\mathrm{H}}^2(D_1, D_2)\right)^{2T}.$$

When $d_{\mathrm{H}}^2(D_1, D_2) < \frac{1}{2}$, we use the inequality $1 - x \geq e^{-2x}$ for $0 < x < \frac{1}{2}$ to conclude that

$$\Pr[j \neq i] \geq \frac{1}{4}\left(e^{-2d_{\mathrm{H}}^2(D_1, D_2)}\right)^{2T} = \frac{1}{4}e^{-4T d_{\mathrm{H}}^2(D_1, D_2)}. \qquad \square$$

## B    Special Cases: Strongly and Weakly Informative Experts

In this section we investigate two special families of conditionally independent distributions that admit $O(\frac{1}{\varepsilon}\log\frac{1}{\delta})$ sample complexity, which is smaller than the $O(\frac{1}{\varepsilon^2}\log\frac{1}{\varepsilon\delta})$ bound for general conditionally independent distributions. In these two cases, the (signals of) experts are either "very informative" or "very non-informative".

**Definition B.1.** *Let $\gamma \in [0, \infty]$ be a parameter. For an expert $i$, we say its signal $s_i \in \mathcal{S}_i$ is*

- *$\gamma$-strongly informative if either $\frac{P(s_i|\omega=1)}{P(s_i|\omega=0)} \geq 1 + \gamma$ or $\frac{P(s_i|\omega=1)}{P(s_i|\omega=0)} \leq \frac{1}{1+\gamma}$ holds.*

- *$\gamma$-weakly informative if $\frac{1}{1+\gamma} \leq \frac{P(s_i|\omega=1)}{P(s_i|\omega=0)} \leq 1 + \gamma$.*

*An expert $i$ is $\gamma$-strongly (or $\gamma$-weakly) informative if all of its signals in $\mathcal{S}_i$ are $\gamma$-strongly (or $\gamma$-weakly) informative.*[7]

A signal $s_i$ being $\gamma$-strongly (or $\gamma$-weakly) informative implies that its corresponding report $r_i$ will be "$\gamma$-away from" (or "$\gamma$-close to") the prior $p = P(\omega = 1)$, in the "$\frac{r_i}{1-r_i}$ and $\frac{p}{1-p}$" ratio form. Specifically, if $s_i$ is $\gamma$-strongly informative, then from Equation (1) we have

$$\frac{r_i}{1-r_i} = \frac{P(s_i|\omega=1)}{P(s_i|\omega=0)}\frac{p}{1-p} \quad \geq (1+\gamma)\rho \ \text{ or } \ \leq \frac{1}{1+\gamma}\rho. \tag{13}$$

As $\gamma$ gets larger, a $\gamma$-strongly informative signal (expert) is *more* informative for predicting whether $\omega = 1$ or $0$. This would make aggregation easier. If $s_i$ is $\gamma$-weakly informative, then:

$$\frac{1}{1+\gamma}\rho \ \leq \ \frac{r_i}{1-r_i} \ \leq \ (1+\gamma)\rho. \tag{14}$$

As $\gamma$ gets smaller, a $\gamma$-weakly informative signal (expert) is *less* informative for predicting $\omega$, but in this case their report $r_i$ will be close to the prior $p$, which allows us to estimate the $\rho^{n-1}$ term in the optimal aggregator $f^*(\boldsymbol{r}) = \frac{1}{1+\rho^{n-1}\prod_{i=1}^n \frac{1-r_i}{r_i}}$ better. Those are some intuitions why both strongly and weakly informative signals can reduce the sample complexity of forecast aggregation.

Formally, for $\gamma$-strongly informative experts with not-too-small $\gamma$, we have the following result:

**Theorem B.2.** *If $n \geq 32 \log \frac{2}{\varepsilon}$ and all experts are $\gamma$-strongly informative with $\frac{\gamma}{1+\gamma} \geq 8\sqrt{\frac{2}{n}\log\frac{2}{\varepsilon}}$, then the sample complexity of forecast aggregation is $\leq O\left(\frac{1}{\varepsilon n(\frac{\gamma}{1+\gamma})^2}\log\frac{1}{\delta} + \frac{1}{\varepsilon}\log\frac{1}{\delta}\right) = O(\frac{1}{\varepsilon}\log\frac{1}{\delta})$.*

We remark that the conditions of the theorem, $n \geq 32 \log \frac{2}{\varepsilon}$ and $\frac{\gamma}{1+\gamma} \geq 8\sqrt{\frac{2}{n}\log\frac{2}{\varepsilon}}$, are easier to be satisfied when the number of experts $n$ increases, if the informativeness parameter $\gamma$ of each expert is a constant or does not decrease with $n$ (which we believe is a reasonable assumption given that experts are independent of each other). Also, if $\gamma$ is fixed or increasing, then as $n$ increases the sample complexity decreases.

The proof of Theorem B.2 is in Appendix G.1. Roughly speaking, we divide each expert's signal set into two sets, $\mathcal{S}_i^1$ and $\mathcal{S}_i^0$: signals that are more likely to occur under $\omega = 1$ (i.e., $\frac{P(s_i|\omega=1)}{P(s_i|\omega=0)} \geq 1 + \gamma$) and under $\omega = 0$ (i.e., $\frac{P(s_i|\omega=1)}{P(s_i|\omega=0)} \leq \frac{1}{1+\gamma}$). If the realized $\omega$ is 1, then one may expect to see $\Omega\left(\frac{\gamma}{1+\gamma}n\right)$ more $\mathcal{S}_i^1$ signals than the $\mathcal{S}_i^0$ signals from the $n$ experts, because the probabilities of these two types of signals differ by $\Omega\left(\frac{\gamma}{1+\gamma}\right)$ for each expert. If $\omega$ is 0 then one may expect to see more $\mathcal{S}_i^0$ signals than the $\mathcal{S}_i^1$ signals. So, by checking which type of signals are more we can tell whether $\omega = 0$ or

---

[7]We note that an expert can be neither $\gamma$-strongly informative nor $\gamma$-weakly informative for any $\gamma$.

1. To tell whether a signal belongs to $\mathcal{S}_i^1$ or $\mathcal{S}_i^0$, we compare the corresponding report $\frac{r_i}{1-r_i}$ with $\rho$ (namely, $\frac{r_i}{1-r_i} \geq (1+\gamma)\rho \iff s_i \in \mathcal{S}_i^1$ and $\frac{r_i}{1-r_i} \leq \frac{1}{1+\gamma}\rho \iff s_i \in \mathcal{S}_i^0$), where $\rho$ is estimated with accuracy $\frac{\gamma}{1+\gamma}$ using $O(\frac{1}{\rho n(\frac{\gamma}{1+\gamma})^2} \log \frac{1}{\delta})$ samples. Dealing with the case of $\rho < \varepsilon$ separately, we obtain the bound $O(\frac{1}{\varepsilon n(\frac{\gamma}{1+\gamma})^2} \log \frac{1}{\delta} + \frac{1}{\varepsilon} \log \frac{1}{\delta})$.

For $\gamma$-weakly informative experts with small $\gamma$, we have:

**Theorem B.3.** *If all experts are $\gamma$-weakly informative with $\gamma \leq 1$, then the sample complexity of forecast aggregation is $\leq \min\left\{O(\frac{\gamma n}{\varepsilon} \log \frac{1}{\delta}), O(\frac{1}{\varepsilon^2} \log \frac{1}{\varepsilon\delta})\right\}$, which is $O(\frac{1}{\varepsilon} \log \frac{1}{\delta})$ if $\gamma = O(\frac{1}{n})$.*

The $O(\frac{1}{\varepsilon^2} \log \frac{1}{\varepsilon\delta})$ term in the sample complexity follows from the result for general conditionally independent distributions (Theorem 5.2). The $O(\frac{\gamma n}{\varepsilon} \log \frac{1}{\delta})$ term is proved in Appendix G.2. We give the rough idea here using $\gamma = \frac{1}{n}$ as an example. The proof relies on the observation that $\mathbb{E}\left[\prod_{i=1}^n \frac{r_i}{1-r_i} \mid \omega = 0\right] = \rho^n$. Since experts are weakly informative, each of their reports $\frac{r_i}{1-r_i}$ is around $\rho$ in the range $[\frac{1}{1+\gamma}\rho, (1+\gamma)\rho] \subseteq [\frac{1}{1+\frac{1}{n}}\rho, (1+\frac{1}{n})\rho]$. Taking the product, we have $\prod_{i=1}^n \frac{r_i}{1-r_i} \in [\frac{\rho^n}{e}, e\rho^n]$, which is in a bounded range. This allows us to use Chernoff bound to argue that the $\rho^n$ (or $\rho^{n-1}$) term in the optimal aggregator $f^*(\boldsymbol{r}) = \frac{1}{1+\rho^{n-1}\prod_{i=1}^n \frac{1-r_i}{r_i}}$ can be estimated, with $O(\sqrt{\varepsilon})$ accuracy, using only $O(\frac{e}{(\sqrt{\varepsilon})^2} \log \frac{1}{\delta}) = O(\frac{1}{\varepsilon} \log \frac{1}{\delta})$ samples of $\prod_{i=1}^n \frac{r_i}{1-r_i}$. The aggregator using the estimate $\hat{\rho}^{n-1}$, $\hat{f}(\boldsymbol{r}) = \frac{1}{1+\hat{\rho}^{n-1}\prod_{i=1}^n \frac{1-r_i}{r_i}}$, turns out to be $O(\varepsilon)$-optimal.

## C  Extension: Multi-Outcome Events

In this section we generalize our model from binary events to multi-outcome events. The event space now becomes $\Omega = \{1, 2, \ldots, |\Omega|\}$ with $|\Omega| > 2$. The joint distribution of event $\omega \in \Omega$ and experts' signals $\boldsymbol{s} = (s_i)_{i=1}^n \in \boldsymbol{\mathcal{S}}$ is still denoted by $P$, which belongs to some class of distributions $\mathcal{P}$. The size of each expert's signal space is still assumed to be $|\mathcal{S}_i| = m < +\infty$ for general distributions and can be infinite for conditionally independent distributions (where $s_1, \ldots, s_n$ are independent conditioned on $\omega$). After observing signal $s_i$, expert $i$ reports its posterior belief of the event given $s_i$, which is $\boldsymbol{r}_i = (r_{ij})_{j \in \Omega}$ where $r_{ij} = P(\omega = j \mid s_i)$. An aggregator now is a vector-valued function $\boldsymbol{f} = (f_j)_{j \in \Omega}$ that maps the joint report $\boldsymbol{r} = (\boldsymbol{r}_i)_{i=1}^n = (r_{ij})_{ij}$ to a probability distribution $\boldsymbol{f}(\boldsymbol{r})$ over $\Omega$, where $f_j(\boldsymbol{r})$ is the aggregated predicted probability for $\omega = j$. We assume $f_j(\boldsymbol{r}) \geq 0$ and $\sum_{j \in \Omega} f_j(\boldsymbol{r}) = 1$. The definition of the (expected) loss of an aggregator $\boldsymbol{f}$ becomes:

$$L_P(\boldsymbol{f}) = \mathbb{E}_P\left[ \sum_{j \in \Omega} \left| f_j(\boldsymbol{r}) - \mathbb{1}[\omega = j] \right|^2 \right]. \tag{15}$$

It is easy to see that the optimal aggregator $\boldsymbol{f}^* = \operatorname{argmin}_{\boldsymbol{f}} L_P(\boldsymbol{f})$ in the multi-outcome case is still the Bayesian aggregator (this is a generalization of Lemma 2.1):

$$\boldsymbol{f}^* = (f_j^*)_{j \in \Omega}, \quad f_j^*(\boldsymbol{r}) = P(\omega = j \mid \boldsymbol{r}) \tag{16}$$

and the difference between the losses of $\boldsymbol{f}$ and $\boldsymbol{f}^*$ satisfies

$$L_P(\boldsymbol{f}) - L_P(\boldsymbol{f}^*) = \mathbb{E}_P\left[ \sum_{j \in \Omega} \left| f_j(\boldsymbol{r}) - f_j^*(\boldsymbol{r}) \right|^2 \right]. \tag{17}$$

So, an $\varepsilon$-optimal aggregator $\boldsymbol{f}$ is an aggregator that satisfies $\mathbb{E}_P\left[ \sum_{j \in \Omega} |f_j(\boldsymbol{r}) - f_j^*(\boldsymbol{r})|^2 \right] \leq \varepsilon$.

Using $T = T_{\mathcal{P}}(\varepsilon, \delta)$ samples $\{(\boldsymbol{r}^{(t)}, \omega^{(t)})\}_{t=1}^T$ from $P$, the principal wants to find an $\varepsilon$-optimal aggregator $\hat{\boldsymbol{f}}$ with probability at least $1 - \delta$, for any distribution $P$ in a class $\mathcal{P}$. We give lower bounds and upper bounds on the sample complexity:

**Theorem C.1.** *The sample complexity of forecast aggregation for multi-outcome events is:*

- *For the class $\mathcal{P}$ of general distributions: $\Omega\left(\frac{m^{n-2}+\log(1/\delta)}{\varepsilon}\right) \leq T_{\mathcal{P}}(\varepsilon, \delta) \leq O\left(\frac{|\Omega|m^n+\log(1/\delta)}{\varepsilon^2}\right)$.*

- *For the class $\mathcal{P}_{\mathrm{ind}}$ of conditionally independent distributions: $\Omega\left(\frac{1}{\varepsilon} \log \frac{1}{\delta}\right) \leq T_{\mathcal{P}_{\mathrm{ind}}}(\varepsilon, \delta) \leq O\left(\frac{|\Omega| \log |\Omega|}{\varepsilon^2} \log \frac{1}{\varepsilon} + \frac{1}{\varepsilon^2} \log \frac{1}{\delta}\right)$.*

## C.1 Proof of Theorem C.1

Before proving the theorem, we note that the loss of any aggregator $\boldsymbol{f}$ satisfying $f_j(\boldsymbol{r}) \geq 0$ and $\sum_{j \in \Omega} f_j(\boldsymbol{r}) = 1$ is bounded by $[0, 2]$:

$$0 \leq \sum_{j \in \Omega} \left| f_j(\boldsymbol{r}) - \mathbb{1}[\omega = j] \right|^2 \leq \sum_{j \in \Omega} \left| f_j(\boldsymbol{r}) - \mathbb{1}[\omega = j] \right| \leq 1 + \sum_{j \in \Omega} f_j(\boldsymbol{r}) = 2. \tag{18}$$

Similarly,

$$0 \leq \sum_{j \in \Omega} \left| f_j(\boldsymbol{r}) - f_j^*(\boldsymbol{r}) \right|^2 \leq 2. \tag{19}$$

### C.1.1 Lower Bounds

The lower bounds directly follow from the lower bounds for the binary case (Theorem 4.1 and Theorem 5.2) because the binary case is a special case of the multi-outcome case. Specifically, we can regard any binary event distribution $P$ as a multi-outcome distribution that puts probability only on outcomes $\{1, 2\} \subseteq \Omega$. If we can learn an $\varepsilon$-optimal aggregator $\hat{\boldsymbol{f}}$ for the multi-outcome case: $\mathbb{E}\big[\sum_{j \in \Omega} |\hat{f}_j(\boldsymbol{r}) - f_j^*(\boldsymbol{r})|^2\big] \leq \varepsilon$, then this aggregator satisfies

$$\mathbb{E}\Big[ \sum_{j \in \{1,2\}} \left| \hat{f}_j(\boldsymbol{r}) - f_j^*(\boldsymbol{r}) \right|^2 \Big] \leq \varepsilon$$

$$\Longleftrightarrow \quad \mathbb{E}\Big[ \left| \hat{f}_1(\boldsymbol{r}) - f_1^*(\boldsymbol{r}) \right|^2 + \left| (1 - \hat{f}_1(\boldsymbol{r})) - (1 - f_1^*(\boldsymbol{r})) \right|^2 \Big] = \mathbb{E}\Big[ 2\left| \hat{f}_1(\boldsymbol{r}) - f_1^*(\boldsymbol{r}) \right|^2 \Big] \leq \varepsilon$$

$$\Longleftrightarrow \quad \mathbb{E}\Big[ \left| \hat{f}_1(\boldsymbol{r}) - f_1^*(\boldsymbol{r}) \right|^2 \Big] \leq \frac{\varepsilon}{2}.$$

So, the aggregator $\hat{f}_1(\cdot)$ is an $\frac{\varepsilon}{2}$-optimal aggregator for the binary case.

### C.1.2 Upper Bounds

**General distributions: the $O\big(\frac{|\Omega|m^n + \log(1/\delta)}{\varepsilon^2}\big)$ upper bound.** The proof for general distributions in the multi-outcome case is the same as the proof for general distributions in the binary case (Section 4.1), except for three differences: (1) $P$ (regarded as a distribution over reports $\boldsymbol{r}$ and event $\omega$) now is a discrete distribution with support size at most $|\Omega|m^n$; (2) the loss $|f(\boldsymbol{r}) - \omega|^2 \in [0, 1]$ in the binary case is replaced by the loss $\sum_{j \in \Omega} |f_j(\boldsymbol{r}) - \mathbb{1}[\omega = j]|^2 \in [0, 2]$ in the multi-outcome case; (3) the $\varepsilon$ in the binary case is replaced by $\frac{\varepsilon}{2}$ because the loss is now upper bounded by 2. Thus, the bound $O\big(\frac{m^n + \log(1/\delta)}{\varepsilon^2}\big)$ in the binary case becomes $O\big(\frac{|\Omega|m^n + \log(1/\delta)}{(\frac{\varepsilon}{2})^2}\big) = O\big(\frac{|\Omega|m^n + \log(1/\delta)}{\varepsilon^2}\big)$ in the multi-outcome case.

**Conditionally independent distributions: the $O\big(\frac{|\Omega| \log |\Omega|}{\varepsilon^2} \log \frac{1}{\varepsilon} + \frac{1}{\varepsilon^2} \log \frac{1}{\delta}\big)$ upper bound.** The sample complexity upper bound for conditionally independent distributions is proved by a *pseudo-dimension* argument. We first show in Lemma C.2 that the optimal aggregator $\boldsymbol{f}^*$ belongs to some parametric family of aggregators. Then, we upper bound the pseudo-dimension of the loss functions associated with this parametric family of aggregators, which will give the desired sample complexity upper bound.

**Lemma C.2.** *For multi-outcome conditionally independent distribution $P$, given signals $\boldsymbol{s} = (s_i)_{i=1}^n$, with corresponding reports $\boldsymbol{r} = (\boldsymbol{r}_i)_{i=1}^n = (r_{ij})_{ij}$ where $r_{ij} = P(\omega = j \,|\, s_i)$, the posterior probability of $\omega = j$ satisfies:*

$$f_j^*(\boldsymbol{r}) = P(\omega = j \mid \boldsymbol{r}) = P(\omega = j \mid \boldsymbol{s}) = \frac{\frac{1}{P(\omega=j)^{n-1}} \prod_{i=1}^n r_{ij}}{\sum_{k \in \Omega} \frac{1}{P(\omega=k)^{n-1}} \prod_{i=1}^n r_{ik}}.$$

*Proof.* By Bayes' rule and the fact that $s_1, \ldots, s_n$ are independent conditioning on $\omega$,

$$P(\omega = j \mid \boldsymbol{s}) = \frac{P(\omega = j) \prod_{i=1}^n P(s_i \mid \omega = j)}{\sum_{k \in \Omega} P(\omega = k) \prod_{i=1}^n P(s_i \mid \omega = k)}$$

$$= \frac{\frac{1}{P(\omega=j)^{n-1}} \prod_{i=1}^n P(\omega = j) P(s_i \mid \omega = j)}{\sum_{k \in \Omega} \frac{1}{P(\omega=k)^{n-1}} \prod_{i=1}^n P(\omega = k) P(s_i \mid \omega = k)}$$

$$= \frac{\frac{1}{P(\omega=j)^{n-1}} \prod_{i=1}^n \frac{P(\omega=j) P(s_i \mid \omega=j)}{P(s_i)}}{\sum_{k \in \Omega} \frac{1}{P(\omega=k)^{n-1}} \prod_{i=1}^n \frac{P(\omega=k) P(s_i \mid \omega=k)}{P(s_i)}}$$

$$= \frac{\frac{1}{P(\omega=j)^{n-1}} \prod_{i=1}^n r_{ij}}{\sum_{k \in \Omega} \frac{1}{P(\omega=k)^{n-1}} \prod_{i=1}^n r_{ik}}.$$

Since the above expression depends only on $r_i$'s but not on $s_i$'s, we have:

$$f_j^*(\boldsymbol{r}) = P(\omega = j \mid \boldsymbol{r}) = P(\omega = j \mid \boldsymbol{s}) = \frac{\frac{1}{P(\omega=j)^{n-1}} \prod_{i=1}^n r_{ij}}{\sum_{k \in \Omega} \frac{1}{P(\omega=k)^{n-1}} \prod_{i=1}^n r_{ik}}.$$

(For the special binary event case,

$$f^*(\boldsymbol{r}) = P(\omega = 1 \mid \boldsymbol{r}) = \frac{\frac{1}{P(\omega=1)^{n-1}} \prod_{i=1}^n r_i}{\frac{1}{P(\omega=1)^{n-1}} \prod_{i=1}^n r_i + \frac{1}{P(\omega=0)^{n-1}} \prod_{i=1}^n (1 - r_i)} = \frac{1}{1 + \left(\frac{P(\omega=1)}{P(\omega=0)}\right)^{n-1} \prod_{i=1}^n \frac{1-r_i}{r_i}}.$$

This proves Lemma 5.1.) $\square$

According to Lemma C.2, the optimal aggregator $\boldsymbol{f}^* = (f_j)_{j \in \Omega}$ belongs to the following family of aggregators, parameterized by $\boldsymbol{\theta} = (\theta_j)_{j \in \Omega} \in \mathbb{R}_+^{|\Omega|}$:

$$\mathcal{F} = \left\{ \boldsymbol{f^\theta} = (f_j^\theta)_{j \in \Omega} \,\middle|\, f_j^\theta(\boldsymbol{r}) = \frac{\theta_j \prod_{i=1}^n r_{ij}}{\sum_{k \in \Omega} \theta_k \prod_{i=1}^n r_{ik}} \right\} \tag{20}$$

(The optimal aggregator $\boldsymbol{f}^*$ has parameter $\theta_j = \frac{1}{P(\omega=j)^{n-1}}$.) Let $\mathcal{G}$ be the family of loss functions associated with $\mathcal{F}$, which is also parameterized by $\boldsymbol{\theta} \in \mathbb{R}_+^{|\Omega|}$,

$$\mathcal{G} = \left\{ g^\theta \,\middle|\, g^\theta(\boldsymbol{r}, \omega) = \sum_{j \in \Omega} \left| f_j^\theta(\boldsymbol{r}) - \mathbb{1}[\omega = j] \right|^2 \right\}. \tag{21}$$

We recall the definition of pseudo-dimension of a family of functions:

**Definition C.3** (e.g., [1]). *Let $\mathcal{H}$ be a family of functions from input space $\mathcal{X}$ to real numbers $\mathbb{R}$. Let $x^{(1)}, \ldots, x^{(d)} \in \mathcal{X}$ be $d$ inputs. Let $t^{(1)}, \ldots, t^{(d)} \in \mathbb{R}$ be $d$ "thresholds". Let $\boldsymbol{b} = (b^{(1)}, \ldots, b^{(d)}) \in \{-1, +1\}^d$ be a vector of labels. We say $\boldsymbol{b}$ can be generated by $\mathcal{H}$ (on inputs $x^{(1)}, \ldots, x^{(d)}$ with thresholds $t^{(1)}, \ldots, t^{(d)}$) if there exists a function $h_{\boldsymbol{b}} \in \mathcal{H}$ such that $h_{\boldsymbol{b}}(x^{(i)}) > t^{(i)}$ if $b^{(i)} = 1$ and $h_{\boldsymbol{b}}(x^{(i)}) < t^{(i)}$ if $b^{(i)} = -1$ (namely, $\text{sign}(h_{\boldsymbol{b}}(x^{(i)}) - t^{(i)}) = b^{(i)}$) for every $i \in \{1, \ldots, d\}$. The pseudo-dimension of $\mathcal{H}$, $\text{Pdim}(\mathcal{H})$, is the largest integer $d$ for which there exist $d$ inputs and $d$ thresholds such that all the $2^d$ label vectors in $\{-1, +1\}^d$ can be generated by $\mathcal{H}$.*

Pseudo-dimension gives a sample complexity upper bound for the uniform convergence of the empirical means of a family of functions to their true means:

**Theorem C.4** (e.g., [1]). *Let $\mathcal{H}$ be a family of functions from $\mathcal{X}$ to $[0, U]$ with $\text{Pdim}(\mathcal{H}) = d$. For any distribution $\mathcal{D}$ on $\mathcal{X}$, with probability at least $1 - \delta$ over the random draw of*

$$T = O\left( \frac{U^2}{\varepsilon^2} \left( d \cdot \log \frac{U}{\varepsilon} + \log \frac{1}{\delta} \right) \right)$$

*samples $x^{(1)}, \ldots, x^{(T)}$ from $\mathcal{D}$, we have: for every $h \in \mathcal{H}$, $\left| \mathbb{E}_{x \sim \mathcal{D}}[h(x)] - \frac{1}{T} \sum_{t=1}^T h(x^{(t)}) \right| \leq \varepsilon$.*

We now upper bound the pseudo-dimension of $\mathcal{G}$:

**Lemma C.5.** $\mathrm{Pdim}(\mathcal{G}) \leq O\big(|\Omega| \log |\Omega|\big)$.

*Proof.* Suppose $\mathrm{Pdim}(\mathcal{G}) = d$. By definition, there exist $d$ inputs $(\boldsymbol{r}^{(1)}, \omega^{(1)}), \ldots, (\boldsymbol{r}^{(d)}, \omega^{(d)})$ and $d$ thresholds $t^{(1)}, \ldots, t^{(d)} \in \mathbb{R}$ such that all the $2^d$ label vectors $\boldsymbol{b} \in \{-1, +1\}^d$ can be generated by some function $g^{\boldsymbol{\theta}_{\boldsymbol{b}}} \in \mathcal{G}$. We will count how many label vectors can actually be generated by $\mathcal{G}$.

Consider the $\ell$-th input $(\boldsymbol{r}^{(\ell)}, \omega^{(\ell)})$ and threshold $t^{(\ell)}$. To simplify notations we omit superscript $(\ell)$, so we have $(\boldsymbol{r}, \omega)$ and $t$. We write $\boldsymbol{x} = (x_j)_{j \in \Omega}$ where $x_j = \prod_{i=1}^n r_{ij}$. For any function $g^{\boldsymbol{\theta}} \in \mathcal{G}$, we have

$$
\begin{aligned}
g^{\boldsymbol{\theta}}(\boldsymbol{r}, \omega) &= \sum_{j \in \Omega} \left| \frac{\theta_j x_j}{\sum_{k \in \Omega} \theta_k x_k} - \mathbb{1}[\omega = j] \right|^2 \\
&= \sum_{j \in \Omega} \left( \frac{\theta_j x_j}{\sum_{k \in \Omega} \theta_k x_k} \right)^2 - 2 \frac{\theta_\omega x_\omega}{\sum_{k \in \Omega} \theta_k x_k} + 1 \\
&= \frac{1}{\big( \sum_{k \in \Omega} \theta_k x_k \big)^2} \left[ \sum_{j \in \Omega} (\theta_j x_j)^2 - 2\theta_\omega x_\omega \sum_{k \in \Omega} \theta_k x_k + \Big( \sum_{k \in \Omega} \theta_k x_k \Big)^2 \right].
\end{aligned}
$$

By definition, the set of parameters that give input $(\boldsymbol{r}, \omega)$ label "$+1$" is $\big\{ \boldsymbol{\theta} \in \mathbb{R}_+^{|\Omega|} \mid g^{\boldsymbol{\theta}}(\boldsymbol{r}, \omega) > t \big\}$. By the above equation, this set is equal to the set

$$
\left\{ \boldsymbol{\theta} \in \mathbb{R}_+^{|\Omega|} \;\middle|\; \sum_{j \in \Omega} (\theta_j x_j)^2 - 2\theta_\omega x_\omega \sum_{k \in \Omega} \theta_k x_k + \Big( \sum_{k \in \Omega} \theta_k x_k \Big)^2 > t \Big( \sum_{k \in \Omega} \theta_k x_k \Big)^2 \right\}.
$$

We note that the above set is the solution set to the quadratic form inequality $\boldsymbol{\theta}^\top A \boldsymbol{\theta} > 0$ for some matrix $A \in \mathbb{R}^{|\Omega| \times |\Omega|}$. Similarly, the set of parameters that give input $(\boldsymbol{r}, \omega)$ label "$-1$" is the solution set to $\boldsymbol{\theta}^\top A \boldsymbol{\theta} < 0$. These two sets share a boundary: the solution set to $\boldsymbol{\theta}^\top A \boldsymbol{\theta} = 0$, which is a hyper-ellipsoid in $\mathbb{R}^{|\Omega|}$. In other words, the input $(\boldsymbol{r}, \omega)$ and threshold $t$ define a hyper-ellipsoid which divides the parameter space $\mathbb{R}_+^{|\Omega|}$ into two regions such that all the parameters in one region generate the same label for that input.

Enumerating all inputs $(\boldsymbol{r}^{(1)}, \omega^{(1)}), \ldots, (\boldsymbol{r}^{(d)}, \omega^{(d)})$. They define $d$ hyper-ellipsoids in total, dividing the parameter space $\mathbb{R}_+^{|\Omega|}$ into several regions. Within each region, all the parameters generate the same label for each input and hence generate the same label vector. So, the number of label vectors that can be generated by all the parameters in $\mathbb{R}_+^{|\Omega|}$ is upper bounded by the number of regions. The number of regions divided by $d$ hyper-ellipsoids in $\mathbb{R}_+^{|\Omega|}$ is in the order of $O(d^{|\Omega|})$. Hence, to generate all the $2^d$ label vectors we need $O(d^{|\Omega|}) \geq 2^d$. This gives $d \leq O(|\Omega| \log |\Omega|)$. $\qquad \square$

By Theorem C.4 and Lemma C.5, plugging in $d = \mathrm{Pdim}(\mathcal{G}) \leq O(|\Omega| \log |\Omega|)$ and $U = 2$ (because $g^{\boldsymbol{\theta}}(\boldsymbol{r}, \omega)$ is bounded by $[0, 2]$ according to (18)), we obtain the following: with probability at least $1 - \delta$ over the random draw of

$$
T = O\left( \frac{U^2}{\varepsilon^2} \Big( d \cdot \log \frac{U}{\varepsilon} + \log \frac{1}{\delta} \Big) \right) = O\left( \frac{|\Omega| \log |\Omega|}{\varepsilon^2} \log \frac{1}{\varepsilon} + \frac{1}{\varepsilon^2} \log \frac{1}{\delta} \right)
$$

samples $(\boldsymbol{r}^{(1)}, \omega^{(1)}), \ldots, (\boldsymbol{r}^{(T)}, \omega^{(T)})$ from $P$, we have for any $g^{\boldsymbol{\theta}} \in \mathcal{G}$,

$$
\left| \mathbb{E}_P \big[ g^{\boldsymbol{\theta}}(\boldsymbol{r}, \omega) \big] - \frac{1}{T} \sum_{t=1}^T g^{\boldsymbol{\theta}}(\boldsymbol{r}^{(t)}, \omega^{(t)}) \right| \leq \varepsilon.
$$

This is equivalent to for any $\boldsymbol{f}^{\boldsymbol{\theta}} \in \mathcal{F}$,

$$
\left| \mathbb{E}_P \Big[ \sum_{j \in \Omega} \big| f_j^{\boldsymbol{\theta}}(\boldsymbol{r}) - \mathbb{1}[\omega = j] \big|^2 \Big] - \frac{1}{T} \sum_{t=1}^T \sum_{j \in \Omega} \big| f_j^{\boldsymbol{\theta}}(\boldsymbol{r}^{(t)}) - \mathbb{1}[\omega^{(t)} = 1] \big|^2 \right| \leq \varepsilon
$$

$$
\iff \left| L_P(\boldsymbol{f}^{\boldsymbol{\theta}}) - L_{\hat{P}_{\mathrm{emp}}}(\boldsymbol{f}^{\boldsymbol{\theta}}) \right| \leq \varepsilon,
$$

where $\hat{P}_{\mathrm{emp}}$ is the empirical distribution $\mathrm{Uniform}\big\{ (\boldsymbol{r}^{(1)}, \omega^{(1)}), \ldots, (\boldsymbol{r}^{(T)}, \omega^{(T)}) \big\}$. By the same logic as the proof of the upper bound in Theorem 4.1 (Section 4.1), the empirically optimal aggregator $\hat{\boldsymbol{f}}^* = \arg\min_{\boldsymbol{f}^{\boldsymbol{\theta}} \in \mathcal{F}} L_{\hat{P}_{\mathrm{emp}}}(\boldsymbol{f}^{\boldsymbol{\theta}})$ is $2\varepsilon$-optimal.

# D Missing Proofs from Section 2

## D.1 Proof of Lemma 2.1

To prove the first item $f^*(\boldsymbol{r}) = P(\omega = 1 \,|\, \boldsymbol{r})$, we simply note that

$$f^* = \operatorname*{argmin}_{f} \mathbb{E}_{\boldsymbol{r}}\Big[\mathbb{E}\big[|f(\boldsymbol{r}) - \omega|^2 \,|\, \boldsymbol{r}\big]\Big]$$

should minimize $\mathbb{E}\big[|f(\boldsymbol{r}) - \omega|^2 \,|\, \boldsymbol{r}\big]$ for almost every $\boldsymbol{r}$. This gives $f^*(\boldsymbol{r}) = \mathbb{E}[\omega \,|\, \boldsymbol{r}] = P(\omega = 1 \,|\, \boldsymbol{r})$.

To prove the second item $L_P(f) - L_P(f^*) = \mathbb{E}_P\big[|f(\boldsymbol{r}) - f^*(\boldsymbol{r})|^2\big]$, we note that, by the definition of $L_P(\cdot)$ and the fact that $f^*(\boldsymbol{r}) = \mathbb{E}[\omega \,|\, \boldsymbol{r}]$ proven above,

$$
\begin{aligned}
L_P(f) - L_P(f^*) &= \mathbb{E}\big[|f(\boldsymbol{r}) - \omega|^2\big] - \mathbb{E}\big[|f^*(\boldsymbol{r}) - \omega|^2\big] \\
&= \mathbb{E}\big[f(\boldsymbol{r})^2\big] - 2\mathbb{E}\big[f(\boldsymbol{r})\omega\big] - \mathbb{E}\big[f^*(\boldsymbol{r})^2\big] + 2\mathbb{E}\big[f^*(\boldsymbol{r})\omega\big] \\
&= \mathbb{E}\big[f(\boldsymbol{r})^2\big] - 2\mathbb{E}_{\boldsymbol{r}}\big[f(\boldsymbol{r})\mathbb{E}[\omega|\boldsymbol{r}]\big] - \mathbb{E}\big[f^*(\boldsymbol{r})^2\big] + 2\mathbb{E}_{\boldsymbol{r}}\big[f^*(\boldsymbol{r})\mathbb{E}[\omega|\boldsymbol{r}]\big] \\
&= \mathbb{E}\big[f(\boldsymbol{r})^2\big] - 2\mathbb{E}_{\boldsymbol{r}}\big[f(\boldsymbol{r})f^*(\boldsymbol{r})\big] - \mathbb{E}\big[f^*(\boldsymbol{r})^2\big] + 2\mathbb{E}_{\boldsymbol{r}}\big[f^*(\boldsymbol{r})^2\big] \\
&= \mathbb{E}\big[f(\boldsymbol{r})^2 - 2f(\boldsymbol{r})f^*(\boldsymbol{r}) + f^*(\boldsymbol{r})^2\big] \\
&= \mathbb{E}\big[|f(\boldsymbol{r}) - f^*(\boldsymbol{r})|^2\big].
\end{aligned}
$$

# E Missing Proofs from Section 4

## E.1 Proof of Lemma 4.3

Recall that we have a family $\mathcal{D}$ of distributions over $\boldsymbol{\mathcal{S}} = \mathcal{S}_1 \times \cdots \times \mathcal{S}_n$ satisfying the three properties in Definition 4.2 ($B$-uniformly bounded, same marginal across distributions, and distinct marginals across signals). We have constructed from $\mathcal{D}$ the family of distributions $\mathcal{P} = \{P_D : D \in \mathcal{D}\}$ for the forecast aggregation problem as follows:

$$
\begin{cases}
P_D(\omega = 0) = P_D(\omega = 1) = \dfrac{1}{2}, \\[2mm]
P_D(\cdot \,|\, \omega = 0) = \mathrm{Uniform}(\boldsymbol{\mathcal{S}}), \quad \text{namely,} \quad P_D(\boldsymbol{s} \,|\, \omega = 0) = \dfrac{1}{|\boldsymbol{\mathcal{S}}|} = \dfrac{1}{m^n}, \\[2mm]
P_D(\cdot \,|\, \omega = 1) = D, \qquad\qquad \text{namely,} \quad P_D(\boldsymbol{s} \,|\, \omega = 1) = D(\boldsymbol{s}).
\end{cases}
\tag{22}
$$

We want to show that $\varepsilon$-optimal aggregation with respect to $\mathcal{P}$ will imply $(1+B)^2\sqrt{\varepsilon}$ total variation distance learning with respect to $\mathcal{D}$, and hence $T_{\mathcal{P}}(\varepsilon, \delta) \geq T_{\mathcal{D}}^{\mathrm{TV}}\big((1+B)^2\sqrt{\varepsilon}, \delta\big)$.

First, we have the following observations about $\mathcal{D}$ and $\mathcal{P}$:

**Fact E.1.** *For a family of distributions $\mathcal{D}$ that satisfies the three properties in Definition 4.2:*

1. *Given signal $s_i$, expert $i$'s report is $r_i = P_D(\omega = 1 \,|\, s_i) = \dfrac{D(s_i)}{\frac{1}{m} + D(s_i)}$, which is the same for all distributions $D \in \mathcal{D}$.*

2. *For every expert $i$, given different signals $s_i \neq s_i'$, its reports $r_i \neq r_i'$. So, there is a one-to-one mapping between $s_i$ and $r_i$ for every $i \in \{1, \ldots, n\}$ and also a one-to-one mapping between the joint signal $\boldsymbol{s} = (s_1, \ldots, s_n)$ and the joint report $\boldsymbol{r} = (r_1, \ldots, r_n)$.*

3. *For any joint signal $\boldsymbol{s}$, with corresponding joint report $\boldsymbol{r}$, we have $D(\boldsymbol{s}) = \dfrac{f^*(\boldsymbol{r})}{m^n(1 - f^*(\boldsymbol{r}))}$.*

*Proof.* We prove the three items one by one:

1. By definition, the marginal distribution of joint signal, $P_D(\boldsymbol{s})$, is

$$P_D(\boldsymbol{s}) = P_D(\omega = 0)P_D(\boldsymbol{s} \,|\, \omega = 0) + P_D(\omega = 1)P_D(\boldsymbol{s} \,|\, \omega = 1) = \frac{1}{2}\Big(\frac{1}{m^n} + D(\boldsymbol{s})\Big).
\tag{23}$$

Fixing $s_i$, summing over $\boldsymbol{s}_{-i} = (s_1, \ldots, s_{i-1}, s_{i+1}, \ldots, s_n) \in \boldsymbol{S}_{-i}$, we get

$$P_D(s_i) = \frac{1}{2}\Big(\frac{|\boldsymbol{S}_{-i}|}{m^n} + \sum_{\boldsymbol{s}_{-i} \in \boldsymbol{S}_{-i}} D(\boldsymbol{s})\Big) = \frac{1}{2}\Big(\frac{1}{m} + D(s_i)\Big).$$

So, given signal $s_i$, expert $i$ reports

$$r_i = P_D(\omega = 1 \mid s_i) = \frac{P_z(\omega = 1)P_D(s_i \mid \omega = 1)}{P_D(s_i)} = \frac{D(s_i)}{\frac{1}{m} + D(s_i)}, \tag{24}$$

and this is the same for all $D \in \mathcal{D}$ since $D(s_i) = D'(s_i)$ by the "same marginal across distributions" property.

2. Given $s_i \neq s_i'$, by the "distinct marginals across signals" property, we have $D(s_i) \neq D(s_i')$. Since $r_i = \frac{D(s_i)}{\frac{1}{m} + D(s_i)}$ and $\frac{x}{\frac{1}{m} + x}$ is a strictly increasing function of $x$, it follows that $r_i \neq r_i'$.

3. According to item 2, there is a one-to-one mapping between $\boldsymbol{s} = (s_1, \ldots, s_n)$ and $\boldsymbol{r} = (r_1, \ldots, r_n)$; in other words, observing signals $s_1, \ldots, s_n$ is equivalent to observing reports $r_1, \ldots, r_n$. Therefore, by Bayes' rule we have

$$f^*(\boldsymbol{r}) = P_D(\omega = 1 \mid \boldsymbol{r}) = P_D(\omega = 1 \mid \boldsymbol{s}) = \frac{P_D(\omega = 1)P_D(\boldsymbol{s} \mid \omega = 1)}{P_D(\boldsymbol{s})}$$

$$\text{by (22) and (23)} = \frac{D(\boldsymbol{s})}{\frac{1}{m^n} + D(\boldsymbol{s})}. \tag{25}$$

Rearranging, we obtain $D(\boldsymbol{s}) = \frac{f^*(\boldsymbol{r})}{m^n(1 - f^*(\boldsymbol{r}))}$. $\qquad\square$

**Claim E.2.** *If we have an aggregator $\hat{f}$ that is $\varepsilon$-optimal with respect to $P_D$, then we can find a distribution $\hat{D}$ such that $d_{\mathrm{TV}}(\hat{D}, D) \leq (1 + B)^2 \sqrt{\varepsilon}$.*

*Proof.* Because $D$ is $B$-uniformly bounded, from (25) we can verify that $f^*(\boldsymbol{r})$ satisfies $f^*(\boldsymbol{r}) \leq \frac{B}{1+B}$. So, we can assume $\hat{f}(\boldsymbol{r}) \leq \frac{B}{1+B}$ as well (if $\hat{f}(\boldsymbol{r}) > \frac{B}{1+B}$, we can let $\hat{f}(\boldsymbol{r})$ be $\frac{B}{1+B}$; this only reduces the approximation error $\mathbb{E}\big[|\hat{f}(\boldsymbol{r}) - f^*(\boldsymbol{r})|^2\big]$). Define $\hat{D}$ by letting $\hat{D}(\boldsymbol{s}) = \frac{\hat{f}(\boldsymbol{r})}{m^n(1 - \hat{f}(\boldsymbol{r}))}$, $\forall \boldsymbol{s} \in \boldsymbol{S}$, where $\boldsymbol{r}$ is the reports corresponding to $\boldsymbol{s}$ (cf., Fact E.1). Then, $d_{\mathrm{TV}}(\hat{D}, D)$ is

$$d_{\mathrm{TV}}(\hat{D}, D) = \frac{1}{2}\sum_{\boldsymbol{s} \in \boldsymbol{S}} \big|\hat{D}(\boldsymbol{s}) - D(\boldsymbol{s})\big| = \frac{1}{2}\sum_{\boldsymbol{s} \in \boldsymbol{S}} \Big|\frac{\hat{f}(\boldsymbol{r})}{m^n(1 - \hat{f}(\boldsymbol{r}))} - \frac{f^*(\boldsymbol{r})}{m^n(1 - f^*(\boldsymbol{r}))}\Big|$$

$$= \frac{1}{2m^n}\sum_{\boldsymbol{s} \in \boldsymbol{S}} \Big|\frac{\hat{f}(\boldsymbol{r})}{1 - \hat{f}(\boldsymbol{r})} - \frac{f^*(\boldsymbol{r})}{1 - f^*(\boldsymbol{r})}\Big|.$$

Because $\hat{f}(\boldsymbol{r}), f^*(\boldsymbol{r}) \leq \frac{B}{1+B}$ and the function $\frac{x}{1-x}$ has derivative $\frac{1}{(1-x)^2} \leq (1+B)^2$ when $x \leq \frac{B}{1+B}$, we have

$$d_{\mathrm{TV}}(\hat{D}, D) \leq \frac{1}{2m^n}(1 + B)^2 \sum_{\boldsymbol{s} \in \boldsymbol{S}} \big|\hat{f}(\boldsymbol{r}) - f^*(\boldsymbol{r})\big|$$

$$\text{by (23)} \leq (1 + B)^2 \sum_{\boldsymbol{s} \in \boldsymbol{S}} P_D(\boldsymbol{s})\big|\hat{f}(\boldsymbol{r}) - f^*(\boldsymbol{r})\big| = (1 + B)^2 \mathbb{E}_{P_D}\Big[\big|\hat{f}(\boldsymbol{r}) - f^*(\boldsymbol{r})\big|\Big].$$

By Jensen's inequality $(\mathbb{E}[X])^2 \leq \mathbb{E}[X^2]$ and by the assumption that $\hat{f}$ is $\varepsilon$-optimal, we have $\big(\mathbb{E}_{P_D}\big[|\hat{f}(\boldsymbol{r}) - f^*(\boldsymbol{r})|\big]\big)^2 \leq \mathbb{E}_{P_D}\big[|\hat{f}(\boldsymbol{r}) - f^*(\boldsymbol{r})|^2\big] \leq \varepsilon$. Thus,

$$d_{\mathrm{TV}}(\hat{D}, D) \leq (1 + B)^2 \sqrt{\varepsilon},$$

which proves the claim. $\qquad\square$

Now, we present the reduction from learning $\mathcal{D}$ in total variation distance to forecast aggregation for $\mathcal{P}$. We use notations $\boldsymbol{x}^{(1)}, \ldots, \boldsymbol{x}^{(T)} \in \boldsymbol{S}$ to represent the samples from $D$. From $\boldsymbol{x}^{(1)}, \ldots, \boldsymbol{x}^{(T)}$ we construct the samples $(\boldsymbol{r}^{(1)}, \omega^{(1)}), \ldots, (\boldsymbol{r}^{(T)}, \omega^{(T)})$ for the forecast aggregation problem. After obtaining a solution $\hat{f}$ to the latter problem, we convert it into a solution $\hat{D}$ to the former. Details are as follows:

**Input:** $T$ i.i.d. samples $\boldsymbol{x}^{(1)}, \ldots, \boldsymbol{x}^{(T)}$ from an unknown distribution $D \in \mathcal{D}$.

**Reduction:**

1. Draw $T$ samples $\omega^{(1)}, \ldots, \omega^{(T)} \sim \text{Uniform}\{0, 1\}$.
2. For each $t = 1, \ldots, T$, do the following:
   - If $\omega^{(t)} = 0$, draw $\boldsymbol{s}^{(t)} \sim \text{Uniform}(\boldsymbol{\mathcal{S}})$.
   - If $\omega^{(t)} = 1$, let $\boldsymbol{s}^{(t)} = \boldsymbol{x}^{(t)}$.
   - For each $i$, compute $r_i^{(t)} = \frac{D(s_i^{(t)})}{\frac{1}{m} + D(s_i^{(t)})}$. Let $\boldsymbol{r}^{(t)} = (r_1^{(t)}, \ldots, r_n^{(t)})$.
3. Feed $S_T = \left\{ (\boldsymbol{r}^{(1)}, \omega^{(1)}), \ldots, (\boldsymbol{r}^{(T)}, \omega^{(T)}) \right\}$ to the forecast aggregation problem. Obtain solution $\hat{f}$.
4. Convert $\hat{f}$ into $\hat{D}$ according to Claim E.2.

**Output:** $\hat{D}$.

We remark that, in the second step of the reduction, the report $r_i^{(t)}$ can be computed even if $D$ is unknown, because the $D(s_i^{(t)})$ is the same for all $D \in \mathcal{D}$ and hence known.

Using the above reduction, we show that the sample complexity of $\varepsilon$-optimal forecast aggregation for $\mathcal{P}$ cannot be smaller than the sample complexity of learning $\mathcal{D}$ within total variation distance $(1 + B)^2 \sqrt{\varepsilon}$, which will prove Lemma 4.3:

**Proof of Lemma 4.3:** First, we verify that the distribution of samples $S_T$ in the above reduction is exactly the distribution of $T$ samples $\{(\boldsymbol{r}^{(1)}, \omega^{(1)}), \ldots, (\boldsymbol{r}^{(T)}, \omega^{(T)})\}$ from $P_D$. This is because: (1) the distribution of $\omega^{(t)}$ is $\text{Uniform}\{0, 1\}$, as defined in $P_D$; (2) given $\omega^{(t)} = 0$, the distribution of $\boldsymbol{s}^{(t)}$ is $\text{Uniform}(\boldsymbol{\mathcal{S}})$, as defined in $P_D$; (3) given $\omega^{(t)} = 1$, the distribution of $\boldsymbol{s}^{(t)}$ is the same as the distribution of $\boldsymbol{x}^{(t)}$, which is $D$, because the random draws of $\omega^{(t)}$ and $\boldsymbol{x}^{(t)}$ are independent; (4) according to Fact E.1, the report $r_i^{(t)} = \frac{D(s_i^{(t)})}{\frac{1}{m} + D(s_i^{(t)})} = P_D(\omega = 1 \mid s_i^{(t)})$, as desired.

Then, by the definition of sample complexity of forecast aggregation, if we are given $T = T_{\mathcal{P}}(\varepsilon, \delta)$ samples $S_T$ for the forecast aggregation problem, then with probability at least $1 - \delta$ we should be able to find an $\varepsilon$-optimal aggregator $\hat{f}$ with respect to $P_D$. According to Claim E.2, we can convert $\hat{f}$ into a $\hat{D}$ such that

$$d_{\text{TV}}(\hat{D}, D) \leq (1 + B)^2 \sqrt{\varepsilon}.$$

By the definition of sample complexity $T_{\mathcal{D}}^{\text{TV}}(\cdot, \delta)$ of distribution learning, $T$ must be at least

$$T \geq T_{\mathcal{D}}^{\text{TV}}\big((1 + B)^2 \sqrt{\varepsilon}, \delta\big),$$

which proves the lemma.

### E.2 Proof of Proposition 4.4

To prove Proposition 4.4 we will construct a family of distributions $\mathcal{D}$ that satisfies the three properties in Definition 4.2 and requires $T_{\mathcal{D}}^{\text{TV}}(\varepsilon_{\text{TV}}, \delta) = \Omega\big(\frac{m^{n-2} + \log(1/\delta)}{\varepsilon_{\text{TV}}^2}\big)$ samples to learn. For simplicity, we write $\varepsilon = \varepsilon_{\text{TV}}$. For technical convenience, we assume $\varepsilon < \frac{1}{40}, \delta < 0.01$.

#### E.2.1 Part 1: Constructing $\mathcal{D}$

We index the distributions $D_z \in \mathcal{D}$ by $z$; the meaning of $z$ will be defined later. Without loss of generality, we assume $|\mathcal{S}_i| = m$ to be an even integer, and denote $\mathcal{S}_i = \{1, \ldots, m\} =: S$. We will define $D_z$ to be a distribution over the joint signal space $\boldsymbol{\mathcal{S}} = \mathcal{S}_1 \times \cdots \times \mathcal{S}_n = S^n = \{1, \ldots, m\}^n$. In the following we will call a joint signal $\boldsymbol{s} \in S^n$ simply a signal. We write a signal $\boldsymbol{s} \in S^n$ as $\boldsymbol{s} = (\boldsymbol{b}, x, y)$, where $\boldsymbol{b} \in S^{n-2}$ and $x, y \in S$. We sort all the $m^n$ signals in $S^n$ by the lexicographical

order, from $(1, \ldots, 1, 1)$, $(1, \ldots, 1, 2)$, ..., to $(m, \ldots, m, m)$. We number the signals from $1$ to $m^n$, using

$$\text{num}(\boldsymbol{s}) = \text{num}(\boldsymbol{b}, x, y) \in \{1, \ldots, m^n\}.$$

to denote their numbers. We divide the whole signal space $S^n$ into $|S^{n-2}| = m^{n-2}$ "buckets", each of size $m^2$ and denoted by $B_{\boldsymbol{b}}$:

$$B_{\boldsymbol{b}} = \big\{(\boldsymbol{b}, x, y) : x, y \in S\big\}, \quad \boldsymbol{b} \in S^{n-2}.$$

We first define a "base" distribution $D_{\text{base}}$, then construct the distributions $D_z$'s by modifying the probabilities of the base distribution within each bucket $B_{\boldsymbol{b}}$. Let $\gamma = 1 + \frac{1}{m^n}$. The base distribution is defined as follows:

$$D_{\text{base}}(\boldsymbol{s}) = \frac{\gamma^{\text{num}(\boldsymbol{s})}}{W}, \quad W = \sum_{\boldsymbol{s} \in S^n} \gamma^{\text{num}(\boldsymbol{s})} = \sum_{\ell=1}^{m^n} \gamma^{\ell}.$$

Because $1 \leq \gamma^{\ell} \leq \gamma^{m^n} \leq (1 + \frac{1}{m^n})^{m^n} \leq e$, we have

$$m^n \leq W \leq e m^n. \tag{26}$$

We assign a sign $z_{\boldsymbol{b}} \in \{+1, -1\}$ to each bucket $B_{\boldsymbol{b}}$, and let $z$ be a vector of length $m^{n-2}$ that includes the signs of all buckets:

$$z = (z_{\boldsymbol{b}})_{\boldsymbol{b} \in S^{n-2}}, \quad z_{\boldsymbol{b}} \in \{+1, -1\}.$$

We have $2^{m^{n-2}}$ different $z$'s in total, and hence $2^{m^{n-2}}$ different distributions $D_z$'s in $\mathcal{D}$ in total. Let $c = 20$, so that $c\varepsilon \leq 1/2$. For each $z$, we define $D_z$ as follows: within each bucket $B_{\boldsymbol{b}}$, for each element $(\boldsymbol{b}, x, y) \in B_{\boldsymbol{b}}$ let

$$D_z(\boldsymbol{b}, x, y) = \begin{cases} D_{\text{base}}(\boldsymbol{b}, x, y) + \frac{z_{\boldsymbol{b}} c\varepsilon}{W} & \text{if } x \leq \frac{m}{2}, \ y \leq \frac{m}{2} \\ D_{\text{base}}(\boldsymbol{b}, x, y) - \frac{z_{\boldsymbol{b}} c\varepsilon}{W} & \text{if } x \leq \frac{m}{2}, \ y > \frac{m}{2} \\ D_{\text{base}}(\boldsymbol{b}, x, y) - \frac{z_{\boldsymbol{b}} c\varepsilon}{W} & \text{if } x > \frac{m}{2}, \ y \leq \frac{m}{2} \\ D_{\text{base}}(\boldsymbol{b}, x, y) + \frac{z_{\boldsymbol{b}} c\varepsilon}{W} & \text{if } x > \frac{m}{2}, \ y > \frac{m}{2} \end{cases}. \tag{27}$$

**Claim E.3.** *The family of distributions $\mathcal{D} = \{D_z\}_z$ defined above satisfies the three properties in Definition 4.2: $B$-uniformly bounded with $B = e + 1/2$, same marginal across distributions, distinct marginals across signals.*

*Proof.* $B$-**uniformly bounded:** For any $\boldsymbol{s}$, any $D_z$, by definition,

$$D_z(\boldsymbol{s}) \leq D_{\text{base}}(\boldsymbol{s}) + \frac{c\varepsilon}{W} \leq \frac{\gamma^{m^n}}{W} + \frac{c\varepsilon}{W} \overset{(26)}{\leq} \frac{e}{m^n} + \frac{1/2}{m^n}.$$

So, the distribution is $B$-uniformly bounded with $B = e + 1/2$.

**Same marginal across distributions:** Consider each $D_z(s_i)$. We want to show that $D_z(s_i)$ does not depend on $z$, and in fact, $D_z(s_i) = D_{\text{base}}(s_i)$. If $i \in \{1, \ldots, n-2\}$, namely, $s_i$ is a component of the vector $\boldsymbol{b}$, then we have

$$D_z(s_i) = \sum_{\boldsymbol{s}_{-i} \in \boldsymbol{S}_{-i}} D_z(s_i, \boldsymbol{s}_{-i}) = \sum_{\boldsymbol{b} \in S^{n-2} : b_i = s_i} \sum_{x=1}^{m} \sum_{y=1}^{m} D_z(\boldsymbol{b}, x, y).$$

We notice that, fixing any $\boldsymbol{b}$, the numbers of $\frac{+z_{\boldsymbol{b}} c\varepsilon}{W}$ and $\frac{-z_{\boldsymbol{b}} c\varepsilon}{W}$ in the summation $\sum_{x=1}^{m} \sum_{y=1}^{m} D_z(\boldsymbol{b}, x, y)$ are the same. So, they cancel out, and we obtain

$$D_z(s_i) = \sum_{\boldsymbol{b} \in S^{n-2} : b_i = s_i} \sum_{x=1}^{m} \sum_{y=1}^{m} D_{\text{base}}(\boldsymbol{b}, x, y) = D_{\text{base}}(s_i).$$

If $i = n - 1$, namely $s_i = x$, then we have:

$$D_z(s_i) = \sum_{\boldsymbol{s}_{-i} \in \boldsymbol{S}_{-i}} D_z(s_i, \boldsymbol{s}_{-i}) = \sum_{\boldsymbol{b} \in S^{n-2}} \sum_{y=1}^{m} D_z(\boldsymbol{b}, x, y).$$

Fixing any $\boldsymbol{b}$, the numbers of $\frac{+z_{\boldsymbol{b}}c\varepsilon}{W}$ and $\frac{-z_{\boldsymbol{b}}c\varepsilon}{W}$ in the summation $\sum_{y=1}^m D_z(\boldsymbol{b}, x, y)$ are the same. So, they cancel out, and we obtain

$$D_z(s_i) = \sum_{\boldsymbol{b} \in S^{n-2}} \sum_{y=1}^m D_{\text{base}}(\boldsymbol{b}, x, y) = D_{\text{base}}(s_i).$$

Finally, if $i = n$, namely $s_i = y$, then by a similar argument as above we have

$$D_z(s_i) = \sum_{\boldsymbol{b} \in S^{n-2}} \sum_{x=1}^m D_{\text{base}}(\boldsymbol{b}, x, y) = D_{\text{base}}(s_i).$$

**Distinct marginals across signals:** By the "same marginal across distributions" property above we have $D_z(s_i) = D_{\text{base}}(s_i)$. So, to prove "distinct marginals across signals" we only need to prove $D_{\text{base}}(s_i) \neq D_{\text{base}}(s_i')$ for $s_i \neq s_i'$. Without loss of generality assume $s_i < s_i'$. By the definition $D_{\text{base}}(\boldsymbol{s}) = \frac{\gamma^{\text{num}(\boldsymbol{s})}}{W}$ and the fact that $\text{num}(s_i, \boldsymbol{s}_{-i}) < \text{num}(s_i', \boldsymbol{s}_{-i})$ for any $\boldsymbol{s}_{-i} \in \boldsymbol{\mathcal{S}}_{-i}$, we have

$$D_{\text{base}}(s_i) = \sum_{\boldsymbol{s}_{-i} \in \boldsymbol{\mathcal{S}}_{-i}} D_{\text{base}}(s_i, \boldsymbol{s}_{-i}) < \sum_{\boldsymbol{s}_{-i} \in \boldsymbol{\mathcal{S}}_{-i}} D_{\text{base}}(s_i', \boldsymbol{s}_{-i}) = D_{\text{base}}(s_i'),$$

so $D_{\text{base}}(s_i) \neq D_{\text{base}}(s_i')$. $\qquad\square$

### E.2.2  Part 2: Sample Complexity Lower Bound of Learning $\mathcal{D}$

**Overview**  We want to prove the proposition that the sample complexity of learning the family of distributions $D = \{D_z\}_z$ defined above is at least $T_{\mathcal{D}}^{\text{TV}}(\varepsilon, \delta) = \Omega\big(\frac{m^{n-2} + \log(1/\delta)}{\varepsilon^2}\big)$. This proof is analogous to a textbook proof of Proposition 3.3 (the sample complexity for learning *all* distributions), which uses reductions from the *distinguishing distributions* problem. Roughly speaking, if one can learn the unknown distribution $D_z$ well then one must be able to guess most of the components of the sign vector $z = (z_{\boldsymbol{b}})_{\boldsymbol{b} \in S^{n-2}}$ correctly, meaning that one can distinguish whether the distribution $D_{z_{\boldsymbol{b}}}$ on bucket $B_{\boldsymbol{b}}$ is $D_{z_{\boldsymbol{b}}=+1}$ or $D_{z_{\boldsymbol{b}}=-1}$. However, since $D_{z_{\boldsymbol{b}}=+1}$ and $D_{z_{\boldsymbol{b}}=-1}$ are "$O(\varepsilon)$-close" to each other, distinguishing them requires $\Omega(\frac{1}{\varepsilon^2})$ samples. In average, there are only $O(\frac{T}{m^{n-2}})$ samples falling into a bucket (because there are $m^{n-2}$ buckets in total and the distribution $D_z$ is close to uniform). We thus need $O(\frac{T}{m^{n-2}}) = \Omega(\frac{1}{\varepsilon^2})$, which gives $T = \Omega(\frac{m^{n-2}}{\varepsilon^2})$. Ignoring logarithmic terms, this proves the proposition.

**Formal argument**  First, we note that if we can learn $D_z$ very well, then we can guess the vector $z$ correctly for a large fraction of its $m^{n-2}$ components. Formally, suppose we obtain from $T$ samples a distribution $\hat{D}$ such that $d_{\text{TV}}(\hat{D}, D_z) \leq \varepsilon$. We find the distribution $D_w$ in $\mathcal{D}$, $w = (w_{\boldsymbol{b}})_{\boldsymbol{b} \in S^{n-2}} \in \{+1, -1\}^{m^{n-2}}$, that is closest to $\hat{D}$ in total variation distance. By definition, we have $d_{\text{TV}}(D_w, \hat{D}) \leq d_{\text{TV}}(D_z, \hat{D}) \leq \varepsilon$. Hence, by triangle inequality,

$$d_{\text{TV}}(D_w, D_z) \leq d_{\text{TV}}(D_w, \hat{D}) + d_{\text{TV}}(\hat{D}, D_z) \leq 2\varepsilon.$$

Let

$$\# = \big|\{\boldsymbol{b} \in S^{n-2} \mid w_{\boldsymbol{b}} \neq z_{\boldsymbol{b}}\}\big|$$

be the number of different components of $w$ and $z$. We claim that:

**Claim E.4.** $d_{\text{TV}}(D_w, D_z) \leq 2\varepsilon$ implies $\# \leq \frac{2W}{cm^2}$.

*Proof.* Whenever we have a different component $w_{\boldsymbol{b}} \neq z_{\boldsymbol{b}}$, this different component contributes the following to the total variation distance between $D_w$ and $D_z$:

$$\frac{1}{2} \sum_{(\boldsymbol{b}, x, y) \in B_{\boldsymbol{b}}} \big|D_w(\boldsymbol{b}, x, y) - D_z(\boldsymbol{b}, x, y)\big| = \frac{1}{2} \sum_{(\boldsymbol{b}, x, y) \in B_{\boldsymbol{b}}} \frac{2c\varepsilon}{W} = \frac{m^2 c\varepsilon}{W}. \tag{28}$$

So, the number of different components of $w$ and $z$ is at most $\frac{d_{\text{TV}}(D_w, D_z)}{\frac{m^2 c\varepsilon}{W}} \leq \frac{2W}{cm^2}$. $\qquad\square$

We first show the $\Omega(\frac{\log(1/\delta)}{\varepsilon^2})$ part in the sample complexity lower bound, and then show the $\Omega(\frac{m^{n-2}}{\varepsilon^2})$ part. Together, they give the lower bound $\max\{\Omega(\frac{\log(1/\delta)}{\varepsilon^2}), \Omega(\frac{m^{n-2}}{\varepsilon^2})\} = \Omega(\frac{m^{n-2}+\log(1/\delta)}{\varepsilon^2})$. Consider the distribution $D_{+1} \in \mathcal{D}$ whose index is the all "+1" vector and the distribution $D_{-1} \in \mathcal{D}$ whose index is the all "−1" vector. According to (28), the total variation distance between $D_{+1}$ and $D_{-1}$ is $d_{\mathrm{TV}}(D_{+1}, D_{-1}) = m^{n-2} \cdot \frac{m^2 c\varepsilon}{W} = \frac{m^n c\varepsilon}{W}$ because $+\mathbf{1}$ and $-\mathbf{1}$ have $m^{n-2}$ different components. Since $W \leq em^n$ (26), we have $d_{\mathrm{TV}}(D_{+1}, D_{-1}) \geq \frac{c\varepsilon}{e} > 2\varepsilon$. Consider the distinguishing distributions problem (defined in Section 3) where we want to distinguish $D_{+1}$ and $D_{-1}$. If we can learn from samples a distribution $\hat{D}$ that is $\varepsilon$-close in total variation distance to the unknown distribution $D_{+1}$ or $D_{-1}$, then we can perfectly tell whether the unknown distribution is $D_{+1}$ or $D_{-1}$ because the two distributions are more than $2\varepsilon$-away from each other in total variation distance. Lemma A.5 implies that, to distinguish $D_{+1}$ and $D_{-1}$ with probability $1 - \delta$, the number of samples must be at least $\Omega(\frac{\log(1/\delta)}{d_{\mathrm{H}}^2(D_{+1}, D_{-1})}) = \Omega(\frac{\log(1/\delta)}{\varepsilon^2})$. This proves the $\Omega(\frac{\log(1/\delta)}{\varepsilon^2})$ part.

We then prove the $\Omega(\frac{m^{n-2}}{\varepsilon^2})$ part. Suppose we first draw the vector $z$ from $\{+1, -1\}^{m^{n-2}}$ uniformly at random, then draw $T$ samples from $D_z$. We obtain the $D_w$ as above. Let's consider the expected number of different components of $w$ and $z$ in this two-step random draw procedure:

$$\mathbb{E}_{z,\,T\text{ samples}}[\#] = \mathbb{E}\Big[ \sum_{\boldsymbol{b} \in S^{n-2}} \mathbb{1}\{z_{\boldsymbol{b}} \neq w_{\boldsymbol{b}}\} \Big] = \sum_{\boldsymbol{b} \in S^{n-2}} \mathbb{E}\big[ \mathbb{1}\{z_{\boldsymbol{b}} \neq w_{\boldsymbol{b}}\} \big]. \qquad (29)$$

We consider each component $\mathbb{E}\big[\mathbb{1}\{z_{\boldsymbol{b}} \neq w_{\boldsymbol{b}}\}\big]$ in the above summation. Suppose that, within the $T$ samples drawn from $D_z$, $T_{\boldsymbol{b}}$ of them fall into the bucket $B_{\boldsymbol{b}}$. So, $T_{\boldsymbol{b}}$ follows the $\mathrm{Binomial}(T, D(B_{\boldsymbol{b}}))$ distribution with

$$D(B_{\boldsymbol{b}}) = \sum_{(\boldsymbol{b},x,y) \in B_{\boldsymbol{b}}} D_z(\boldsymbol{b}, x, y) = \frac{1}{W} \sum_{(\boldsymbol{b},x,y) \in B_{\boldsymbol{b}}} \gamma^{\mathrm{num}(\boldsymbol{b},x,y)}.$$

(Notice that the $+\frac{z_{\boldsymbol{b}} c\varepsilon}{W}$ and $-\frac{z_{\boldsymbol{b}} c\varepsilon}{W}$ cancel out in the summation and hence $D(B_{\boldsymbol{b}})$ does not depend on $z$.) Let $D_{z_{\boldsymbol{b}}}$ denote the "$B_{\boldsymbol{b}}$-part" of distribution $D_z$, namely, $D_z$ conditioning on $B_{\boldsymbol{b}}$:

$$D_{z_{\boldsymbol{b}}}(\boldsymbol{s}) = \frac{D_z(\boldsymbol{s})}{D(B_{\boldsymbol{b}})}, \quad \forall \boldsymbol{s} \in B_{\boldsymbol{b}}.$$

We think of the random draw of the vector $z$ and the $T$ samples as follows: first, we draw $T_{\boldsymbol{b}}$ from $\mathrm{Binomial}(T, D(B_{\boldsymbol{b}}))$; second, we draw $z_{\boldsymbol{b}} \in \{+1, -1\}$ uniformly at random; third, we draw $T_{\boldsymbol{b}}$ samples from the conditional distribution $D_{z_{\boldsymbol{b}}}$; forth, we draw the remaining vector $z_{-\boldsymbol{b}}$ and the remaining $T - T_{\boldsymbol{b}}$ samples (which are samples outside of $B_{\boldsymbol{b}}$). Only writing the first two steps explicitly, we have

$$\mathbb{E}\big[\mathbb{1}\{z_{\boldsymbol{b}} \neq w_{\boldsymbol{b}}\}\big] = \mathbb{E}_{T_{\boldsymbol{b}}}\bigg[ \mathbb{E}_{z_{\boldsymbol{b}}}\Big[ \mathbb{E}\big[\mathbb{1}\{z_{\boldsymbol{b}} \neq w_{\boldsymbol{b}}\} \mid z_{\boldsymbol{b}}, T_{\boldsymbol{b}}\big] \Big] \bigg]$$

$$= \mathbb{E}_{T_{\boldsymbol{b}}}\bigg[ \frac{1}{2}\mathbb{E}\big[\mathbb{1}\{z_{\boldsymbol{b}} \neq w_{\boldsymbol{b}}\} \mid z_{\boldsymbol{b}} = +1, T_{\boldsymbol{b}}\big] + \frac{1}{2}\mathbb{E}\big[\mathbb{1}\{z_{\boldsymbol{b}} \neq w_{\boldsymbol{b}}\} \mid z_{\boldsymbol{b}} = -1, T_{\boldsymbol{b}}\big] \bigg]$$

$$= \mathbb{E}_{T_{\boldsymbol{b}}}\bigg[ \frac{1}{2}\Pr\big[z_{\boldsymbol{b}} \neq w_{\boldsymbol{b}} \mid z_{\boldsymbol{b}} = +1, T_{\boldsymbol{b}}\big] + \frac{1}{2}\Pr\big[z_{\boldsymbol{b}} \neq w_{\boldsymbol{b}} \mid z_{\boldsymbol{b}} = -1, T_{\boldsymbol{b}}\big] \bigg]. \quad (30)$$

**Claim E.5.** *For any $T_{\boldsymbol{b}}$, $\frac{1}{2}\Pr\big[z_{\boldsymbol{b}} \neq w_{\boldsymbol{b}} \mid z_{\boldsymbol{b}} = +1, T_{\boldsymbol{b}}\big] + \frac{1}{2}\Pr\big[z_{\boldsymbol{b}} \neq w_{\boldsymbol{b}} \mid z_{\boldsymbol{b}} = -1, T_{\boldsymbol{b}}\big] \geq \frac{1}{2} - 2c\varepsilon\sqrt{T_{\boldsymbol{b}}}$.*

*Proof.* We notice that $\frac{1}{2}\Pr\big[z_{\boldsymbol{b}} \neq w_{\boldsymbol{b}} \mid z_{\boldsymbol{b}} = +1, T_{\boldsymbol{b}}\big] + \frac{1}{2}\Pr\big[z_{\boldsymbol{b}} \neq w_{\boldsymbol{b}} \mid z_{\boldsymbol{b}} = -1, T_{\boldsymbol{b}}\big]$ is the probability that we make a mistake when guessing the sign $z_{\boldsymbol{b}}$ using $w_{\boldsymbol{b}}$, if (1) $z_{\boldsymbol{b}}$ is chosen from $\{-1, +1\}$ uniformly at random; (2) we are given $T_{\boldsymbol{b}}$ samples from $D_{z_{\boldsymbol{b}}}$; (3) we then draw the remaining vector $z_{-\boldsymbol{b}}$ and the remaining samples; (4) finally, we use the $D_w$ computed from all samples to get $w_{\boldsymbol{b}}$. The steps (3) and (4) define a randomized function that maps the $T_{\boldsymbol{b}}$ samples of $D_{z_{\boldsymbol{b}}}$ to $w_{\boldsymbol{b}} \in \{0, 1\}$, and therefore, according to the first item of Lemma A.5, we have

$$\frac{1}{2}\Pr\big[z_{\boldsymbol{b}} \neq w_{\boldsymbol{b}} \mid z_{\boldsymbol{b}} = +1, T_{\boldsymbol{b}}\big] + \frac{1}{2}\Pr\big[z_{\boldsymbol{b}} \neq w_{\boldsymbol{b}} \mid z_{\boldsymbol{b}} = -1, T_{\boldsymbol{b}}\big] \geq \frac{1}{2} - \sqrt{\frac{T_{\boldsymbol{b}}}{2}}\, d_{\mathrm{H}}(D_{z_{\boldsymbol{b}}=+1}, D_{z_{\boldsymbol{b}}=-1}).$$
$$(31)$$

Then we consider $d_{\mathrm{H}}(D_{z_{\boldsymbol{b}}=+1}, D_{z_{\boldsymbol{b}}=-1})$. We use Lemma A.2 to do so. For any $\boldsymbol{s} \in B_{\boldsymbol{b}}$, we have, on the one hand,

$$\frac{D_{z_{\boldsymbol{b}}=+1}(\boldsymbol{s})}{D_{z_{\boldsymbol{b}}=-1}(\boldsymbol{s})} = \frac{\gamma^{\mathrm{num}(\boldsymbol{s})} \pm c\varepsilon}{\gamma^{\mathrm{num}(\boldsymbol{s})} \mp c\varepsilon} \geq \frac{\gamma^{\mathrm{num}(\boldsymbol{s})} - c\varepsilon}{\gamma^{\mathrm{num}(\boldsymbol{s})} + c\varepsilon} \geq 1 - 2c\varepsilon,$$

because $\frac{a-b}{a+b} = 1 - \frac{2b}{a+b} \geq 1 - 2b$ for $a = \gamma^{\mathrm{num}(\boldsymbol{s})} \geq 1$. On the other hand,

$$\frac{D_{z_{\boldsymbol{b}}=+1}(\boldsymbol{s})}{D_{z_{\boldsymbol{b}}=-1}(\boldsymbol{s})} \leq \frac{\gamma^{\mathrm{num}(\boldsymbol{s})} + c\varepsilon}{\gamma^{\mathrm{num}(\boldsymbol{s})} - c\varepsilon} \leq 1 + 4c\varepsilon,$$

because $\frac{a+b}{a-b} = 1 + \frac{2b}{a-b} \leq 1 + 4b$ for $a = \gamma^{\mathrm{num}(\boldsymbol{s})} \geq 1$ and $b = c\varepsilon \leq \frac{1}{2}$. Therefore, by Lemma A.2 we have

$$d_{\mathrm{H}}^2(D_{z_{\boldsymbol{b}}=+1}, D_{z_{\boldsymbol{b}}=-1}) \leq \frac{1}{2}(4c\varepsilon)^2 = 8c^2\varepsilon^2. \tag{32}$$

Combining (31) and (32) proves our claim. $\qquad\square$

By (30) and Claim E.5, we have $\mathbb{E}\big[\mathbb{1}\{z_{\boldsymbol{b}} \neq w_{\boldsymbol{b}}\}\big] \geq \underset{T_{\boldsymbol{b}}}{\mathbb{E}}\big[\frac{1}{2} - 2c\varepsilon\sqrt{T_{\boldsymbol{b}}}\big]$. Summing over $\boldsymbol{b} \in S^{n-2}$, (29) becomes

$$\sum_{\boldsymbol{b} \in S^{n-2}} \mathbb{E}\big[\mathbb{1}\{z_{\boldsymbol{b}} \neq w_{\boldsymbol{b}}\}\big] \geq \sum_{\boldsymbol{b} \in S^{n-2}} \underset{T_{\boldsymbol{b}}}{\mathbb{E}}\Big[\frac{1}{2} - 2c\varepsilon\sqrt{T_{\boldsymbol{b}}}\Big] = \frac{m^{n-2}}{2} - 2c\varepsilon \sum_{\boldsymbol{b} \in S^{n-2}} \mathbb{E}[\sqrt{T_{\boldsymbol{b}}}]$$

$$\text{(by Jensen's inequality } \mathbb{E}[\sqrt{X}] \leq \sqrt{\mathbb{E}[X]}) \; \geq \; \frac{m^{n-2}}{2} - 2c\varepsilon \sum_{\boldsymbol{b} \in S^{n-2}} \sqrt{\mathbb{E}[T_{\boldsymbol{b}}]}.$$

Because $\mathbb{E}[T_{\boldsymbol{b}}] = T \cdot D(B_{\boldsymbol{b}}) = \frac{T}{W} \sum_{(\boldsymbol{b},x,y) \in B_{\boldsymbol{b}}} \gamma^{\mathrm{num}(\boldsymbol{b},x,y)} \leq \frac{T}{W} m^2 \gamma^{m^n}$, we have

$$\sum_{\boldsymbol{b} \in S^{n-2}} \mathbb{E}\big[\mathbb{1}\{z_{\boldsymbol{b}} \neq w_{\boldsymbol{b}}\}\big] \geq \frac{m^{n-2}}{2} - 2c\varepsilon \sum_{\boldsymbol{b} \in S^{n-2}} \sqrt{\frac{T}{W} m^2 \gamma^{m^n}} = \frac{m^{n-2}}{2} - 2c\varepsilon m^{n-2}\sqrt{\frac{T}{W} m^2 \gamma^{m^n}}. \tag{33}$$

Now, let's consider the probability with which we can obtain $\hat{D}$ such that $d_{\mathrm{TV}}(\hat{D}, D_z) \leq \varepsilon$. We will show that this probability is at most $0.99 < 1 - \delta$ if $T$ is less than $10^{-5} \cdot \frac{m^{n-2}}{\varepsilon^2}$. Recall from Claim E.4 that $d_{\mathrm{TV}}(\hat{D}, D_z) \leq \varepsilon$ implies $\# = \sum_{\boldsymbol{b} \in S^{n-2}} \mathbb{1}\{z_{\boldsymbol{b}} \neq w_{\boldsymbol{b}}\} \leq \frac{2W}{cm^2}$. So, the probability is at most

$$\Pr\Big[ \sum_{\boldsymbol{b} \in S^{n-2}} \mathbb{1}\{z_{\boldsymbol{b}} \neq w_{\boldsymbol{b}}\} \leq \frac{2W}{cm^2} \Big] = \Pr\Big[ \sum_{\boldsymbol{b} \in S^{n-2}} \mathbb{1}\{z_{\boldsymbol{b}} = w_{\boldsymbol{b}}\} \geq m^{n-2} - \frac{2W}{cm^2} \Big]$$

$$\text{(by Markov's inequality)} \; \leq \; \frac{\mathbb{E}\big[\sum_{\boldsymbol{b} \in S^{n-2}} \mathbb{1}\{z_{\boldsymbol{b}} = w_{\boldsymbol{b}}\}\big]}{m^{n-2} - \frac{2W}{cm^2}}$$

$$\text{(by (33))} \; \leq \; \frac{\frac{m^{n-2}}{2} + 2c\varepsilon m^{n-2}\sqrt{\frac{T}{W} m^2 \gamma^{m^n}}}{m^{n-2} - \frac{2W}{cm^2}}$$

$$(\gamma^{m^n} \leq e \text{ and } m^n \leq W \leq em^n \text{ by (26)}) \; \leq \; \frac{\frac{m^{n-2}}{2} + 2c\varepsilon m^{n-2}\sqrt{\frac{eT}{m^{n-2}}}}{m^{n-2} - \frac{2em^n}{cm^2}}$$

$$= \; \frac{\frac{1}{2} + 2c\sqrt{\frac{e\varepsilon^2}{m^{n-2}}T}}{1 - \frac{2e}{c}} < 0.99 < 1 - \delta,$$

when $c = 20$, $T < 10^{-5} \cdot \frac{m^{n-2}}{\varepsilon^2}$, and $\delta < 0.01$. This means that, in order to obtain such $\hat{D}$ with probability at least $1 - \delta$, at least $10^{-5} \cdot \frac{m^{n-2}}{\varepsilon^2}$ samples are needed.

# F Missing Proofs from Section 5

## F.1 Proof of Theorem 5.2: the $\Omega\left(\frac{1}{\varepsilon}\log\frac{1}{\delta}\right)$ Lower Bound

The proof uses a reduction from the distinguishing distributions problem (defined in Section 3). We construct two conditionally independent distributions $P^1, P^2$ over the space $\Omega \times \mathcal{S}_1 \times \cdots \times \mathcal{S}_n$ with each $|\mathcal{S}_i| = 2$, $\mathcal{S}_i = \{a, b\}$. Given $T$ samples from either $P^1$ or $P^2$, we want to tell which distribution the samples are coming from. We will show that, if we can solve the forecast aggregation problem, then we can distinguish the two distributions (with high probability), which requires $T = \Omega\left(\frac{1}{d_H^2(P_1,P_2)}\log\frac{1}{\delta}\right) = \Omega\left(\frac{1}{\varepsilon}\log\frac{1}{\delta}\right)$ samples according to Lemma A.5.

Let $c = 32$. We assume $\varepsilon < 2^{-18}$, so that $c\sqrt{\varepsilon} < \frac{1}{16}$. For $P^1$, we let

$$P^1(\omega = 1) = 0.5 - \frac{1}{16n} + \frac{c\sqrt{\varepsilon}}{n} =: p^1.$$

For $P^2$, we let

$$P^2(\omega = 1) = 0.5 - \frac{1}{16n} - \frac{c\sqrt{\varepsilon}}{n} =: p^2.$$

We require that, in the forecast aggregation problem under both distributions $P^1$ and $P^2$, whenever expert $i$ sees signal $a, b$, she reports

$$r_a = 0.5, \qquad r_b = 0,$$

respectively. This gives the following conditional probabilities $P^1(\cdot \mid \omega), P^2(\cdot \mid \omega)$:

$$\begin{bmatrix} P^1(a \mid \omega = 0) \\ P^1(a \mid \omega = 1) \end{bmatrix} = \frac{p^1 - r_b}{r_a - r_b}\begin{bmatrix} \frac{1-r_a}{1-p^1} \\ \frac{r_a}{p^1} \end{bmatrix} = \begin{bmatrix} 1 - \frac{1}{8n} + \frac{2c\sqrt{\varepsilon}}{n} \\ 1 + \frac{1}{8n} - \frac{2c\sqrt{\varepsilon}}{n} \\ 1 \end{bmatrix}, \qquad \begin{bmatrix} P^1(b \mid \omega = 0) \\ P^1(b \mid \omega = 1) \end{bmatrix} = \begin{bmatrix} \frac{1}{4n} - \frac{4c\sqrt{\varepsilon}}{n} \\ 1 + \frac{1}{8n} - \frac{2c\sqrt{\varepsilon}}{n} \\ 0 \end{bmatrix}.$$

$$\tag{34}$$

$$\begin{bmatrix} P^2(a \mid \omega = 0) \\ P^2(a \mid \omega = 1) \end{bmatrix} = \frac{p^2 - r_b}{r_a - r_b}\begin{bmatrix} \frac{1-r_a}{1-p^2} \\ \frac{r_a}{p^2} \end{bmatrix} = \begin{bmatrix} 1 - \frac{1}{8n} - \frac{2c\sqrt{\varepsilon}}{n} \\ 1 + \frac{1}{8n} + \frac{2c\sqrt{\varepsilon}}{n} \\ 1 \end{bmatrix}, \qquad \begin{bmatrix} P^2(b \mid \omega = 0) \\ P^2(b \mid \omega = 1) \end{bmatrix} = \begin{bmatrix} \frac{1}{4n} + \frac{4c\sqrt{\varepsilon}}{n} \\ 1 + \frac{1}{8n} + \frac{2c\sqrt{\varepsilon}}{n} \\ 0 \end{bmatrix}.$$

$$\tag{35}$$

Given $T$ samples from the unknown distribution $P \in \{P^1, P^2\}$, each of which is a vector of $\omega^{(t)}$ and all experts' signals $s_i^{(t)} \in \{a, b\}$, we feed the corresponding reports $r_i^{(t)} \in \{r_a, r_b\}$ and $\omega^{(t)}$ to the forecast aggregation problem and obtain a solution $\hat{f}$, which is an $\varepsilon$-optimal aggregator. We want to use $\hat{f}$ to estimate the prior $p = P(\omega = 1) \in \{p^1, p^2\}$ so that we can tell apart $P^1$ and $P^2$. Recall from Lemma 5.1 that $f^*(\boldsymbol{r}) = \frac{1}{1 + \rho^{n-1}\prod_{i=1}^n \frac{1-r_i}{r_i}}$, where $\rho = \frac{p}{1-p}$. Writing $\rho$ in terms of $f^*(\boldsymbol{r})$, we have

$$\rho = \sqrt[n-1]{\left(\frac{1}{f^*(\boldsymbol{r})} - 1\right)\prod_{i=1}^n \frac{r_i}{1-r_i}}.$$

In particular, when $r_i = r_a = 0.5$ for all $i \in \{1, \dots, n\}$, we have:

$$\rho = \sqrt[n-1]{\frac{1}{f^*(\boldsymbol{r}_{0.5})} - 1}, \qquad \boldsymbol{r}_{0.5} = (0.5, \dots, 0.5).$$

So, we estimate $\rho$ by:

$$\hat{\rho} = \sqrt[n-1]{\frac{1}{\hat{f}(\boldsymbol{r}_{0.5})} - 1}.$$

Now, we want to argue that, if $\hat{f}$ is $\varepsilon$-optimal, then $|\hat{\rho} - \rho|$ is at most $O(\frac{\sqrt{\varepsilon}}{n})$. Consider the function:

$$h(x) = \sqrt[n-1]{\frac{1}{x} - 1},$$

whose derivative is

$$h'(x) = -\frac{1}{n-1}\left(\frac{x}{1-x}\right)^{1-\frac{1}{n-1}}\frac{1}{x^2}.$$

By definition, we have

$$\left|\hat{\rho} - \rho\right| = \left|h\big(\hat{f}(\boldsymbol{r}_{0.5})\big) - h\big(f^*(\boldsymbol{r}_{0.5})\big)\right|. \tag{36}$$

**Claim F.1.** $\frac{1}{2} \le f^*(\boldsymbol{r}_{0.5}) \le \frac{2}{3}$.

*Proof.* For $P \in \{P^1, P^2\}$, its $\rho = \frac{p}{1-p}$ satisfies

$$1 \ge \rho \ge \rho^{n-1} \ge \rho^n \ge \left(\frac{0.5 - \frac{1}{16n} - \frac{\sqrt{\varepsilon}}{cn}}{0.5 + \frac{1}{16n} + \frac{c\sqrt{\varepsilon}}{n}}\right)^n$$

$$= \left(\frac{1 - \frac{1}{8n} - \frac{2c\sqrt{\varepsilon}}{n}}{1 + \frac{1}{8n} + \frac{2c\sqrt{\varepsilon}}{n}}\right)^n > \left(1 - 2\big(\tfrac{1}{8n} + \tfrac{2c\sqrt{\varepsilon}}{n}\big)\right)^n \ge 1 - 2\big(\tfrac{1}{8} + 2c\sqrt{\varepsilon}\big) > \tfrac{1}{2}, \tag{37}$$

where in the last three transitions we used the inequalities $\frac{1-x}{1+x} > 1 - 2x$ and $(1 - x/n)^n \ge 1 - x$ for $x \in (0, 1)$ and the fact that $c\sqrt{\varepsilon} < \frac{1}{16}$. So,

$$f^*(\boldsymbol{r}_{0.5}) = \frac{1}{1 + \rho^{n-1}} \in \left[\frac{1}{1+1}, \frac{1}{1 + \frac{1}{2}}\right] = \left[\frac{1}{2}, \frac{2}{3}\right],$$

which proves the claim. $\qquad\square$

With Claim F.1, we can without loss of generality assume $\frac{1}{2} \le \hat{f}(\boldsymbol{r}_{0.5}) \le \frac{2}{3}$ as well (otherwise, we can truncate $\hat{f}(\boldsymbol{r}_{0.5})$ to this range; this only reduces the approximation error $\mathbb{E}\big[|\hat{f}(\boldsymbol{r}) - f^*(\boldsymbol{r})|^2\big]$).

**Claim F.2.** For $\frac{1}{2} \le x \le \frac{2}{3}$, $|h'(x)| \le \frac{8}{n-1}$.

*Proof.*

$$|h'(x)| = \frac{1}{n-1}\Big(\frac{x}{1-x}\Big)^{1 - \frac{1}{n-1}}\frac{1}{x^2} \le \frac{1}{n-1}\Big(\frac{\frac{2}{3}}{1 - \frac{2}{3}}\Big)^{1 - \frac{1}{n-1}}\frac{1}{(\frac{1}{2})^2} = \frac{4}{n-1}\cdot 2^{1 - \frac{1}{n-1}} \le \frac{8}{n-1}. \quad\square$$

**Claim F.3.** If $\hat{f}$ is $\varepsilon$-optimal, then $|\hat{f}(\boldsymbol{r}_{0.5}) - f^*(\boldsymbol{r}_{0.5})| < 2\sqrt{\varepsilon}$.

*Proof.* If $\hat{f}$ is $\varepsilon$-optimal, i.e., $\mathbb{E}\big[|\hat{f}(\boldsymbol{r}) - f^*(\boldsymbol{r})|^2\big] \le \varepsilon$, then, by Jensen's inequality $\mathbb{E}[X^2] \ge \mathbb{E}[X]^2$, we have

$$\sqrt{\varepsilon} \ge \mathbb{E}\big[|\hat{f}(\boldsymbol{r}) - f^*(\boldsymbol{r})|\big] = \sum_{\boldsymbol{r}} P(\boldsymbol{r})|\hat{f}(\boldsymbol{r}) - f^*(\boldsymbol{r})| \ge P(\boldsymbol{r}_{0.5})|\hat{f}(\boldsymbol{r}_{0.5}) - f^*(\boldsymbol{r}_{0.5})|. \tag{38}$$

For both $P \in \{P^1, P^2\}$, we have

$$P(\boldsymbol{r}_{0.5}) = p \cdot P(\boldsymbol{r}_{0.5} \mid \omega = 1) + (1-p) \cdot P(\boldsymbol{r}_{0.5} \mid \omega = 0)$$
$$= p \cdot P(a \mid \omega = 1)^n + (1-p) \cdot P(a \mid \omega = 0)^n$$
$$\ge p \cdot 1 + (1-p) \cdot \left(\frac{1 - \frac{1}{8n} - \frac{2c\sqrt{\varepsilon}}{n}}{1 + \frac{1}{8n} + \frac{2c\sqrt{\varepsilon}}{n}}\right)^n \overset{\text{by (37)}}{>} \frac{1}{2},$$

Plugging $P(\boldsymbol{r}_{0.5}) > \frac{1}{2}$ into (38), we get $|\hat{f}(\boldsymbol{r}_{0.5}) - f^*(\boldsymbol{r}_{0.5})| < 2\sqrt{\varepsilon}$. $\qquad\square$

From (36), Claim F.1, Claim F.2, and Claim F.3, we get

$$\left|\hat{\rho} - \rho\right| = \left|h\big(\hat{f}(\boldsymbol{r}_{0.5})\big) - h\big(f^*(\boldsymbol{r}_{0.5})\big)\right| \le \frac{8}{n-1}|\hat{f}(\boldsymbol{r}_{0.5}) - f^*(\boldsymbol{r}_{0.5})| < \frac{8}{n-1}\cdot 2\sqrt{\varepsilon} = \frac{16}{n-1}\sqrt{\varepsilon}. \tag{39}$$

Since $p = \frac{\rho}{1+\rho}$ as a function of $\rho$ has a bounded derivative $\frac{\partial p}{\partial \rho} = \frac{1}{(1+\rho)^2} \le 1$, Equation (39) implies

$$|\hat{p} - p| < \frac{16}{n-1}\sqrt{\varepsilon}$$

if we use $\hat{p} = \frac{\hat{\rho}}{1+\hat{\rho}}$ as an estimate of $p$. This allows us to tell part $P^1$ and $P^2$ because the difference between $p^1$ and $p^2$ is greater than twice of our estimation error $|\hat{p} - p|$:

$$|p^1 - p^2| = \frac{2c\sqrt{\varepsilon}}{n} = \frac{64\sqrt{\varepsilon}}{n} \geq \frac{64\sqrt{\varepsilon}}{2(n-1)} = \frac{32\sqrt{\varepsilon}}{n-1} > 2|\hat{p} - p|.$$

Therefore, we can tell part $P^1$ and $P^2$ by checking whether $p^1$ or $p^2$ is closer to $\hat{p}$.

Finally, we upper bound the squared Hellinger distance between $P^1$ and $P^2$. This will give the sample complexity lower bound we want.

**Claim F.4.** $d_{\mathrm{H}}^2(P^1, P^2) \leq O(c^2\varepsilon)$.

*Proof.* For the marginal distributions of $\omega$, $P_{\omega}^1$ and $P_{\omega}^2$, according to Lemma A.2 and the fact that $1 \geq \frac{P_{\omega}^2(\omega)}{P_{\omega}^1(\omega)} = \frac{1 - \frac{1}{8n} - \frac{2c\sqrt{\varepsilon}}{n}}{1 - \frac{1}{8n} + \frac{2c\sqrt{\varepsilon}}{n}} = 1 - \frac{\frac{4c\sqrt{\varepsilon}}{n}}{1 - \frac{1}{8n} + \frac{2c\sqrt{\varepsilon}}{n}} = 1 - O(\frac{c\sqrt{\varepsilon}}{n})$, we have

$$d_{\mathrm{H}}^2(P_{\omega}^1, P_{\omega}^2) \leq O\left(\left(\frac{c\sqrt{\varepsilon}}{n}\right)^2\right) = O\left(\frac{c^2\varepsilon}{n^2}\right). \tag{40}$$

Given $\omega = 0$ or $1$, we consider the conditional distributions of each $s_i$, $P_{s_i|\omega}^1$ and $P_{s_i|\omega}^2$. For $s_i = a$, we have

$$1 \geq \frac{P^2(a \mid \omega)}{P^1(a \mid \omega)} \geq \frac{1 - \frac{1}{8n} - \frac{2c\sqrt{\varepsilon}}{n}}{1 + \frac{1}{8n} + \frac{2c\sqrt{\varepsilon}}{n}} \cdot \frac{1 + \frac{1}{8n} - \frac{2c\sqrt{\varepsilon}}{n}}{1 - \frac{1}{8n} + \frac{2c\sqrt{\varepsilon}}{n}}$$

$$= \frac{1 + \frac{1}{8n} - \frac{2c\sqrt{\varepsilon}}{n}}{1 + \frac{1}{8n} + \frac{2c\sqrt{\varepsilon}}{n}} \cdot \frac{1 - \frac{1}{8n} - \frac{2c\sqrt{\varepsilon}}{n}}{1 - \frac{1}{8n} + \frac{2c\sqrt{\varepsilon}}{n}}$$

$$\left(\frac{a-x}{a+x} \geq 1 - \frac{2x}{a}\right) \geq \left(1 - \frac{\frac{4c\sqrt{\varepsilon}}{n}}{1 + \frac{1}{8n}}\right) \cdot \left(1 - \frac{\frac{4c\sqrt{\varepsilon}}{n}}{1 - \frac{1}{8n}}\right) = 1 - O\left(\frac{c\sqrt{\varepsilon}}{n}\right).$$

For $s_i = b$, we have

$$1 \geq \frac{P^1(b \mid \omega)}{P^2(b \mid \omega)} \geq \frac{\frac{1}{4n} - \frac{4c\sqrt{\varepsilon}}{n}}{1 + \frac{1}{8n} - \frac{2c\sqrt{\varepsilon}}{n}} \cdot \frac{1 + \frac{1}{8n} + \frac{2c\sqrt{\varepsilon}}{n}}{\frac{1}{4n} + \frac{4c\sqrt{\varepsilon}}{n}} \geq \frac{\frac{1}{4n} - \frac{4c\sqrt{\varepsilon}}{n}}{\frac{1}{4n} + \frac{4c\sqrt{\varepsilon}}{n}} = \frac{1 - 16c\sqrt{\varepsilon}}{1 + 16c\sqrt{\varepsilon}} = 1 - O(c\sqrt{\varepsilon}).$$

So, $d_{\mathrm{H}}^2(P_{s_i|\omega}^1, P_{s_i|\omega}^2)$ can be upper bounded as follows:

$$d_{\mathrm{H}}^2(P_{s_i|\omega}^1, P_{s_i|\omega}^2) = \frac{1}{2}\left[\left(\sqrt{P^1(a \mid \omega)} - \sqrt{P^2(a \mid \omega)}\right)^2 + \left(\sqrt{P^1(b \mid \omega)} - \sqrt{P^2(b \mid \omega)}\right)^2\right]$$

$$= \frac{1}{2}\left[P^1(a \mid \omega)\left(1 - \sqrt{\frac{P^2(a \mid \omega)}{P^1(a \mid \omega)}}\right)^2 + P^2(b \mid \omega)\left(1 - \sqrt{\frac{P^1(b \mid \omega)}{P^2(b \mid \omega)}}\right)^2\right]$$

$$\leq \frac{1}{2}\left[1 \cdot \left(1 - \sqrt{1 - O\left(\frac{c\sqrt{\varepsilon}}{n}\right)}\right)^2 + \left(\frac{1}{4n} + \frac{4c\sqrt{\varepsilon}}{n}\right) \cdot \left(1 - \sqrt{1 - O(c\sqrt{\varepsilon})}\right)^2\right]$$

$$(\text{since } 1 - \sqrt{1-x} \leq x) \leq \frac{1}{2}\left[O\left(\frac{c\sqrt{\varepsilon}}{n}\right)^2 + O\left(\frac{1}{n}\right) \cdot O(c\sqrt{\varepsilon})^2\right]$$

$$= O\left(\frac{c^2\varepsilon}{n}\right). \tag{41}$$

Since $P^1 = P_{\omega}^1 \cdot \prod_{i=1}^n P_{s_i|\omega}^1$ and $P^2 = P_{\omega}^2 \cdot \prod_{i=1}^n P_{s_i|\omega}^2$, we have, by Lemma A.3 and Lemma A.4,

$$d_{\mathrm{H}}^2(P^1, P^2) \leq d_{\mathrm{H}}^2(P_{\omega}^1, P_{\omega}^2) + \max_{\omega \in \{0,1\}}\left\{d_{\mathrm{H}}^2\left(\prod_{i=1}^n P_{s_i|\omega}^1, \prod_{i=1}^n P_{s_i|\omega}^2\right)\right\}$$

$$\leq d_{\mathrm{H}}^2(P_{\omega}^1, P_{\omega}^2) + \max_{\omega \in \{0,1\}}\left\{n \cdot d_{\mathrm{H}}^2(P_{s_i|\omega}^1, P_{s_i|\omega}^2)\right\}$$

$$(40) \text{ and } (41) \leq O\left(\frac{c^2\varepsilon}{n^2}\right) + \max_{\omega \in \{0,1\}}\left\{n \cdot O\left(\frac{c^2\varepsilon}{n}\right)\right\}$$

$$= O(c^2\varepsilon). \qquad \square$$

Therefore, according to Lemma A.5, to tell apart $P^1$ and $P^2$ with probability at least $1 - \delta$ we need at least

$$T = \Omega\Big(\frac{1}{d_{\mathrm{H}}^2(P^1, P^2)} \log \frac{1}{\delta}\Big) = \Omega\Big(\frac{1}{c^2 \varepsilon} \log \frac{1}{\delta}\Big)$$

samples. This concludes the proof.

## G  Missing Proofs from Section B

### G.1  Proof of Theorem B.2

#### G.1.1  Additional Notations and Lemmas

We introduce some additional notations and lemmas for the proof. Let $\mu_0$ be the expected average report of all experts conditioning on $\omega = 0$:

$$\mu_0 = \frac{1}{n} \sum_{i=1}^{n} \mathbb{E}[r_i \mid \omega = 0] = \frac{1}{n} \sum_{i=1}^{n} \mathbb{E}_{s_i \mid \omega = 0}\big[P(\omega = 1 \mid s_i) \mid \omega = 0\big], \tag{42}$$

which is equal to the expected prediction of $\omega$ given expert $i$'s signal $s_i$ where $s_i$ is distributed conditioning on $\omega = 0$, averaged over all experts. Symmetrically, let

$$\mu_1 = \frac{1}{n} \sum_{i=1}^{n} \mathbb{E}[1 - r_i \mid \omega = 1] = \frac{1}{n} \sum_{i=1}^{n} \mathbb{E}_{s_i \mid \omega = 1}\big[P(\omega = 0 \mid s_i) \mid \omega = 1\big]. \tag{43}$$

Recall that $p = P(\omega = 1)$.

**Fact G.1.** $(1 - p)\mu_0 = p\mu_1$.

*Proof.* For each expert $i$, by the law of total expectation and the fact that $r_i = P(\omega = 1 \mid s_i)$, we have the following equations:

$$(1 - p) \cdot \mathbb{E}[r_i \mid \omega = 0] + p \cdot \mathbb{E}[r_i \mid \omega = 1] = P(\omega = 0) \cdot \mathbb{E}[r_i \mid \omega = 0] + P(\omega = 1) \cdot \mathbb{E}[r_i \mid \omega = 1]$$
$$= \mathbb{E}[r_i]$$
$$= \mathbb{E}_{s_i}\big[P(\omega = 1 \mid s_i)\big] = P(\omega = 1) = p.$$

Subtracting $p$ from both sides, we get

$$(1 - p) \cdot \mathbb{E}[r_i \mid \omega = 0] - p \cdot \mathbb{E}[1 - r_i \mid \omega = 1] = 0.$$

Averaging over all experts $i \in \{1, \ldots, n\}$, we conclude that

$$(1 - p) \cdot \frac{1}{n} \sum_{i=1}^{n} \mathbb{E}[r_i \mid \omega = 0] - p \cdot \frac{1}{n} \sum_{i=1}^{n} \mathbb{E}[1 - r_i \mid \omega = 1] = 0. \qquad \square$$

**Lemma G.2.** *If $(1 - p)\mu_0 = p\mu_1 < \frac{\varepsilon}{2}$, then the averaging aggregator $f_{avg}(\boldsymbol{r}) = \frac{1}{n} \sum_{i=1}^{n} r_i$ is $\varepsilon$-optimal.*

*Proof.* If $(1 - p)\mu_0 = p\mu_1 < \frac{\varepsilon}{2}$, then the expected loss of $f_{avg}$ is at most

$$L_P(f_{avg}) = \mathbb{E}\big[|f_{avg} - \omega|^2\big]$$
$$= p\mathbb{E}\big[(f_{avg}(\boldsymbol{r}) - 1)^2 \mid \omega = 1\big] + (1 - p)\mathbb{E}\big[(f_{avg}(\boldsymbol{r}) - 0)^2 \mid \omega = 0\big]$$
$$\leq p\mathbb{E}\big[1 - f_{avg}(\boldsymbol{r}) \mid \omega = 1\big] + (1 - p)\mathbb{E}\big[f_{avg}(\boldsymbol{r}) \mid \omega = 0\big]$$
$$= p\mathbb{E}\big[1 - \frac{1}{n} \sum_{i=1}^{n} r_i \mid \omega = 1\big] + (1 - p)\mathbb{E}\big[\frac{1}{n} \sum_{i=1}^{n} r_i \mid \omega = 0\big]$$
$$= p\mu_1 + (1 - p)\mu_0$$
$$< \frac{\varepsilon}{2} + \frac{\varepsilon}{2} = \varepsilon,$$

which implies that $f_{avg}$ is $\varepsilon$-optimal. $\qquad \square$

The following lemma says that $O(\frac{1}{\varepsilon} \log \frac{1}{\delta})$ samples are sufficient to tell whether the mean of a random variable is below $\varepsilon$ or above $\frac{\varepsilon}{2}$:

**Lemma G.3.** *Given $T = \frac{40}{\varepsilon} \log \frac{2}{\delta}$ i.i.d. samples $X^{(1)}, \ldots, X^{(T)}$ of a random variable $X \in [0, 1]$ with unknown mean $\mathbb{E}[X] = \mu$, with probability at least $1 - \delta$ we can tell whether $\mu < \varepsilon$ or $\mu \geq \frac{\varepsilon}{2}$. This can be done by checking whether the empirical mean $\hat{\mu} = \frac{1}{T} \sum_{t=1}^{T} X^{(t)}$ is $< \frac{3}{4}\varepsilon$ or $\geq \frac{3}{4}\varepsilon$.*

*Proof.* If $\mu \geq \varepsilon$, using the multiplicative version of Chernoff bound we have

$$\Pr\left[\hat{\mu} < \frac{3}{4}\varepsilon\right] \leq \Pr\left[\hat{\mu} < \frac{3}{4}\mu\right] \leq e^{-\frac{(\frac{1}{4})^2 \mu T}{2}} \leq e^{-\frac{\varepsilon T}{32}} \leq \delta.$$

Namely, with probability at least $1 - \delta$, it holds that

$$\hat{\mu} \geq \frac{3}{4}\varepsilon.$$

If $\mu < \varepsilon$, then using the additive version of Chernoff–Hoeffding theorem, we have

$$\Pr\left[\hat{\mu} < \mu - \frac{\varepsilon}{4}\right] \leq e^{-D(\mu - \frac{\varepsilon}{4}||\mu)T},$$

$$\Pr\left[\hat{\mu} > \mu + \frac{\varepsilon}{4}\right] \leq e^{-D(\mu + \frac{\varepsilon}{4}||\mu)T},$$

where $D(x||y) = x \ln \frac{x}{y} + (1 - x) \ln \frac{1-x}{1-y}$. Using the inequality $D(x||y) \geq \frac{(x-y)^2}{2y}$ for $x \leq y$ and $D(x||y) \geq \frac{(x-y)^2}{2x}$ for $x \geq y$, we obtain:

$$\Pr\left[\hat{\mu} < \mu - \frac{\varepsilon}{4}\right] \leq e^{-\frac{(\frac{\varepsilon}{4})^2}{2\mu}T} \leq e^{-\frac{(\frac{\varepsilon}{4})^2}{2\varepsilon}T} = e^{-\frac{\varepsilon}{32}T} \leq \frac{\delta}{2},$$

$$\Pr\left[\hat{\mu} > \mu + \frac{\varepsilon}{4}\right] \leq e^{-\frac{(\frac{\varepsilon}{4})^2}{2(\mu + \frac{\varepsilon}{4})}T} \leq e^{-\frac{(\frac{\varepsilon}{4})^2}{2(\frac{5\varepsilon}{4})}T} = e^{-\frac{\varepsilon}{40}T} \leq \frac{\delta}{2}.$$

By a union bound, with probability at least $1 - \delta$, we have

$$|\hat{\mu} - \mu| \leq \frac{\varepsilon}{4}.$$

Combining the case of $\mu \geq \varepsilon$ and $\mu < \varepsilon$, we conclude that: with probability at least $1 - \delta$,

- If $\hat{\mu} < \frac{3}{4}\varepsilon$, then we must have $\mu < \varepsilon$.

- If $\hat{\mu} \geq \frac{3}{4}\varepsilon$, then we have $\mu \geq \varepsilon$ or $\varepsilon > \mu \geq \hat{\mu} - \frac{\varepsilon}{4} \geq \frac{\varepsilon}{2}$. In either case, we have $\mu \geq \frac{\varepsilon}{2}$.

$\square$

The last lemma we will use shows how to estimate the unknown value of $\rho = \frac{p}{1-p} = \frac{P(\omega=1)}{P(\omega=0)}$ with accuracy $\Delta$ using $T = O(\frac{1}{n\Delta^2} \log \frac{1}{\delta})$ samples. Notice that, if one simply uses the empirical value $\hat{\rho} = \frac{\sum_{t=1}^{T} \mathbb{1}\{\omega^{(t)}=1\}}{\sum_{t=1}^{T} \mathbb{1}\{\omega^{(t)}=0\}}$ to estimate $\rho$, then by Chernoff bound this needs $T = O(\frac{1}{\Delta^2} \log \frac{1}{\delta})$ samples, which is larger than what we claim by a factor of $n$. This sub-optimality is because one only uses the $\omega^{(t)}$'s in the samples to estimate $\rho$, wasting the reports $r_i^{(t)}$'s. By using $r_i^{(t)}$'s to estimate $\rho$, we can reduce the number of samples by a factor of $n$. The basic idea is the following: According to Fact G.1, we have $\rho = \frac{p}{1-p} = \frac{\mu_0}{\mu_1} = \frac{\mathbb{E}[\sum_{i=1}^{n} r_i | \omega=0]}{\mathbb{E}[\sum_{i=1}^{n} (1-r_i) | \omega=1]}$. The numerator $\mathbb{E}[\sum_{i=1}^{n} r_i | \omega = 0]$ and the denominator $\mathbb{E}[\sum_{i=1}^{n} (1 - r_i) | \omega = 1]$ can be estimated from samples of $r_i^{(t)}$'s where $\omega^{(t)} = 0$ and 1 respectively. The total number of $r_i^{(t)}$'s is $Tn$, because we have $n$ experts per sample. This reduces the needed number of samples by a factor of $n$. Formally:

**Lemma G.4.** *For conditionally independent distribution $P$, we can estimate $\rho = \frac{P(\omega=1)}{P(\omega=0)}$ with accuracy $\frac{|\hat{\rho}-\rho|}{\rho} \leq \Delta < 1$ (equivalently, $\frac{\hat{\rho}}{\rho} \in 1 \pm \Delta$) with probability at least $1 - \delta$ using*

$$T = O\left(\frac{1}{(1-p)\mu_0 \cdot n \cdot \Delta^2} \log \frac{1}{\delta} + \frac{1}{\min\{p, 1-p\}} \log \frac{1}{\delta}\right)$$

*samples of $(\omega^{(t)}, \boldsymbol{r}^{(t)})$'s, by letting $\hat{\rho} = \dfrac{\frac{1}{\#_0}\sum_{t:\omega^{(t)}=0}\sum_{i=1}^{n} r_i^{(t)}}{\frac{1}{\#_1}\sum_{t:\omega^{(t)}=1}\sum_{i=1}^{n}(1-r_i^{(t)})}$, where $\#_0$ and $\#_1$ are the numbers of samples with $\omega^{(t)} = 0$ and $1$ respectively.*

*Proof.* Recall that $p = P(\omega = 1)$, $1 - p = P(\omega = 0)$, and $\rho = \frac{p}{1-p}$. According to Fact G.1 (which says $(1-p)\mu_0 = p\mu_1$), we have

$$\rho = \frac{p}{1-p} = \frac{\mu_0}{\mu_1} = \frac{\sum_{i=1}^{n}\mathbb{E}[r_i \mid \omega = 0]}{\sum_{i=1}^{n}\mathbb{E}[1 - r_i \mid \omega = 1]}. \tag{44}$$

Consider the following way of estimating $\rho$ from $T$ samples $(\omega^{(t)}, \boldsymbol{r}^{(t)})_{t=1}^{T}$: Let $\#_0$, $\#_1$ be the numbers of samples where $\omega^{(t)} = 0, 1$, respectively:

$$\#_0 = \sum_{t=1}^{T}\mathbb{1}\{\omega^{(t)} = 0\}, \qquad \#_1 = \sum_{t=1}^{T}\mathbb{1}\{\omega^{(t)} = 1\}.$$

We let

$$\hat{\rho} = \frac{\frac{1}{\#_0}\sum_{t:\omega^{(t)}=0}\sum_{i=1}^{n} r_i^{(t)}}{\frac{1}{\#_1}\sum_{t:\omega^{(t)}=1}\sum_{i=1}^{n}(1 - r_i^{(t)})}. \tag{45}$$

Now, we compare the $\hat{\rho}$ in (45) and the $\rho$ in (44): we see that $\frac{1}{\#_0}\sum_{t:\omega^{(t)}=0}\sum_{i=1}^{n} r_i^{(t)}$ is an (unbiased) estimate of the numerator $\sum_{i=1}^{n}\mathbb{E}[r_i \mid \omega = 0] = n\mu_0$ and that $\frac{1}{\#_1}\sum_{t:\omega^{(t)}=1}\sum_{i=1}^{n}(1 - r_i^{(t)})$ is an (unbiased) estimate of the denominator $\sum_{i=1}^{n}\mathbb{E}[1 - r_i \mid \omega = 1] = n\mu_1$. We use Chernoff bounds to argue that the accuracy of the two estimates is within $\Delta$ with high probability if $\#_0$ and $\#_1$ are big enough. Suppose that, when drawing the $T$ samples, we draw all the $\omega^{(t)}$'s first (and hence $\#_0$, $\#_1$ are determined), and then draw all the $r_i^{(t)}$'s. After all the $\omega^{(t)}$'s are drawn, the $r_i^{(t)}$'s become independent, because the signals $s_i^{(t)}$'s are conditionally independent given $\omega^{(t)}$. Therefore, we can use Chernoff bounds:

$$\Pr\left[\left|\frac{1}{\#_0}\sum_{t:\omega^{(t)}=0}\sum_{i=1}^{n} r_i^{(t)} - n\mu_0\right| > \Delta n\mu_0\right] \le 2e^{-\frac{\#_0 n\mu_0 \Delta^2}{3}},$$

$$\Pr\left[\left|\frac{1}{\#_1}\sum_{t:\omega^{(t)}=1}\sum_{i=1}^{n}(1 - r_i^{(t)}) - n\mu_1\right| > \Delta n\mu_1\right] \le 2e^{-\frac{\#_1 n\mu_1 \Delta^2}{3}}.$$

Requiring $\delta \ge 2e^{-\frac{\#_0 n\mu_0 \Delta^2}{3}}$ and $\delta \ge 2e^{-\frac{\#_1 n\mu_1 \Delta^2}{3}}$, namely,

$$\#_0 \ge \frac{3}{n\mu_0 \Delta^2}\log\frac{2}{\delta}, \qquad \#_1 \ge \frac{3}{n\mu_1 \Delta^2}\log\frac{2}{\delta}, \tag{46}$$

we have, with probability at least $1 - 2\delta$, both of the following hold:

$$\frac{1}{\#_0}\sum_{t:\omega^{(t)}=0}\sum_{i=1}^{n} r_i^{(t)} \in (1 \pm \Delta)n\mu_0, \qquad \frac{1}{\#_1}\sum_{t:\omega^{(t)}=1}\sum_{i=1}^{n}(1 - r_i^{(t)}) \in (1 \pm \Delta)n\mu_1, \tag{47}$$

Then, we argue that (46) can be satisfied with high probability if $T$ is large enough. This is again done by a Chernoff bound: since $\mathbb{E}[\#_j] = \mathbb{E}[\sum_{t=1}^{T}\mathbb{1}\{\omega^{(t)} = j\}] = T \cdot P(\omega = j)$, for $j = 0, 1$, we have

$$\Pr\left[\left|\#_0 - T(1-p)\right| \ge \tfrac{1}{2}T(1-p)\right] \le 2e^{-\frac{T(1-p)(\frac{1}{2})^2}{3}}, \qquad \Pr\left[\left|\#_1 - Tp\right| \ge \tfrac{1}{2}Tp\right] \le 2e^{-\frac{Tp(\frac{1}{2})^2}{3}}. \tag{48}$$

So, if we are given

$$T \ge \frac{12}{\min\{p, 1 - p\}}\log\frac{2}{\delta} \tag{49}$$

samples, then we can ensure that with probability at least $1 - 2\delta$, it holds $\#_0 \geq \frac{1}{2}T(1 - p)$ and $\#_1 \geq \frac{1}{2}Tp$. Then, in order for (46) to be satisfied, we can let

$$\frac{1}{2}T(1-p) \geq \frac{3}{n\mu_0\Delta^2}\log\frac{2}{\delta}, \qquad \frac{1}{2}Tp \geq \frac{3}{n\mu_1\Delta^2}\log\frac{2}{\delta}.$$

This gives

$$T \geq \max\left\{\frac{6}{(1-p)\mu_0 \cdot n\Delta^2}\log\frac{2}{\delta}, \frac{6}{p\mu_1 \cdot n\Delta^2}\log\frac{2}{\delta}\right\} \overset{(1-p)\mu_0=p\mu_1}{=} \frac{6}{(1-p)\mu_0 \cdot n\Delta^2}\log\frac{2}{\delta}. \tag{50}$$

Both (49) and (50) are satisfied when

$$T \geq \frac{6}{(1-p)\mu_0 \cdot n\Delta^2}\log\frac{2}{\delta} + \frac{12}{\min\{p, 1-p\}}\log\frac{2}{\delta}.$$

To conclude, if we are given $T = \frac{6}{(1-p)\mu_0 \cdot n\Delta^2}\log\frac{2}{\delta} + \frac{12}{\min\{p,1-p\}}\log\frac{2}{\delta}$ samples, then with probability at least $1 - 4\delta$, (47) holds, which implies

$$\hat{\rho} \in \frac{(1\pm\Delta)\mu_0}{(1\pm\Delta)\mu_1} = \frac{(1\pm\Delta)}{(1\pm\Delta)}\rho \subseteq (1\pm4\Delta)\rho \implies \frac{|\hat{\rho} - \rho|}{\rho} \leq 4\Delta,$$

for $\Delta < \frac{1}{4}$. $\qquad\qquad\qquad\qquad\qquad\qquad\qquad\qquad\qquad\qquad\qquad\qquad\qquad\qquad$ $\square$

### G.1.2 The Proof

We want to show the $O(\frac{1}{\varepsilon n(\frac{\gamma}{1+\gamma})^2}\log\frac{1}{\delta} + \frac{1}{\varepsilon}\log\frac{1}{\delta})$ sample complexity upper bound for the case where experts have $\gamma$-strongly informative signals.

We first use $O(\frac{1}{\varepsilon}\log\frac{1}{\delta})$ samples tell whether $(1-p)\mu_0 = p\mu_1 < \frac{\varepsilon}{2}$ or $(1-p)\mu_0 = p\mu_1 \geq \frac{\varepsilon}{4}$. We note that

$$(1-p)\mu_0 = P(\omega = 0) \cdot \mathbb{E}\left[\frac{1}{n}\sum_{i=1}^n r_i \mid \omega = 0\right] = \mathbb{E}\left[\mathbb{1}\{\omega = 0\} \cdot \frac{1}{n}\sum_{i=1}^n r_i\right],$$

which is the expectation of the random variable $X = \mathbb{1}\{\omega = 0\} \cdot \frac{1}{n}\sum_{i=1}^n r_i$. So, according to Lemma G.3, we can tell whether $(1-p)\mu_0 < \frac{\varepsilon}{2}$ or $\geq \frac{\varepsilon}{4}$ with probability at least $1 - \delta$ using $O(\frac{1}{\varepsilon}\log\frac{1}{\delta})$ samples of $X$. If $(1-p)\mu_0 = p\mu_1 < \frac{\varepsilon}{2}$, then according to Lemma G.2, the averaging aggregator $f_{avg}(\boldsymbol{r}) = \frac{1}{n}\sum_{i=1}^n r_i$ is $\varepsilon$-optimal. We hence obtained an $\varepsilon$-optimal aggregator in this case. So, in the following proof, we assume $(1-p)\mu_0 = p\mu_1 \geq \frac{\varepsilon}{4}$.

For each expert $i$, let $\mathcal{S}_i^1 = \{s_i \in \mathcal{S}_i : \frac{P(s_i|\omega=1)}{P(s_i|\omega=0)} \geq 1 + \gamma\}$ be its set of $\gamma$-strongly informative signals that are more likely to be realized under $\omega = 1$ than under $\omega = 0$. Let $\mathcal{S}_i^0 = \mathcal{S}_i \setminus \mathcal{S}_i^1 = \{s_i \in \mathcal{S}_i : \frac{P(s_i|\omega=1)}{P(s_i|\omega=0)} \leq \frac{1}{1+\gamma}\}$ be the set of signals that are more likely to be realized under $\omega = 0$. Since $\frac{r_i}{1-r_i} = \frac{P(s_i|\omega=1)}{P(s_i|\omega=0)}\rho$ by Equation (1), whenever an expert receives a signal in $\mathcal{S}_i^1$, its report satisfies

$$\frac{r_i}{1 - r_i} \geq (1 + \gamma)\rho, \quad \forall s_i \in \mathcal{S}_i^1; \tag{51}$$

and whenever it receives a signal in $\mathcal{S}_i^0$, its report satisfies

$$\frac{r_i}{1 - r_i} \leq \frac{1}{1 + \gamma}\rho, \quad \forall s_i \in \mathcal{S}_i^0. \tag{52}$$

We will use the notation $P(\mathcal{S}_i^u \mid \omega) = P(s_i \in \mathcal{S}_i^u \mid \omega) = \sum_{s_i \in \mathcal{S}_i^u} P(s_i \mid \omega)$, for $u \in \{0, 1\}$. Given a set of $n$ signals $s_1, \ldots, s_n$, one per expert, we let $X^1 = \sum_{i=1}^n \mathbb{1}\{s_i \in \mathcal{S}_i^1\}$ be the total number of signals that belong to the $\mathcal{S}_i^1$ sets, and similarly let $X^0 = \sum_{i=1}^n \mathbb{1}\{s_i \in \mathcal{S}_i^0\}$. We have $X^0 + X^1 = n$, and by definition,

$$\mathbb{E}[X^1 \mid \omega = 1] = \sum_{i=1}^n P(\mathcal{S}_i^1 \mid \omega = 1) \geq (1+\gamma)P(\mathcal{S}_i^1 \mid \omega = 0) = (1+\gamma)\mathbb{E}[X^1 \mid \omega = 0]. \tag{53}$$

$$\mathbb{E}[X^0 \mid \omega = 0] = \sum_{i=1}^n P(\mathcal{S}_i^0 \mid \omega = 0) \geq (1+\gamma)P(\mathcal{S}_i^0 \mid \omega = 1) = (1+\gamma)\mathbb{E}[X^0 \mid \omega = 1]. \tag{54}$$

**Claim G.5.** *At least one of $\mathbb{E}[X^1 \mid \omega = 1]$ and $\mathbb{E}[X^0 \mid \omega = 0]$ is $\geq \frac{n}{2}$.*

*Proof.* Suppose on the contrary both $\mathbb{E}[X^1 \mid \omega = 1]$ and $\mathbb{E}[X^0 \mid \omega = 0]$ are $< \frac{n}{2}$. Then, from (54) we have

$$\mathbb{E}[X^0 \mid \omega = 1] \leq \frac{1}{1 + \gamma} \mathbb{E}[X^0 \mid \omega = 0] < \frac{n}{2}.$$

This implies $n = \mathbb{E}[X^0 + X^1 \mid \omega = 1] < \frac{n}{2} + \frac{n}{2} = n$, a contradiction. $\qquad\square$

Let $u \in \{0, 1\}$ be an index such that

$$\mathbb{E}[X^u \mid \omega = u] \geq \frac{n}{4}. \tag{55}$$

Claim G.5 guarantees that such a $u$ exists. We construct a "hypothetical" aggregator $f_{\text{hypo}}$ that, having access to $\rho$ and $\mathbb{E}[X^u \mid \omega]$, predicts whether $\omega = 0$ or 1 by counting the number $X^u$ of signals that belong to the $\mathcal{S}_i^u$ sets and comparing it with its expectations under $\omega = 0$ and 1, $\mathbb{E}[X^u \mid \omega = 0]$ and $\mathbb{E}[X^u \mid \omega = 1]$, respectively. Specifically, given reports $\boldsymbol{r} = (r_1, \ldots, r_n)$ as input, with corresponding unobserved signals $\boldsymbol{s} = (s_1, \ldots, s_n)$, $f_{\text{hypo}}$ does the following:

(1) If $u = 1$, count how many reports $r_i$'s satisfy $\frac{r_i}{1 - r_i} \geq (1 + \gamma)\rho$; If $u = 0$, count how many reports $r_i$'s satisfy $\frac{r_i}{1 - r_i} \leq \frac{1}{1 + \gamma}\rho$. According to (51) and (52), this number is exactly equal to the number of signals that belong to the $\mathcal{S}_i^u$ sets, $X^u$.

(2) Then, check whether $X^u$ is closer (in terms of absolute difference) to $\mathbb{E}[X^u \mid \omega = u]$ or $\mathbb{E}[X^u \mid \omega = 1 - u]$. If $X^u$ is closer to $\mathbb{E}[X^u \mid \omega = u]$, output $f_{\text{hypo}}(\boldsymbol{r}) = u$; otherwise, output $f_{\text{hypo}}(\boldsymbol{r}) = 1 - u$.

We claim that $f_{\text{hypo}}$ is $\varepsilon$-optimal.

**Claim G.6.** *Given $\frac{\gamma}{1+\gamma} \geq 8\sqrt{\frac{2}{n} \log \frac{2}{\varepsilon}}$ and $\mathbb{E}[X^u \mid \omega = u] \geq \frac{n}{4}$, $f_{\text{hypo}}$ is $\varepsilon$-optimal.*

*Proof.* Given either $\omega = 0$ or 1, consider the conditional random draw of signals $s_1, \ldots, s_n$. Because $X^u = \sum_{i=1}^n \mathbb{1}\{s_i \in \mathcal{S}_i^u\}$ and the random variables $\mathbb{1}\{s_i \in \mathcal{S}_i^u\}$, $i = 1, \ldots, n$, are $[0, 1]$-bounded and independent conditioning $\omega$, by Hoeffding's inequality we have

$$\Pr\left[\left|X^u - \mathbb{E}[X^u \mid \omega]\right| \geq a \,\Big|\, \omega\right] \leq 2e^{-\frac{2a^2}{n}}.$$

Let

$$a = \sqrt{\frac{n}{2} \log \frac{2}{\varepsilon}}. \tag{56}$$

Then with probability at least $1 - 2e^{-\frac{2a^2}{n}} = 1 - \varepsilon$, it holds

$$\left|X^u - \mathbb{E}[X^u \mid \omega]\right| < a. \tag{57}$$

Consider the difference between $\mathbb{E}[X^u \mid \omega = u]$ and $\mathbb{E}[X^u \mid \omega = 1 - u]$. By (53) and (54), we have

$$\mathbb{E}[X^u \mid \omega = 1 - u] \leq \frac{1}{1 + \gamma} \mathbb{E}[X^u \mid \omega = u] = \left(1 - \frac{\gamma}{1 + \gamma}\right) \mathbb{E}[X^u \mid \omega = u].$$

By the assumption $\mathbb{E}[X^u \mid \omega = u] \geq \frac{n}{4}$,

$$\mathbb{E}[X^u \mid \omega = 1 - u] \leq \mathbb{E}[X^u \mid \omega = u] - \frac{\gamma}{1 + \gamma} \cdot \frac{n}{4}.$$

By the assumption $\frac{\gamma}{1+\gamma} \geq 8\sqrt{\frac{2}{n} \log \frac{2}{\varepsilon}}$, we have $\frac{\gamma}{1+\gamma} \cdot \frac{n}{4} \geq 8\sqrt{\frac{2}{n} \log \frac{2}{\varepsilon}} \cdot \frac{n}{4} = 4\sqrt{\frac{n}{2} \log \frac{2}{\varepsilon}} = 4a$. Therefore

$$\mathbb{E}[X^u \mid \omega = u] - \mathbb{E}[X^u \mid \omega = 1 - u] \geq 4a. \tag{58}$$

Because we already had $\left|X^u - \mathbb{E}[X^u \mid \omega]\right| < a$ (which happened with probability at least $1 - \varepsilon$), if $X^u$ turns out to be closer to $\mathbb{E}[X^u \mid \omega = u]$ it must be that $\omega = u$; if $X^u$ turns out to be closer to $\mathbb{E}[X^u \mid \omega = 1 - u]$ it must be that $\omega = 1 - u$. In either case, our output $f_{\text{hypo}}(\boldsymbol{r})$ is equal to

$\omega$, having a loss 0. If $\left|X^u - \mathbb{E}[X^u \mid \omega]\right| < a$ did not happen, our loss is at most 1. Therefore, the expected loss of our aggregator $f_{\text{hypo}}$ is at most

$$L_P(f_{\text{hypo}}) = \mathbb{E}_\omega\Big[\mathbb{E}\big[|f_{\text{hypo}}(\boldsymbol{r}) - \omega|^2 \mid \omega\big]\Big] \leq \mathbb{E}_\omega\Big[(1-\varepsilon)\cdot 0 + \varepsilon \cdot 1\Big] = \varepsilon.$$

Since the expected loss of the optimal aggregator $f^*$ is non-negative, $f_{\text{hypo}}$ is $\varepsilon$-optimal. $\qquad\square$

In the remaining proof, we show how to use samples to learn a "real" aggregator $\hat{f}$ that implements the same functionality as the hypothetical aggregator $f_{\text{hypo}}$ and hence is $\varepsilon$-optimal. We have two learning tasks: First, we need to *estimate* $\rho$, so that we can implement the step (1) of $f_{\text{hypo}}$ which tells apart $\frac{r_i}{1-r_i} \geq (1+\gamma)\rho$ and $\frac{r_i}{1-r_i} \leq \frac{1}{1+\gamma}\rho$. Second, we need to *find an index* $u \in \{0,1\}$ *such that* $\mathbb{E}[X^u \mid \omega = u] \geq \frac{n}{4}$ *and estimate* $\mathbb{E}[X^u \mid \omega = u]$, so that we can implement the step (2) of $f_{\text{hypo}}$ which tells whether $X^u$ is closer to $\mathbb{E}[X^u \mid \omega = u]$ or $\mathbb{E}[X^u \mid \omega = 1-u]$. We show that these two tasks can be achieved using $O(\frac{1}{\varepsilon n(\frac{\gamma}{1+\gamma})^2}\log\frac{1}{\delta} + \frac{1}{\varepsilon}\log\frac{1}{\delta})$ samples, with probability at least $1 - O(\delta)$.

**Task 1: estimate $\rho$, using $T_1 = O(\frac{1}{\varepsilon n(\frac{\gamma}{1+\gamma})^2}\log\frac{1}{\delta} + \frac{1}{\varepsilon}\log\frac{1}{\delta})$ samples.** We want to use samples to obtain an estimate $\hat{\rho}$ of $\rho$ such that $\frac{1}{1+\gamma}\rho < \hat{\rho} < (1+\gamma)\rho$. So, by checking whether $\frac{r_i}{1-r_i} > \hat{\rho}$ or $\frac{r_i}{1-r_i} < \hat{\rho}$ we can tell apart $\frac{r_i}{1-r_i} \geq (1+\gamma)\rho$ and $\frac{r_i}{1-r_i} \leq \frac{1}{1+\gamma}\rho$. Using Lemma G.4 with $\Delta = \frac{\gamma}{1+\gamma}$, we obtain a $\hat{\rho}$ such that

$$\hat{\rho} \in (1 \pm \Delta)\rho,$$

with probability at least $1 - \delta$ using

$$T_1 = O\left(\frac{1}{(1-p)\mu_0 n\Delta^2}\log\frac{1}{\delta} + \frac{1}{\min\{p, 1-p\}}\log\frac{1}{\delta}\right) \leq O\left(\frac{1}{\varepsilon n(\frac{\gamma}{1+\gamma})^2}\log\frac{1}{\delta} + \frac{1}{\varepsilon}\log\frac{1}{\delta}\right)$$

samples (recall that we have $\min\{p, 1-p\} \geq (1-p)\mu_0 = p\mu_1 \geq \frac{\varepsilon}{4}$). The $\hat{\rho}$ then satisfies

$$\hat{\rho} < \left(1 + \frac{\gamma}{1+\gamma}\right)\rho < (1+\gamma)\rho \quad\text{and}\quad \hat{\rho} > \left(1 - \frac{\gamma}{1+\gamma}\right)\rho = \frac{1}{1+\gamma}\rho,$$

as desired.

**Task 2: find $u$ such that $\mathbb{E}[X^u \mid \omega = u] \geq \frac{n}{4}$ and estimate $\mathbb{E}[X^u \mid \omega = u]$, using $T_2 = O(\frac{1}{\varepsilon}\log\frac{1}{\delta})$ samples.** First, we show how to use $T_2 = O(\frac{1}{\varepsilon}\log\frac{1}{\delta})$ samples to estimate both $\mathbb{E}[X^0 \mid \omega = 0]$ and $\mathbb{E}[X^1 \mid \omega = 1]$ with accuracy $a = \sqrt{\frac{n}{2}\log\frac{2}{\varepsilon}}$. By the same argument as in the proof of Lemma G.4 (Equations 48 and 49), we know that with probability at least $1 - 2\delta$ over the random draws of

$$T_2 \geq \frac{12}{\min\{p, 1-p\}}\log\frac{2}{\delta} \tag{59}$$

samples, the numbers of samples $(\omega^{(t)}, r_1^{(t)}, \ldots, r_n^{(t)})$'s where $\omega^{(t)} = 0$ and $\omega^{(t)} = 1$, denoted by $\#_0$ and $\#_1$, must satisfy

$$\#_0 \geq \frac{1}{2}(1-p)T_2, \qquad \#_1 \geq \frac{1}{2}pT_2.$$

We consider the samples where $\omega^{(t)} = 0$. There are $\#_0 n$ total number of $r_i^{(t)}$'s. Suppose we have accomplished Task 1. Then, for each $r_i^{(t)}$, we can tell whether the corresponding signal $s_i^{(t)}$ belongs to $\mathcal{S}_i^0$ by checking whether $\frac{r_i^{(t)}}{1-r_i^{(t)}} < \hat{\rho}$. So, we can exactly compute the total number of such signals in the $t$-th sample, $X^{0(t)} = \sum_{i=1}^n \mathbb{1}\{s_i^{(t)} \in \mathcal{S}_i^0\}$, whose expected value is $\mathbb{E}[X^0 \mid \omega = 0]$. Because signals are independent given $\omega^{(t)} = 0$, by Hoeffding's inequality we have

$$\Pr\left[\Big|\underbrace{\sum_{t:\omega^{(t)}=0}\sum_{i=1}^n \mathbb{1}\{s_i^{(t)} \in \mathcal{S}_i^0\}}_{X^{0(t)}} - \#_0\mathbb{E}[X^0 \mid \omega = 0]\Big| \geq \#_0 a\right] \leq 2e^{-\frac{2(\#_0 a)^2}{\#_0 n}} = 2e^{-\frac{2\#_0 a^2}{n}}.$$

Plugging in $a = \sqrt{\frac{n}{2} \log \frac{2}{\varepsilon}}$ and $\#_0 \geq \frac{1}{2}(1-p)T_2$, we get

$$\Pr\left[\left|\frac{1}{\#_0} \sum_{t:\omega^{(t)}=0} X^{0(t)} - \mathbb{E}[X^0 \mid \omega = 0]\right| \geq a\right] \leq 2e^{-\#_0 \log \frac{2}{\varepsilon}} \leq 2e^{-\frac{1}{2}(1-p)T_2 \log \frac{2}{\varepsilon}}.$$

Similarly, considering the samples where $\omega^{(t)} = 1$, we get

$$\Pr\left[\left|\frac{1}{\#_1} \sum_{t:\omega^{(t)}=1} X^{1(t)} - \mathbb{E}[X^1 \mid \omega = 1]\right| \geq a\right] \leq 2e^{-\#_1 \log \frac{2}{\varepsilon}} \leq 2e^{-\frac{1}{2}pT_2 \log \frac{2}{\varepsilon}}.$$

Therefore, if we require

$$T_2 \geq \frac{2\log(2/\delta)}{\min\{p, 1-p\}\log(2/\varepsilon)}, \tag{60}$$

then with probability at least $1 - 2\delta$, both

$$\left|\frac{1}{\#_0} \sum_{t:\omega^{(t)}=0} X^{0(t)} - \mathbb{E}[X^0 \mid \omega = 0]\right| < a, \qquad \left|\frac{1}{\#_1} \sum_{t:\omega^{(t)}=1} X^{1(t)} - \mathbb{E}[X^1 \mid \omega = 1]\right| < a$$

hold. Namely, $\frac{1}{\#_0} \sum_{t:\omega^{(t)}=0} X^{0(t)}$ and $\frac{1}{\#_1} \sum_{t:\omega^{(t)}=1} X^{1(t)}$ are $a$-accurate estimates of $\mathbb{E}[X^0 \mid \omega = 0]$ and $\mathbb{E}[X^1 \mid \omega = 1]$. Equations (59) and (60) together imply that $T_2 = O(\frac{1}{\min\{p,1-p\}} \log \frac{2}{\delta}) \leq O(\frac{1}{\varepsilon} \log \frac{1}{\delta})$ samples suffice.

Then, we identify an index $u \in \{0, 1\}$ such that $\mathbb{E}[X^u \mid \omega = u] \geq \frac{n}{4}$. By Claim G.5, there exists a $v \in \{0, 1\}$ with $\mathbb{E}[X^v \mid \omega = v] \geq \frac{n}{2}$. Since $\frac{1}{\#_v} \sum_{t:\omega^{(t)}=v} X^{v(t)}$ is an $a$-accurate estimate of $\mathbb{E}[X^v \mid \omega = v]$, we must have

$$\frac{1}{\#_v} \sum_{t:\omega^{(t)}=v} X^{v(t)} \geq \mathbb{E}[X^v \mid \omega = v] - a \geq \frac{n}{2} - a.$$

So, at least one of $u \in \{0, 1\}$ must satisfy $\frac{1}{\#_u} \sum_{t:\omega^{(t)}=u} X^{u(t)} \geq \frac{n}{2} - a$. By picking any such a $u$, we are guaranteed that $\mathbb{E}[X^u \mid \omega = u] \geq \frac{1}{\#_u} \sum_{t:\omega^{(t)}=u} X^{u(t)} - a \geq \frac{n}{2} - 2a$. Given the assumption $n \geq 32 \log \frac{2}{\varepsilon}$ in the statement of the theorem, we have

$$\frac{a}{n} = \sqrt{\frac{1}{2n} \log \frac{2}{\varepsilon}} \leq \frac{1}{8}.$$

Hence, $\mathbb{E}[X^u \mid \omega = u] \geq \frac{n}{2} - 2a \geq \frac{n}{2} - 2(\frac{n}{8}) = \frac{n}{4}$.

Finally, as argued above, an $a$-accurate estimate of $\mathbb{E}[X^u \mid \omega = u]$ is given by $\frac{1}{\#_u} \sum_{t:\omega^{(t)}=u} X^{u(t)}$.

**Constructing $\hat{f}$.** After accomplishing Tasks 1 and 2 using $T_1 + T_2 = O(\frac{1}{\varepsilon n(\frac{\gamma}{1+\gamma})^2} \log \frac{1}{\delta} + \frac{1}{\varepsilon} \log \frac{1}{\delta})$ samples, we construct a $\hat{f}$ that implements the same functionality as $f_{\text{hypo}}$. Let

$$M = \frac{1}{\#_u} \sum_{t:\omega^{(t)}=u} X^{u(t)} - 2a,$$

where $\frac{1}{\#_u} \sum_{t:\omega^{(t)}=u} X^{u(t)}$ is our estimate of $\mathbb{E}[X^u \mid \omega = u]$ in Task 2 and $a = \sqrt{\frac{n}{2} \log \frac{2}{\varepsilon}}$.

**Claim G.7.** $\mathbb{E}[X^u \mid \omega = u] - a > M > \mathbb{E}[X^u \mid \omega = 1 - u] + a$.

*Proof.* Because $\frac{1}{\#_u} \sum_{t:\omega^{(t)}=u} X^{u(t)}$ is an $a$-accurate estimate of $\mathbb{E}[X^u \mid \omega = u]$, we have

$$\mathbb{E}[X^u \mid \omega = u] > \frac{1}{\#_u} \sum_{t:\omega^{(t)}=u} X^{u(t)} - a = M + a.$$

Recall from Equation (58) that $\mathbb{E}[X^u \mid \omega = 1 - u] \leq \mathbb{E}[X^u \mid \omega = u] - 4a$. So,

$$\mathbb{E}[X^u \mid \omega = 1 - u] < \left(\frac{1}{\#_u} \sum_{t:\omega^{(t)}=u} X^{u(t)} + a\right) - 4a = M - a.$$

The above two inequalities prove the claim. □

Given reports $\boldsymbol{r} = (r_1, \ldots, r_n)$ as input, we let $\hat{f}$ do the following:

(1) If $u = 1$, count how many reports $r_i$'s satisfy $\frac{r_i}{1-r_i} > \hat{\rho}$; If $u = 0$, count how many reports $r_i$'s satisfy $\frac{r_i}{1-r_i} < \hat{\rho}$. Let this number be $X$;

(2) Then, check whether $X > M$ or $X \leq M$. If $X > M$, output $\hat{f}(\boldsymbol{r}) = u$; otherwise, output $\hat{f}(\boldsymbol{r}) = 1 - u$.

We argue that $\hat{f}$ implements the same functionality as $f_{\text{hypo}}$: (1) In Task 1 we got $\frac{1}{1+\gamma}\rho < \hat{\rho} < (1+\gamma)\rho$. So, by checking whether $\frac{r_i}{1-r_i} > \hat{\rho}$ or $< \hat{\rho}$ we can exactly tell whether $\frac{r_i}{1-r_i} \geq (1+\gamma)\rho$ or $\leq \frac{1}{1+\gamma}\rho$. Hence, we have $X = X^u$, the number of signals that belong to the $\mathcal{S}_i^u$ sets. (2) Recall from Equation (57) that with probability at least $1 - \varepsilon$, $X$ is $a$-close to its expectation $\mathbb{E}[X^u \mid \omega]$. Then, according to Claim G.7, $X > M$ implies that $X$ is closer to $\mathbb{E}[X^u \mid \omega = u]$; $X < M$ implies that $X$ is closer to $\mathbb{E}[X^u \mid \omega = 1 - u]$. So, $\hat{f}$ implements both of the two steps in $f_{\text{hypo}}$. Hence, according to Claim G.6, $\hat{f}$ is $\varepsilon$-optimal.

### G.2 Proof of Theorem B.3

According to Lemma 5.1, the optimal aggregator is

$$f^*(\boldsymbol{r}) = \frac{1}{1 + \rho^{n-1} \prod_{i=1}^{n} \frac{1-r_i}{r_i}}, \tag{61}$$

where $\rho = \frac{p}{1-p}$. We claim that an approximately optimal aggregator can be obtained by first estimating $\rho$ from samples and then use the aggregator with the estimate $\hat{\rho}$:

$$\hat{f}(\boldsymbol{r}) = \frac{1}{1 + \hat{\rho}^{n-1} \prod_{i=1}^{n} \frac{1-r_i}{r_i}}. \tag{62}$$

**Claim G.8.** *If $\frac{|\hat{\rho}-\rho|}{\rho} \leq \frac{2\sqrt{\varepsilon}}{n-1} < \frac{1}{2}$, then the aggregator $\hat{f}$ defined above is $\varepsilon$-optimal.*

*Proof.* Consider the function $g(\hat{\rho}) = \frac{1}{1 + \hat{\rho}^{n-1} \prod_{i=1}^{n} \frac{1-r_i}{r_i}}$ (where $\hat{\rho}$ is the variable and $r_i$'s are constants). We claim that

$$|g'(\hat{\rho})| \leq \frac{n-1}{4\hat{\rho}}. \tag{63}$$

To see this, we note that if $r_i = 0$ for some $i$ then $g(\hat{\rho}) = 0$ and $g'(\hat{\rho}) = 0$. Otherwise, we let $y = \prod_{i=1}^{n} \frac{1-r_i}{r_i} < +\infty$ and take the derivative with respect to $\hat{\rho}$,

$$g'(\hat{\rho}) = -\frac{1}{(1 + \hat{\rho}^{n-1}y)^2}(n-1)\hat{\rho}^{n-2}y = -(n-1)\frac{(\hat{\rho}^{\frac{n}{2}-1}\sqrt{y})^2}{(1 + \hat{\rho}^{n-1}y)^2} = -(n-1)\frac{1}{\left(\frac{1}{\hat{\rho}^{\frac{n}{2}-1}\sqrt{y}} + \hat{\rho}^{\frac{n}{2}}\sqrt{y}\right)^2}.$$

By the AM-GM inequality $a + b \geq 2\sqrt{ab}$, we get

$$|g'(\hat{\rho})| \leq (n-1)\frac{1}{\left(2\sqrt{\frac{1}{\hat{\rho}^{\frac{n}{2}-1}\sqrt{y}} \cdot \hat{\rho}^{\frac{n}{2}}\sqrt{y}}\right)^2} = (n-1)\frac{1}{4\hat{\rho}},$$

as claimed.

Using (63), we have, for $\hat{\rho} \geq \frac{\rho}{2}$,

$$|\hat{f}(\boldsymbol{r}) - f^*(\boldsymbol{r})| = |g(\hat{\rho}) - g(\rho)| \leq \frac{n-1}{4\min\{\hat{\rho}, \rho\}} \cdot |\hat{\rho} - \rho| \leq \frac{n-1}{2} \cdot \frac{|\hat{\rho} - \rho|}{\rho}. \tag{64}$$

So, to obtain $\varepsilon$-approximation $\mathbb{E}[|\hat{f}(\boldsymbol{r}) - f^*(\boldsymbol{r})|^2] \leq \varepsilon$, we can require $|\hat{f}(\boldsymbol{r}) - f^*(\boldsymbol{r})| \leq \frac{n-1}{2} \cdot \frac{|\hat{\rho}-\rho|}{\rho} \leq \sqrt{\varepsilon}$. This can be satisfied if the error in estimating $\rho$ is at most $\frac{|\hat{\rho}-\rho|}{\rho} \leq \frac{2\sqrt{\varepsilon}}{n-1}$. $\qquad\square$

We then show how to use $O(\frac{\gamma n}{\varepsilon} \log \frac{1}{\delta})$ samples to estimate the value of $\rho$ with $O(\frac{\sqrt{\varepsilon}}{n-1})$ accuracy, which will give us an $\varepsilon$-optimal aggregator according to Claim G.8. Recall from (14) that when signals are $\gamma$-weakly informative, the reports always satisfy

$$\frac{1}{1+\gamma}\rho \leq \frac{r_i}{1-r_i} \leq (1+\gamma)\rho. \tag{65}$$

The following observation is the key:

**Lemma G.9.** *For each expert $i$, we have*

- $\mathbb{E}\big[\frac{r_i}{1-r_i} \mid \omega = 0\big] = \rho$;

- $\mathbb{E}\big[\frac{1-r_i}{r_i} \mid \omega = 1\big] = \frac{1}{\rho}$.

*As corollaries, for $k$ conditionally independent reports $r_1, \ldots, r_k$, we have $\mathbb{E}\big[\prod_{i=1}^{k} \frac{r_i}{1-r_i} \mid \omega = 0\big] = \rho^k$ and $\mathbb{E}\big[\prod_{i=1}^{k} \frac{1-r_i}{r_i} \mid \omega = 1\big] = \frac{1}{\rho^k}$.*

*Proof.* Because $\frac{r_i}{1-r_i} = \frac{P(s_i|\omega=1)}{P(s_i|\omega=0)}\rho$ (from (13)), we have

$$\mathbb{E}\Big[\frac{r_i}{1-r_i} \mid \omega = 0\Big] = \sum_{s_i \in \mathcal{S}_i} P(s_i \mid \omega = 0)\frac{P(s_i \mid \omega = 1)}{P(s_i \mid \omega = 0)}\rho = \sum_{s_i \in \mathcal{S}_i} P(s_i \mid \omega = 1)\rho = \rho.$$

For conditionally independent $r_1, \ldots, r_k$, we have

$$\mathbb{E}\Big[\prod_{i=1}^{k} \frac{r_i}{1-r_i} \mid \omega = 0\Big] = \prod_{i=1}^{k} \mathbb{E}\Big[\frac{r_i}{1-r_i} \mid \omega = 0\Big] = \rho^k.$$

Similarly, we can prove $\mathbb{E}\big[\frac{1-r_i}{r_i} \mid \omega = 1\big] = \frac{1}{\rho}$ and $\mathbb{E}\big[\prod_{i=1}^{k} \frac{1-r_i}{r_i} \mid \omega = 1\big] = \frac{1}{\rho^k}$. $\square$

Let $\Delta = \frac{\sqrt{\varepsilon}}{3\gamma n}$ and suppose we are given $T = \frac{6e}{\gamma n \Delta^2} \log \frac{2}{\delta} = \frac{54e\gamma n}{\varepsilon} \log \frac{2}{\delta} = O(\frac{\gamma n}{\varepsilon} \log \frac{1}{\delta})$ samples. Suppose when drawing the samples we first draw the events $\omega^{(t)}$'s, and then draw the reports $r_i^{(t)}$'s conditioning on $\omega^{(t)}$ being 0 or 1. After the first step, the numbers of samples with $\omega^{(t)} = 0$ and $\omega^{(t)} = 1$ are determined, which we denote by $\#_0$ and $\#_1$. Since $\#_0 + \#_1 = T$, one of them must be at least $T/2$. We argue that whether $\#_0 \geq \frac{T}{2}$ or $\#_1 \geq \frac{T}{2}$ we can estimate $\rho^{1/\gamma}$ with accuracy $3\Delta$. (For simplicity, we assume that $1/\gamma$ is an integer.)

- If $\#_0 \geq T/2$, then we consider the $\#_0 n$ reports $r_i^{(t)}$'s in the samples with $\omega^{(t)} = 0$. We divide these $\#_0 n$ reports evenly into $\#_0 n\gamma$ groups, each of size $1/\gamma$, denoted by $G_1, \ldots, G_{\#_0 n\gamma}$. Consider the product of $\frac{r_i^{(t)}}{1-r_i^{(t)}}$'s in a group $G_j$: because $r_i^{(t)}$'s are independent given $\omega = 0$, by Lemma G.9 we have

$$\mathbb{E}\Big[\prod_{r_i^{(t)} \in G_j} \frac{r_i^{(t)}}{1-r_i^{(t)}} \mid \omega = 0\Big] = \rho^{1/\gamma}.$$

Using (65) and the inequality $(1+\gamma)^{1/\gamma} \leq e$, we have

$$\frac{1}{e}\rho^{1/\gamma} \leq \frac{1}{(1+\gamma)^{1/\gamma}}\rho^{1/\gamma} \leq \prod_{r_i^{(t)} \in G_j} \frac{r_i^{(t)}}{1-r_i^{(t)}} \leq (1+\gamma)^{1/\gamma}\rho^{1/\gamma} \leq e\rho^{1/\gamma}.$$

Let $X_j$ be the random variable $\frac{1}{e\rho^{1/\gamma}}\prod_{r_i^{(t)} \in G_j} \frac{r_i^{(t)}}{1-r_i^{(t)}}$. From the above equation and inequality we have $\mathbb{E}[X_j] = \frac{1}{e}$ and $X_j \in [\frac{1}{e^2}, 1] \subseteq [0, 1]$. So, by Chernoff bound,

$$\Pr\Big[\frac{1}{\#_0 n\gamma}\sum_{j=1}^{\#_0 n\gamma} X_j \in (1 \pm \Delta)\frac{1}{e} \mid \omega = 0\Big] \geq 1 - 2e^{-\frac{\#_0 n\gamma\Delta^2}{3e}} \geq 1 - 2e^{-\frac{Tn\gamma\Delta^2}{6e}} = 1 - \delta,$$

given our choice of $T$. Multiplying $\frac{1}{\#_0 n\gamma} \sum_{j=1}^{\#_0 n\gamma} X_j$ by $e\rho^{1/\gamma}$, we obtain the following estimate of $\rho^{1/\gamma}$:

$$\hat{\rho}_0^{1/\gamma} := \frac{1}{\#_0 n\gamma} \sum_{j=1}^{\#_0 n\gamma} \prod_{r_i^{(t)} \in G_j} \frac{r_i^{(t)}}{1 - r_i^{(t)}} \in (1 \pm \Delta)\rho^{1/\gamma}.$$

Dividing by $\rho^{1/\gamma}$, we get $(\frac{\hat{\rho}_0}{\rho})^{1/\gamma} \in 1 \pm \Delta$.

- If $\#_1 \geq T/2$, then by considering the $\#_1 n$ reports in the samples with $\omega^{(t)} = 1$, dividing them into $\#_1 n\gamma$ groups of size $1/\gamma$, $H_1, \ldots, H_{\#_1 n\gamma}$, and similarly defining random variable $Y_j = \frac{\rho^{1/\gamma}}{e} \prod_{r_i^{(t)} \in H_j} \frac{1 - r_i^{(t)}}{r_i^{(t)}}$, we obtain the following estimate of $(\frac{1}{\rho})^{1/\gamma}$:

$$(\frac{1}{\hat{\rho}_1})^{1/\gamma} := \frac{1}{\#_1 n\gamma} \sum_{j=1}^{\#_1 n\gamma} \prod_{r_i^{(t)} \in H_j} \frac{1 - r_i^{(t)}}{r_i^{(t)}} \in (1 \pm \Delta)(\frac{1}{\rho})^{1/\gamma}.$$

Multiplying by $\rho^{1/\gamma}$, we get $(\frac{\rho}{\hat{\rho}_1})^{1/\gamma} \in 1 \pm \Delta$. Taking the reciprocal and noticing that $\frac{1}{1 \pm \Delta} \subseteq 1 \pm 3\Delta$ when $\Delta < \frac{1}{3}$, we obtain $(\frac{\hat{\rho}_1}{\rho})^{1/\gamma} \in 1 \pm 3\Delta$.

From the discussion above we obtained an estimate $\hat{\rho} \in \{\hat{\rho}_0, \hat{\rho}_1\}$ of $\rho$ such that $(\frac{\hat{\rho}}{\rho})^{1/\gamma} \in 1 \pm 3\Delta$. Raising to the power of $\gamma$, and using the inequality $(1-x)^\gamma \geq 1 - x\gamma$ and $(1+x)^\gamma \leq e^{x\gamma} \leq 1 + 2x\gamma$ for $x\gamma \leq 1$, we get

$$\frac{\hat{\rho}}{\rho} \in (1 \pm 3\Delta)^\gamma \subseteq [1 - 3\Delta\gamma, \, e^{3\Delta\gamma}] \subseteq [1 - 3\Delta\gamma, \, 1 + 6\Delta\gamma].$$

In particular, this implies $\frac{|\hat{\rho} - \rho|}{\rho} \leq 6\Delta\gamma = \frac{2\sqrt{\varepsilon}}{n}$. Then, according to Claim G.8, the aggregator $\hat{f}$ defined by $\hat{f}(\boldsymbol{r}) = \frac{1}{1 + \hat{\rho}^{n-1} \prod_{i=1}^{n} \frac{1 - r_i}{r_i}}$ is $\varepsilon$-optimal. We hence obtained an $\varepsilon$-optimal aggregator.

# H   Missing Proofs from Section 7

## H.1   Proof of Theorem 7.1

Regard $P$ as a joint distribution over reports $\boldsymbol{r} = (r_1, \ldots, r_n)$ and the state $\omega$, where $r_i$ is sampled by first sampling $s_i \in \mathcal{S}_i$ and then letting $r_i = P(\omega = 1 \mid s_i)$. Since $|\mathcal{S}_i| = m$, there are at most $m$ different values of $r_i$ that can be sampled, so there are at most $2m^n$ different tuples of $(\boldsymbol{r}, \omega)$ in the support of $P$. For each such tuple $(\boldsymbol{r}, \omega)$, consider the empirical probability of this tuple:

$$\hat{P}(\boldsymbol{r}, \omega) = \frac{1}{T} \sum_{t=1}^{T} \mathbb{1}[(\boldsymbol{r}^{(t)}, \omega^{(t)}) = (\boldsymbol{r}, \omega)].$$

By the Chernoff bound, we have

$$\Pr\left[|\hat{P}(\boldsymbol{r}, \omega) - P(\boldsymbol{r}, \omega)| > \Delta P(\boldsymbol{r}, \omega)\right] \leq 2e^{-\frac{TP(\boldsymbol{r},\omega)\Delta^2}{3}}.$$

Using a union bound for all the $2m^n$ tuples and the fact that $P(\boldsymbol{r}, \omega) \geq P(\boldsymbol{s}, \omega) > \frac{c}{m^n}$ (where $\boldsymbol{s} \in \mathcal{S}$ are some signals that generate $\boldsymbol{r}$), we have

$$|\hat{P}(\boldsymbol{r}, \omega) - P(\boldsymbol{r}, \omega)| \leq \Delta P(\boldsymbol{r}, \omega) \tag{66}$$

holds for all tuples $(\boldsymbol{r}, \omega)$ except with probability at most

$$2m^n \cdot 2e^{-\frac{TP(\boldsymbol{r},\omega)\Delta^2}{3}} \leq 4m^n \cdot e^{-\frac{cT\Delta^2}{3m^n}} = \delta$$

if

$$T \geq \frac{3m^n}{c\Delta^2} \log \frac{4m^n}{\delta}. \tag{67}$$

Assuming (66) holds, we consider the "empirical" Bayesian aggregator:

$$\hat{f}(\boldsymbol{r}) = \frac{\hat{P}(\boldsymbol{r}, \omega)}{\hat{P}(\boldsymbol{r})}.$$

Since (66) implies $\hat{P}(\boldsymbol{r}, \omega) \in (1 \pm \Delta)P(\boldsymbol{r}, \omega)$ and $\hat{P}(\boldsymbol{r}) \in (1 \pm \Delta)P(\boldsymbol{r})$, we have

$$\hat{f}(\boldsymbol{r}) \geq \frac{1 - \Delta}{1 + \Delta} \cdot \frac{P(\boldsymbol{r}, \omega)}{P(\boldsymbol{r})} = \left(1 - \frac{2\Delta}{1 + \Delta}\right)f^*(\boldsymbol{r}) \geq (1 - 2\Delta)f^*(\boldsymbol{r})$$

and

$$\hat{f}(\boldsymbol{r}) \leq \frac{1 + \Delta}{1 - \Delta} \cdot \frac{P(\boldsymbol{r}, \omega)}{P(\boldsymbol{r})} = \left(1 + \frac{2\Delta}{1 - \Delta}\right)f^*(\boldsymbol{r}) \leq (1 + 4\Delta)f^*(\boldsymbol{r}),$$

if $\Delta < \frac{1}{2}$. Putting these two inequalities together:

$$|\hat{f}(\boldsymbol{r}) - f^*(\boldsymbol{r})| \leq 4\Delta f^*(\boldsymbol{r}).$$

We note that this holds for all possible $\boldsymbol{r}$ in the support of $P$. So, the expected approximation error of $\hat{f}$ is at most:

$$
\begin{aligned}
\mathbb{E}\big[|\hat{f}(\boldsymbol{r}) - f^*(\boldsymbol{r})|^2\big] &= \sum_{\boldsymbol{r}} P(\boldsymbol{r})|\hat{f}(\boldsymbol{r}) - f^*(\boldsymbol{r})|^2 \\
&\leq \sum_{\boldsymbol{r}} P(\boldsymbol{r}) 16\Delta^2 f^*(\boldsymbol{r})^2 \\
&= 16\Delta^2 \sum_{\boldsymbol{r}} P(\boldsymbol{r})\left(\frac{P(\boldsymbol{r}, \omega)}{P(\boldsymbol{r})}\right)^2 \\
&= 16\Delta^2 \sum_{\boldsymbol{r}} \frac{P(\boldsymbol{r}, \omega)^2}{P(\boldsymbol{r})} \leq 16\Delta^2 \sum_{\boldsymbol{r}} \frac{P(\boldsymbol{r})^2}{P(\boldsymbol{r})} = 16\Delta^2.
\end{aligned}
$$

Letting $16\Delta^2 = \varepsilon$, namely $\Delta^2 = \frac{\varepsilon}{16}$, we have $\mathbb{E}\big[|\hat{f}(\boldsymbol{r}) - f^*(\boldsymbol{r})|^2\big] \leq \varepsilon$, so $\hat{f}$ is an $\varepsilon$-optimal aggregator. Plugging $\Delta^2 = \frac{\varepsilon}{16}$ into (67) we obtain the sample complexity

$$\frac{3m^n}{c\Delta^2} \log \frac{m^n}{\delta} = \frac{48m^n}{c\varepsilon} \log \frac{m^n}{\delta} = O\left(\frac{m^n}{c\varepsilon}\left(n\log m + \log\frac{1}{\delta}\right)\right).$$

