# OpenReview forum: "Sample Complexity of Forecast Aggregation"
_NeurIPS.cc/2023/Conference — NeurIPS 2023 spotlight_

### Official Review · Reviewer_bpux · 2023-07-01

**Soundness:** 3 good
**Presentation:** 3 good
**Contribution:** 3 good
**Rating:** 7
**Confidence:** 2

**Summary:**

The authors study the problem of forecast aggregation in a Bayesian setting. Here n experts observe an individual signal that is correlated with the truth, and each expert reports their posterior belief to the principal. The principal aggregates these reports and outputs a prediction, whose quality is assessed using square loss. The object of interest is the minimax sample complexity necessary to obtain an additive excess error of $\epsilon$ compared to the best aggregator that knows the conditional distribution of truth given reports.

The authors derive results for arbitrary distributions $P(\omega, s)$ on truth and signals, as well as distributions that factorize as $P(\omega,s) = P(\omega) \prod_i P(s_i|\omega)$. They show that there is an exponential gap in sample complexity between these two cases, and in the latter case the complexity does not depend on the number of experts $n$.

**Strengths:**

- Well written and appears technically sound
- Introduces the study of sample complexity to the forecast aggregation literature
- Interesting and general results that cover multiple natural settings
- A novel lower bound construction for distribution estimation that allows reduction from estimation to aggregation

**Weaknesses:**

- Non-matching upper and lower bounds in most cases
- Main technical difficulty lies in construction of lower bound, the upper bounds don't require significant new ideas
- Techniques seem limited to the specific loss function + discrete truth/signal setting. Commenting on possible extensions in these directions would be interesting

**Questions:**

Have the authors thought about results for alternative loss functions (e.g. different weight for type2 vs type1 errors) or continuous truth/signal distributions where for example the experts report the condition mean of the truth given their signal?

**Limitations:**

The authors address limitations.

---

> ### Author Rebuttal · Authors · 2023-08-09
>
> **Alternative loss functions:**
>
> Our response to Reviewer YcHV mentions several reasons why we focus on the squared loss $E[ | f(r) - \omega |^2 ]$ in this work.
> We did think about other loss functions like the logarithmic loss $E[(\omega \log(f(r)) + (1-\omega)\log(1-f(r)))]$ and the absolute loss $E[ | f(r) - \omega | ]$.  Some of our techniques for the squared loss can be applied to other loss functions.
>
> *Logarithmic loss:* Because the logarithmic loss can be unbounded, giving an upper bound on its sample complexity is not easy and may need significantly different techniques than our current work.
> For the lower bound, our results for the squared loss can be applied.  Due to Pinsker's inequality, the logarithmic loss difference is greater than or equal to 2 times the squared loss difference, so if we get an $\epsilon$-optimal aggregator under the logarithmic loss, then this aggregator is automatically $\epsilon/2$-optimal under the squared loss. This means that our sample complexity lower bound for the squared loss is automatically a sample complexity lower bound for the logarithmic loss.
>
> *Absolute loss:*  Our upper bound argument for the squared loss can be adapted to the absolute loss.  But the lower bound argument cannot. The squared loss has the property that the difference between the squared losses of any aggregator and the optimal aggregator can be conveniently written as their expected squared difference: $E[|f(r)-\omega|^2] - E[|f^*(r)-\omega|^2] = E[|f(r)-f^*(r)|^2]$ (Lemma 2.1).  We use this property in the argument.
> But the absolute loss $E[ | f(r) - \omega | ]$ does not have this property, and its sample complexity lower bound may need a different argument.
>
> We believe that analyzing other loss functions is an interesting direction for future works.
>
> **Continuous signal/truth distribution:**
>
> We did think about continuous distributions for signal or truth.
> For continuous signal distribution, our results for conditionally independent distributions do cover this case -- we make no assumption on the signal space.  For not conditionally independent distributions, note that our results show that the sample complexity grows with the size of the discrete signal space.  This means that with a continuous signal space the sample complexity is infinite in the worst case.  So, to obtain a meaningful sample complexity result, one has to make some assumptions on the continuous signal distribution, for example, the distribution belongs to some parameterized family.  Then, one may obtain very different results than the results in this paper, using different techniques.
>
> With continuous truth distribution, the problem will be very different if the experts only report the posterior means of the truth instead of the posterior distributions of the truth.  And similar to continuous signals, to obtain a meaningful result one has to make some assumptions on the continuous truth distribution.  Depending on different assumptions one may get different results. We believe continuous truth/signal distribution is an interesting direction to explore for future works.

---

### Official Review · Reviewer_YcHV · 2023-07-04

**Soundness:** 4 excellent
**Presentation:** 3 good
**Contribution:** 3 good
**Rating:** 7
**Confidence:** 4

**Summary:**

The paper initiates the study of sample complexity of forecast aggregation under a Bayesian forecasting model. In this problem, n experts, each observe a private signal about an unknown binary event, and then report their posterior beliefs about the event to a principal, who then aggregates the reports into a single prediction. The underlying joint distribution is unknown to the principal, but he has access to i.i.d. samples from the distribution. Using these samples, the principal aims to find an epsilon-approximate optimal aggregator, where optimality is measured in terms of the mean squared error between the aggregated prediction and the real event. The authors show that the sample complexity grows exponentially in the number of experts n, but that if the experts’ signals are conditionally independent, then the sample complexity does not depend on the number of experts at all. They further consider the case of non-binary events and weakly/strongly informative experts.

**Strengths:**

The paper is very elegantly written. It presents the setup of the problem clearly, motivates it thoroughly, and initiates an interesting discussion on the fundamental limits of the problem. The proof sketches are quite intuitive and convincing and their implications are reflected and discussed.

**Weaknesses:**

One weakness for me is in the particular choice of mean squared error as an optimality measure. It seems counter-intuitive, given that the experts report their posterior beliefs, which is in essence a minimum error probability optimality measure. It would seem more natural for the principal to look for an aggregation that minimizes the probability of error given the exerts reports. Another weakness is the gap between the upper and lower bounds with respect to epsilon, which might follow from the relatively simple upper bound proposed in the paper. This also bleeds over to the conditionally independent variant. Finally, the choice of averaged squared error as an optimality measure in the multi-outcome events section brings forth quite bizarre looking results. It follows, under this measure, that is the number of events is omega(1/epsilon^2) our task succeeds with zero samples! The authors do clarify this in the appendix, but I would remove it altogether from the paper (or change the measure to additive MSE).

**Questions:**

Did you consider using the maximum a posterior optimality measure for the principal? An added explanation on why MSE is a good choice can benefit your work.

**Limitations:**

The authors adequately addressed the limitations of their work

---

> ### Author Rebuttal · Authors · 2023-08-09
>
> **Response to Questions:**
>
> Maximum a posterior (MAP), or posterior mode, is the point estimate $f(r)$ that minimizes the absolute loss $E[ | f(r) - \omega | ]$.
> We choose to use the squared loss $E[ | f(r) - \omega |^2 ]$ instead in our work for a few reasons:
>
> (1) Squared loss is a very popular loss function used in many learning problems.
>
> (2) Squared loss is a proper loss function for eliciting probability distributions.  Our model requires the experts to report posterior beliefs to the principal.  To incentivize the experts to do so, the principal needs to reward the experts according to a proper loss, for example the squared loss $-E[|r_i - \omega|^2]$.  The experts maximize their expected rewards by reporting their beliefs (posterior distribution of $\omega$) truthfully to the principal.  (We wrote this in Footnote 3.)  So, to be consistent with the expert's loss, we measure the principal's loss by the squared loss $E[|f(r) - \omega|^2]$ as well.  A non-proper loss like the absolute loss $-E[|r_i - \omega|]$, on the other hand, does not elicit the beliefs from the experts.
>
> (3) Squared loss has the property that the difference between the squared losses of any aggregator and the optimal aggregator can be conveniently written as their expected squared difference: $E[|f(r)-\omega|^2] - E[|f^*(r)-\omega|^2] = E[|f(r)-f^*(r)|^2]$ (Lemma 2.1).  This property is used in our sample complexity lower bound argument.  The absolute loss doesn't have this property and its sample complexity lower bound may need a different argument.
>
> We think it will indeed be interesting to consider other losses as directions of future work, for example the absolute loss $E[|f(r)-\omega|]$, which is minimized by posterior mode (MAP).
>
>
> **Response to Weaknesses:**
>
> Regarding the ``bizarre looking result" in the multi-outcome case, this result is indeed an artifact of the use of the *average* squared loss (where we divide the loss by $|\Omega|$).  (You can see our response to Reviewer ey4x for details.)  We will change the loss to the additive squared loss (not dividing $|\Omega|$) to avoid this confusion.  Thanks for suggesting this!

---

> > ### Comment · Reviewer_YcHV · 2023-08-11
> > **Response to rebuttal**
> >
> > I have read the rebuttal, the authors addressed the comments and questions raised thoroughly.

---

### Official Review · Reviewer_ey4x · 2023-07-05

**Soundness:** 3 good
**Presentation:** 4 excellent
**Contribution:** 4 excellent
**Rating:** 8
**Confidence:** 4

**Summary:**

The paper studies the aggregation of expert opinions in a Bayesian setting for discrete (binary) distributions. The setting further limits each expert to have at most m different opinions, making the problem fully discrete. This reduction enables the analysis of the sampling complexity of the problem, i.e., the minimum number of observed aggregation events (expert opinions and true outcome) required to get an epsilon tight estimate on the true distriution in the total variation norm.

**Strengths:**

The paper approaches the more and more important problem of how to combine different "expert" opinions in a thorough and interesting way. It is well written, clear and, as far as I managed to go into detail, correct.
Their approach sparks many new ideas on how to approach aggregation problems and opens up interesting questions on the relevance of (historical) data in aggregation.

**Weaknesses:**

The proof of the lower bound in Section 4.2 doesn't seem intuitive. The mere construction of an example is of course sufficient for a formal proof, but an explanation of why the authors had the idea for this specific construction might be more illuminating.
The authors sometimes speak of the "number of signals an expert can possibly observe." Maybe this is better explained as the "cardinality of the signal the experts observe"

**Questions:**

Is it essential in the proof that the experts decide on Bayesian arguments? Or does it suffice to assume that each expert always just makes one of m possible reports, i.e., can we focus only on the joint distribution of r^t and omega and ignore s^t?
In Section 6, does bigger Omega really reduce the complexity? This is counterintuitive.
What is the source of Theorem 7.1?

**Limitations:**

The authors clearly state the limitations of their work, are honest and clear about the current mismatch in their bounds in epsilon, and don't oversell their results.
I do not see potential societal impact.

---

> ### Author Rebuttal · Authors · 2023-08-09
>
> 1. Experts decide on Bayesian arguments?  This is an interesting observation!  Indeed, our results for the case of general distributions do not need the experts to be Bayesian.  It is OK if an expert reports 0.1 even if its posterior belief given signal is 0.8, as long as the expert reports different numbers given different siganls.  What matters is the correlation between all experts' reported numbers and the event outcome.  The principal can learn such correlation from samples and then do aggregation.  However, for the case of conditionally independent distributions, we do need the experts to be Bayesian in order to apply Lemma 5.1 to get the smaller sample complexity upper bound.
>
> 2. Does $|\Omega|$ really reduce the sample complexity?  No.  As we explain in the end of Appendix C, this is an artifact of normalization.  When defining the loss of an aggregator for multi-outcome events, we normalize the loss by dividing $|\Omega|$ to make sure the loss is bounded in $[0, 1]$.  But for an aggregator that always outputs a probability distribution, its loss is actually bounded by $[0, 1/|\Omega|]$, causing the illusion that the sample complexity decreases as $|\Omega|$ increases.  We will change the loss to the unnormalized loss (not dividing $|\Omega|$) to avoid this confusion.
>
> 3. Theorem 7.1.  The source of this theorem is that the minimum joint probability $P(s, \omega)$ is bounded away from $0$ (by a margin of $c/m^n$ with constant $c$).  In the proof of this theorem (Appendix I) we used the "empirical Bayes aggregator" $\hat f(r) = \frac{\hat P(r, \omega)}{\hat P(r)}$, namely, the Bayes rule with empirical probability estimated from samples.  For this aggregator to work well, the denominator $\hat P(r)$ must be away from $0$.  This is guaranteed if the minimum joint probability is away from $0$.  In contrast, Theorem 4.1, which is more general but has a looser upper bound, does not need this assumption.

---

> > ### Comment · Reviewer_ey4x · 2023-08-16
> >
> > Thank you for your detailed answer, I don't have any further questions.

---

### Official Review · Reviewer_mdHA · 2023-07-06

**Soundness:** 3 good
**Presentation:** 4 excellent
**Contribution:** 3 good
**Rating:** 7
**Confidence:** 4

**Summary:**

This paper studies the problem of forecast aggregation: There are n experts who are each given signals about some unknown binary event, and each output a posterior probability based on these signals. We have access to k such reports (corresponding to k events). The task is to aggregate these reports to produce a prediction that is close to the unknown event in expected squared distance. The paper shows that there is an exponential gap in sample complexity (ie k) between the cases when the experts' signals are independent conditioned on the event, versus the general case. In addition, a (slightly loose) upper bound is provided for the arbitrary case, and a lower bound is provided for the conditionally independent case.

**Strengths:**

This is a very clearly written paper, and it studies an interesting problem. The story of the exponential sample complexity difference in the arbitrary signals vs conditionally independent signal setting is a compelling one. Overall, I think this is an interesting paper that deserves to be accepted to NeurIPS.

More comments:
- It studies a natural extension where the aggregator sees multiple samples from each expert. This is a more realistic setting than the one-shot problem.
- There are natural connections to distribution estimation here that are interesting. For this reason, I think this paper will appeal to the CS theory community.
- The future directions mentioned are compelling, and will likely lead to several interesting results.

**Weaknesses:**

I don't see any real weaknesses.

**Questions:**

- Is it possible to say something about when you expect the signals to be conditionally independent in the real world? For instance, your paper points to a Kaggle dataset - is there any evidence that there is conditional independence there? Do you expect your algorithms to work well on that dataset?
- Can you draw any connections between (variants of) this problem and distribution testing?

**Limitations:**

Yes

---

> ### Author Rebuttal · Authors · 2023-08-09
>
> Question 1:
>
> Signals are conditionally independent if they are independent draws from some distributions but the distributions are determined by the unknown state of the world. For example, the unknown state of the world can be the quality of a school, which is either high or low. If the school quality is high, the probability for a student from the school to pass a state exam is 0.9. If the school quality is low, the probability for a student to pass the state exam is only 0.6. This can be viewed as an example of conditionally independent signals, as given the quality of the school, students' exam outcomes are independent draws from a distribution that depends on the true quality of the school. Conditionally independent signals capture many real-world settings where agents have independent observations but the distribution of the observations depends on the state of the world.
>
> We think Kaggle competitions can be reasonably modeled using conditionally independent signals. However, the dataset contains heterogeneous non-repeated tasks and hence is not a good match for our algorithm, which needs iid repeated tasks.
>
>
> Question 2:
> Many distribution testing problems (e.g., testing whether a distribution is uniform) can be solved by distribution estimation: first use samples to compute the empirical distribution as an estimate of the true distribution, then check whether the empirical distribution satisfies the property we are testing.  This means that distribution estimation is more difficult than (or as difficult as) those distribution testing problems, in the sense that the number of samples needed for distribution testing is <= the number of samples needed for distribution estimation.  Our results show that forecast aggregation for general distributions is essentially as difficult as distribution estimation, so it is also more difficult than those distribution testing problems.  But the forecast aggregation problem for conditionally independent distributions is not directly comparable to distribution estimation and we don't see a direct connection with distribution testing, either.  Nevertheless, testing whether a distribution is conditionally independent is itself a property testing problem.

---

> > ### Comment · Reviewer_mdHA · 2023-08-14
> >
> > Hi,
> >
> > Thanks for your responses. For 2, I was wondering if there's a different but related problem that you can come up with that directly relates to distribution testing (say uniformity testing).
> >
> > In any case, I am impressed with this paper, and maintain my rating.

---

### Decision · Program_Chairs · 2023-09-21

**Decision:**

Accept (spotlight)

**Comment:**

All reviewers are unanimous in their support for this paper. The forecast aggregation problem is no doubt of significant importance, the technical results are solid, and the presentation is excellent. The "path" of this paper, as outlined by potential future works in this direction, is very interesting. This work will be a great contribution to the conference proceedings.